# Improving Self-Supervised Learning by Characterizing Idealized Representations

**Yann Dubois, Tatsunori Hashimoto, Stefano Ermon, Percy Liang**
Stanford University
{yanndubs,thashim,ermon,pliang}@stanford.edu

## Abstract

Despite the empirical successes of self-supervised learning (SSL) methods, it is unclear what characteristics of their representations lead to high downstream accuracies. In this work, we characterize properties that SSL representations should *ideally* satisfy. Specifically, we prove necessary and sufficient conditions such that for any task invariant to given data augmentations, desired probes (e.g., linear or MLP) trained on that representation attain perfect accuracy. These requirements lead to a unifying conceptual framework for improving existing SSL methods and deriving new ones. For contrastive learning, our framework prescribes simple but significant improvements to previous methods such as using asymmetric projection heads. For non-contrastive learning, we use our framework to derive a simple and novel objective. Our resulting SSL algorithms outperform baselines on standard benchmarks, including SwAV+multicrops on linear probing of ImageNet.

## 1 Introduction

We study self-supervised learning (SSL), where the goal is to learn representations from minimal supervision, such that simple probes trained on these representations achieve high downstream accuracy. Recently, there has been many different SSL methods achieving impressive empirical results (e.g. SimCLR [1], SwAV [2]) using label-preserving augmentations (e.g. cropping or color jittering) as supervision. We dub this setting *invariant SSL* (ISSL). Despite these empirical successes, it remains unclear how these various SSL methods relate to one another, how to improve them, and how to derive new ones. Our goal is to provide a simple conceptual framework to think about those questions.

To derive such a framework, we ask ourselves: *what are the ideal requirements that ISSL representations should aim to satisfy?* We prove necessary and sufficient requirements to ensure that probes from a specified family, e.g. linear or multi-layer perceptron (MLP), perfectly classify any task that is invariant to desired data augmentations. This complements theoretical work in ISSL [3–8], which analyze specific ISSL algorithms. Our work instead focuses on properties of representations that should serve as a goal for any ISSL algorithm. These ideal properties are: (i) desired predictors should be able to distinguish positive and negative examples from the representation; (ii) the dimensionality of the representation should be sufficiently large; (iii) augmented inputs should map to the same representations.

Previous ISSL methods can be seen as approximations of our ideal requirements. Using our requirements, we derive simple improvements to those approximations as well as new ISSL objectives. Our theory thus results in a unifying conceptual ISSL framework, with practical prescriptions including:

- improvements to existing methods, such as increasing the dimensionality of representations and using asymmetric projections heads, which lead to $5\%$ point gains on TinyImageNet;
- a novel non-contrastive ISSL objective that outperforms all baselines, including SwAV+multicrops, by at least $1\%$ point on linear classification of ImageNet;
- extensions of ISSL algorithms to learn representations that are better suited for non-linear probes.

36th Conference on Neural Information Processing Systems (NeurIPS 2022).

## 2 Problem statement

### 2.1 Invariant Self-Supervised Learning

SSL learns an encoder $\phi$ that maps an input $x$ from a finite space $\mathcal{X}$ (e.g., $256 \times 256$ images) into a representation $\phi(x) \in \mathbb{R}^d$. Given the encoder and a dataset $\mathcal{D}_t$ drawn from some task of interest $p_t(X, Y)$, we fit a classifier $f$ from a desired family of probes $\mathcal{F}$. Families of probes are sets of $k$-ary classifiers for any $k \geq 2$ such as linear or MLP probes. For clarity, we consider linear probes $\mathcal{F}$ until Sec. 5.

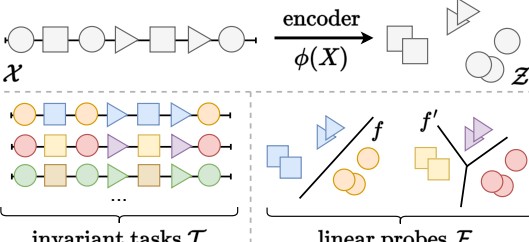

Figure 1: ISSL setting. (Top) 1D inputs, partitioned into 3 equivalence classes (shapes), are encoded by $\phi$ into a 2D representation. (Bot. left) 3 $\sim$-invariant tasks, where labels are the colors. (Bot. right) examples of probes for 2 of the invariant tasks.

Supervision for ISSL comes from unlabeled data $p_x$ and label-preserving augmentations. Augmentations are ways of sampling "positive" $x, x^+$ examples that are equivalent for downstream classification tasks. We formalize this using an equivalence relation $x \sim x^+$ that partitions the inputs $\mathcal{X}$ into equivalent classes $[x] \in \mathcal{X}/\sim$, and we consider the following downstream tasks $\mathcal{T}$ whose labelings are deterministic and constant within these classes (we allow stochastic labeling in appendices).

**Definition 1.** The $\sim$-*invariant tasks* $\mathcal{T}$, is the set of all input-label distributions $p_t(X, Y)$ such that the labeling $p_t(Y|X)$ is deterministic and invariant to $\sim$, i.e.,

$$\text{for all } p_t \in \mathcal{T}, \ x, x^+ \in \mathcal{X}: \quad x \sim x^+ \implies \underset{y \in \mathcal{Y}}{\arg\max}\, p_t(y|x) = \underset{y \in \mathcal{Y}}{\arg\max}\, p_t(y|x^+). \quad (1)$$

As an illustration, consider the 3 classes (triangle, square, and circle) shown in Fig. 1. Then $\mathcal{T}$ consists of all tasks that are predictable from those shapes, e.g., recognizing shapes with vertices (blue/orange in Fig. 1) or recognizing the shape (yellow/red/purple in Fig. 1). Importantly, equivalence classes (here shapes) are different from—and essentially refinements of—downstream classes $\mathcal{Y}$ (here colors). Note that equivalence relations can model arbitrary transformations including cropping and adding Gaussian noise, which contrasts with typical restrictions to augmentations defined by group actions[9–11].

### 2.2 Idealized representations for ISSL

In this section, we define the optimal encoders, those that induce idealized representations that ISSL should be striving for. Although such encoders exist, they will likely not be learned in practice. Those idealized representations will nevertheless allow us to derive practical algorithms in Sec. 4. Note that our approach to defining ideal representations is to take into account how they will be used downstream, and thus depends on the desired family of probe $\mathcal{F}$ and potential invariant tasks $\mathcal{T}$.

The goal of ISSL is to learn representations from which (typically simple) probes $f \in \mathcal{F}$ classify well downstream tasks $p_t \in \mathcal{T}$, i.e., they achieve low 0-1 risk $\mathrm{R}_t(\phi, f) := \mathbb{E}_{p_t(X,Y)}[\mathbb{1}[Y \neq f(\phi(X))]]$. Ideally, for any task of interest $p_t \in \mathcal{T}$ there will be a probe that can classify it perfectly.

**Definition 2.** An encoder $\phi$ is *population optimal* for $\mathcal{T}, \mathcal{F}$, denoted as $\phi \in \Phi_{\text{pop}}$, iff predictors realize the Bayes error on any invariant task, i.e., for all $p_t \in \mathcal{T}$ we have $\inf_{f \in \mathcal{F}} \mathrm{R}_t(\phi, f) = 0$.

Population optimality ensures the existence of perfect downstream probes, which would be learned with infinite downstream data. In practice, however, predictors will be trained from finite datasets $\mathcal{D}_t$ of possibly small size $n$ with empirical risk minimization. When $n$ is small a fitted probe (ERM) $\hat{f} \in \widehat{\mathcal{F}}(\mathcal{D}_t, \phi) := \arg\min_{f \in \mathcal{F}} |\mathcal{D}_t|^{-1} \sum_{x,y \in \mathcal{D}_t} \mathbb{1}[y \neq f(\phi(x))]$ could be a terrible population predictor even when the underlying encoder is population optimal. Ideally, representations would thus also guarantee that *any* ERM performs as well as possible for *any* desired task and dataset size $n$. This suggests minimizing the following worst-case expected risk over tasks and ERMs.

$$\mathrm{W}_n(\phi, \mathcal{F}, \mathcal{T}) := \sup_{t \in \mathcal{T}} \mathbb{E}_{\mathcal{D}_t \overset{\text{iid}}{\approx} p_t^n(X,Y)} \left[ \sup_{\hat{f} \in \widehat{\mathcal{F}}(\mathcal{D}_t, \phi)} \mathrm{R}_t(\phi, \hat{f}) \right]. \quad (2)$$

**Definition 3.** An encoder $\phi^*$ is *sample optimal* for $\mathcal{T}, \mathcal{F}$ iff if it is population optimal and minimizes the worst-case expected risk of ERMs for arbitrary sample sizes, i.e.,

$$\text{for all } n \geq 1: \quad \phi^* \in \underset{\phi \in \Phi_{\text{pop}}}{\arg\min}\, \mathrm{W}_n(\phi, \mathcal{F}, \mathcal{T}). \quad (3)$$

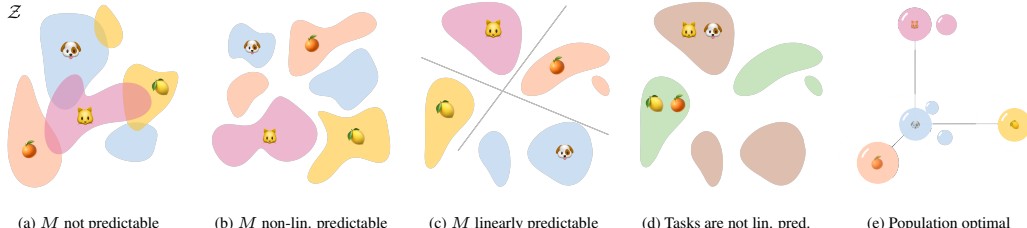

| (a) $M$ not predictable | (b) $M$ non-lin. predictable | (c) $M$ linearly predictable | (d) Tasks are not lin. pred. | (e) Population optimal |

Figure 2: Representations are *population optimal* for linear $\mathcal{F}$ iff their dimensionality is sufficiently large and equivalence classes $M(X)$ are linearly classifiable. The first 2 figures show representations from which $M(X)$ is (a) never or (b) only non-linearly classifiable. Although (c) ensures linear classification of $M(X)$, there exist invariant tasks, e.g. (d), that are not classifiable linearly. (e) Linear classifiability of $M(X)$ ensures population optimality iff the dimensionality is at least $|\mathcal{X}/\sim| - 1$.

## 3 Theoretical framework for linear probes $\mathcal{F}$

### 3.1 Characterizing optimal encoders for linear ISSL

In this section, we characterize all sample-optimal encoders (Def. 3), with simple properties that give insights into the form of the idealized representation. The key for our theory is that any $\sim$-invariant function $g$ can be written as a composition between some function $c_g$ and a *maximal invariant* $M$ [12], i.e., $g = c_g \circ M$ where $M : \mathcal{X} \rightarrow \{1, \ldots, |\mathcal{X}/\sim|\}$ indexes the equivalence class:

$$\text{for any } x, x^+ \in \mathcal{X}: \quad x \sim x^+ \iff M(x) = M(x^+). \tag{4}$$

To build intuition for the final characterization, let us first discuss population optimal encoders (Def. 2) for unconstrained probes, then linear probes, and finally sample optimality (Def. 3).

**Unconstrained probes.** By Def. 1, labels of downstream tasks $p_t \in \mathcal{T}$ are $\sim$-invariant. Labels can thus be written as some function $c_t$ of $M$, i.e., $\arg\max_y p_t(y|X) = c_t(M(X))$. This shows that $M(X)$ contains all and only information about desired tasks $\mathcal{T}$. If probes are unconstrained, an encoder is population optimal iff $M(X)$ is predictable/classifiable, i.e., there exist an $h_M$ such that $M(X) = h_M(\phi(X))$. Indeed, this ensures that the probe defined by $c_t \circ h_M$ can classify the task $p_t$.

**Linear probes $\mathcal{F}$.** The problem with constrained probes is that they might not be able to use the information about desired tasks. In particular, the previous probe might not be linear $c_t \circ h_M \notin \mathcal{F}$. As an illustration, consider 4 equivalence classes: cats, dogs, oranges, and lemons. Fig. 2b shows a representation from which $M(X)$ is predictable but invariant tasks are not linearly classifiable. In fact, even when $M(X)$ is linearly predictable, i.e., $h_M \in \mathcal{F}$ as in Fig. 2c, downstream labels might not be, i.e., $c_t \circ h_M \notin \mathcal{F}$ as shown in Fig. 2d. By standard VC dimension arguments [13, 14], such binary task is linearly predictable if the representation's dimensionality $d$ is one less than the number of equivalence classes $|\mathcal{X}/\sim| - 1$, as in Fig. 2e. Building on this intuition we prove that population optimal encoders are essentially[1] those that induce $d \geq |\mathcal{X}/\sim| - 1$ dimensional representations from which $M(X)$ is linearly predictable.

**Sample optimality.** Although population optimality ensures the existence of a perfect linear probe, ERM probes trained on finite samples might still be bad due to generalization issues (Fig. 3 left). Intuitively, one can remove such bad ERMs by mapping equivalent examples to the same representation, i.e., by using invariant encoders. Indeed, this ensures that ERMs that correctly predict one example in an equivalence class also correctly predict all the other ( Fig. 3 right). We prove that such invariance of population optimal encoders is necessary and sufficient for sample optimality.

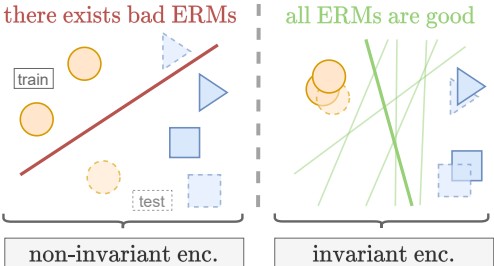

Figure 3: Invariance of population-optimal encoders is (a) necessary and (b) sufficient to ensure that there exists no bad ERM.

---

[1]The difference with learning theory is that instead of (binary) shatterability of all examples, we want $k$-ary shatterability of all equivalence classes from representations. The key is that both notions coincide for specific probes (e.g. linear) and invariant encoders, which are necessary for sample optimality.

Putting all together gives the following necessary and sufficient properties of sample-optimal encoders.

**Theorem 1.** An encoder $\phi^*$ is sample optimal for $\sim$-invariant tasks $\mathcal{T}$ and linear $\mathcal{F}$ if and only if

- $\mathcal{F}$-**predictability of** $M$: there exists a max. invariant $M$ and an $f \in \mathcal{F}$ s.t. $M(X) = f(\phi^*(X))$;
- **Invariance**: $\phi^*$ is $\sim$-invariant, i.e., for any $x, x^+ \in \mathcal{X}$ we have $x \sim x^+ \implies \phi^*(x) = \phi^*(x^+)$;
- **Dimensionality**: the effective dimensionality of the representations is at least one less than the number of equivalence classes, i.e., $\dim(\text{span}(\{\phi^*(x)|x \in \mathcal{X}\})) \geq |\mathcal{X}/\sim| - 1$.

### 3.2 The impact of augmentations on downstream performance

Let us compute the worst-case excess risk $W_n(\phi, \mathcal{F}, \mathcal{T})$ of sample optimal encoders and show its dependence on the invariance structure. The key is that since sample optimal encoders are invariant (Theorem 1), ERMs only need to be trained on a single example per equivalence class to perfectly classify all other examples from that class. The risk then depends on the proportion of equivalence classes seen when training the probe, which can be computed in closed form as a function of the number of equivalence classes $|\mathcal{X}/\sim|$ and downstream samples $n$. For other similar results refer to Appx. B.3.

**Proposition 2.** The worst expected risk of a sample optimal $\phi^*$ for $\sim$-invariant tasks $\mathcal{T}$ and any $\mathcal{F}$ is

$$W_n(\phi^*, \mathcal{F}, \mathcal{T}) = \left(1 - \frac{1}{|\mathcal{X}/\sim|}\right)^n \tag{5}$$

Prop. 2 shows that fewer equivalence classes, i.e., coarser $\sim$, leads to better downstream sample efficiency. Of course, the convergence rate will be slower for practical encoders than for sample optimal ones. The result nevertheless suggests that good augmentations should be as strong as possible (induce coarser $\sim$) while being label-preserving. Examples of coarse augmentations are those from CLIP [15], which map many images to similar sentences.

## 4 Practical ISSL objectives for linear probes $\mathcal{F}$

The main remaining question is how to practically enforce requirements from Theorem 1. In this section, we derive simple objectives that learn optimal encoders in *ideal settings* (infinite data, perfect optimizers, universal approximators). In the process, we shed light on why previously proposed ISSL methods work, and how to improve them in practice.

Our key insight is that we can learn sample-optimal encoders by jointly training an encoder and a logistic regression to predict $M(X)$ from the representations. Indeed, the resulting representations will be characterized by three properties, each of which implies a requirement from Theorem 1 ($\mathcal{F}$-predictability, invariance, effective dimensionality). First, the representations will allow linear predictability of $M(X)$ due to the linearity of logistic regression. Second, the variance of same-class representations will be minimized due to Jensen's inequality. Finally, the effective dimensionality will be maximal. Indeed, logistic regression favors maximal angles between representations of different classes to increase the confidence of the predicted class. Specifically, the representations learned by this joint procedure form a simplex equiangular tight frame (sETF)—as illustrated in Fig. 4 and discussed in the neural collapse literature [16–21]—and we prove that those are optimal.

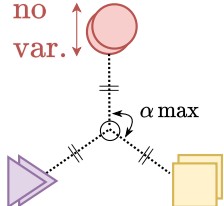

Figure 4: sETF are idealized representations that collapse same-class examples and maximize the angle across classes.

More formally, we prove in Prop. 3 that high dimension encoders trained to minimize the following multinomial logistic regression of $M(X)$, dubbed $\sim$-*ISSL log loss*, will be sample optimal,

$$\mathcal{L}_I(\phi; p_x) := \inf_{w \in \mathcal{W}_1} \mathbb{E}_{p_X}\left[-\log \frac{\exp\left(w(M(X))^\top \phi(X)\right)}{\sum_{m'=1}^{|\mathcal{X}/\sim|} \exp(w(m')^\top \phi(X))}\right], \tag{6}$$

where $w$ maps classes to weights, e.g., by indexing a weight matrix. Following previous work on neural collapse, we assume for this section that classes are equiprobable $p_x([x]) = 1/|\mathcal{X}/\sim|$, and that classifier's weights and representations are unit-normalized, i.e., $w \in \mathcal{W}_1 := \{1, \ldots, |\mathcal{X}/\sim|\} \to \mathcal{S}$ and $\phi \in \Phi_1 := \mathcal{X} \to \mathcal{S}$ where $\mathcal{S}$ denotes the $(d-1)$-sphere (these assumptions can be relaxed e.g. [22, 23]). We then have the desired relation between ISSL log loss and sample-optimal encoders.

**Proposition 3.** Let $p_X$ be a distribution with support $\mathcal{X}$ and equiprobable equivalence classes $p_X([x]) = 1/|\mathcal{X}/\sim|$, $\forall x \in \mathcal{X}$. If $d \geq |\mathcal{X}/\sim| - 1$ then any unit-normalized encoder that minimizes the $\sim$-ISSL log loss $\phi^* \in \arg\min_{\phi \in \Phi_1} \mathcal{L}_I(\phi; p_X)$ is sample-optimal for $\sim$-invariant tasks and linear $\mathcal{F}$.

Prop. 3 shows that the ISSL log loss is a perfect pretext task in ideal settings. This suggests optimizing the ISSL log loss in practice and provides a formal relation between self-supervised and classical supervised learning. The challenge is that we typically neither have access to the maximal invariant $M(X)$ nor the number of equivalence classes $|\mathcal{X}/\sim|$ required to compute the denominator of Eq. (6). Instead, knowledge about the equivalence structure comes from data augmentations $A(\tilde{X}|X)$ from which we can sample examples that are equivalent to the input $\tilde{X} \sim X$.

Having established our framework, we can re-interpret previous ISSL methods as practical approximations of the ISSL log loss using data augmentations (e.g., SimCLR, SwAV, DINO [24], SimSiam [25]). These approximations are nevertheless suboptimal: none of them learn sample- (nor population-) optimal encoders in idealized settings. By directly deriving ISSL methods from Eq. (6), we will prescribe improvements to previous ISSL objectives that ensure that sample-optimal encoders are learned in idealized settings. We broadly categorize prior methods into two families depending on whether they explicitly select the number of equivalence classes $|\mathcal{X}/\sim|$ or if it is implicitly inferred from augmentations. We call these approaches distillation and contrastive ISSL respectively. For derivations and Pytorch implementation see Appx. C.

### 4.1 Contrastive ISSL (CISSL)

Inspired by previous contrastive objectives [1, 26–28], we show how to optimize the ISSL log loss using data augmentations and negative samples to remove the need of knowing $M(X)$ and the number of classes $|\mathcal{X}/\sim|$. The resulting objective, dubbed *CISSL*, corresponds to SimCLR using only a projection head $g$ on one branch. See Fig. 5. This asymmetry is necessary to learn population optimal encoders for linear $\mathcal{F}$.

CISSL bypasses the need for $M(X)$ by noticing that it only appears in the ISSL log loss through the class weights $w(M(X))$. CISSL thus learns a function $g$ mapping equivalent (augmented) inputs $\tilde{X}$ to $w(M(\tilde{X}))$. Such mapping exists since $M$ is invariant, e.g., $g := w \circ M$.

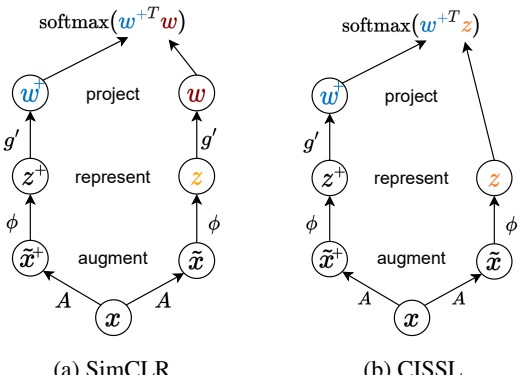

(a) SimCLR                    (b) CISSL

Figure 5: CISSL corresponds to SimCLR with a single (asymmetric) projection head $g'(z^+)^\top z$. This ensures linear predictability of downstream tasks, which will be computed by $W_t^\top z$.

When augmentations satisfy the Markov Chain $\tilde{X} - M(X) - X$, we show that we can replace the class weights $w(M(X))$ in the ISSL loss by $g(\tilde{X})$, where $g$ is optimized over. Intuitively, this is because predicting $\tilde{X}$ contains only information about $M(X)$ due to the data processing inequality.

The only remaining challenge is computing the denominator of Eq. (6) without summing over the unknown classes $\mathcal{X}/\sim$. To do so we use ranking conditional noise contrastive estimation (NCE; [29–31]). NCE replaces classification of equivalence classes by classification of positives in a batch $\tilde{\mathbf{X}} := \{\tilde{X}^+, \tilde{X}_1^-, \ldots, \tilde{X}_k^-\}$ where the positive $\tilde{X}^+$ is sampled from the conditional $A(\tilde{X} \mid X)$, while the $k$ negatives $\tilde{X}_i^-$ come from the marginal $A(\tilde{X}) = \mathbb{E}_{p_X}[A(\tilde{X}|X)]$. Using Monte Carlo (MC) estimates with an unlabeled dataset $\mathcal{D} \overset{\text{i.i.d.}}{\sim} p_X$ we get our final empirical CISSL objective

$$\hat{\mathcal{L}}_C(\phi; \mathcal{D}) := \inf_{g \in \mathcal{S}^{\mathcal{X}}} \sum_{x \in \mathcal{D}} \mathbb{E}_{p(\tilde{\mathbf{X}}|x,\mathcal{D})} \left[ -\log \frac{\exp g(\tilde{X}^+)^\top \phi(x)}{\sum_{\tilde{X}' \in \tilde{\mathbf{x}}} \exp g(\tilde{X}')^\top \phi(x)} \right]. \tag{7}$$

By Prop. 3 and NCE's consistency [31], encoders trained with CISSL $\phi^* \in \arg\min_{\phi \in \Phi_1} \hat{\mathcal{L}}_C(\phi; \mathcal{D})$ are sample optimal for linear $\mathcal{F}$ in our ideal setting assumption ($|\mathcal{D}| \to \infty$ and unconstrained $g$) and when $\tilde{X} - M(X) - X$ forms a Markov Chain. While consistency (and thus optimality) holds for $k \geq 1$, more negatives $k$ improves statistical efficiency [31].

Typically $\tilde{X}, X$ take value in the same space (e.g. images). If so, we can tie parameters by encoding $\tilde{X}$, i.e., we can replace $g$ by $g' \circ \phi$ where $g' : \mathbb{R}^d \to \mathcal{S}$ is called a projection head. The logits

inside of Eq. (7) are then computed by $g'(\phi(\tilde{X}))^\top \phi(x)$ as shown in Fig. 5b. This is very similar to SimCLR's objective $g'(\phi(\tilde{X}))^\top g'(\phi(x))$ shown in Fig. 5a. The difference is that, by projecting the current representation, *SimCLR learns encoders that are not even population optimal for linear $\mathcal{F}$.* In contrast, CISSL learns sample optimal encoders in ideal settings. Intuitively, this is because CISSL trains the representations in the same way as they will be used in downstream tasks $p_t$. Indeed, representations $\phi(X)$ will be dotted with the downstream tasks' weights $W_t^\top \phi(x)$ to compute logits. In ISSL, representations $\phi(x)$—rather than their projections $g'(\phi(x))$—should thus be used in the inner product with ISSL weights $w(M(X)) = g'(\phi(\tilde{X}))$ to compute logits. CISSL thus derives, from first principles, an asymmetric use of projection heads.

### 4.2 Distillation ISSL (DISSL)

In practice, CISSL requires contrasting many negatives. To avoid contrastive estimation, we can directly approximate $M(X)$—instead of the weight $w(M(X))$—so that the denominator of Eq. (6) can be computed exactly. The resulting method, dubbed *DISSL* (Fig. 6), is a simpler and theoretically-grounded version of non-contrastive losses (e.g. SwAV, DINO, SimSiam). The challenge and main difference between each of those methods is how they estimate $M(X)$.

As $M(X)$ is discrete, we use a conditional categorical distribution $q(\hat{M} \mid X)$ to estimate $M(X)$ with a random variable $\hat{M}$. Replacing terms in the ISSL log loss then gives the following objective. Differences with Eq. (6) are in red.

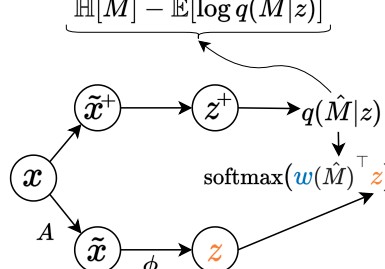

Figure 6: DISSL. The teacher (top branch) is trained with the top loss to ensure $\hat{M}$ is a maximal invariant r.v. The student (bottom branch) distills the teacher by predicting $\hat{M}$.

$$\inf_{w \in \mathcal{W}_1} \mathbb{E}_{p_X q(\hat{M} \mid X)}\Big[-\log s_{\phi,w}(\hat{M} \mid X)\Big], \quad s_{\phi,w}(m \mid x) = \frac{\exp\big(w(m)^\top \phi(x)\big)}{\sum_{m=1}^{C} \exp(w(m)^\top \phi(x))}, \quad (8)$$

To highlight similarities with previous methods [2, 24, 32] we refer to Eq. (8) as *distilling* the *teacher* $q(\hat{M} \mid X)$ into the student $s_{\phi,w}(\hat{M} \mid X)$. Importantly, Eq. (8) is exactly the ISSL log loss if the teacher outputs a maximal invariant random variable, i.e., $\hat{M} = M(X)$. By Prop. 3, distillation thus learns sample-optimal encoders when the teacher takes at least $C \geq |\mathcal{X}/\sim|$ values and satisfies the following requirements, which correspond to the definition of maximal invariance (Eq. (4)).

**Deterministic** the teacher is a deterministic distribution $\max_{m \in \{1,...,C\}} q(m \mid X) = 1$;
**Invariant** the teacher maps positives together $x \sim x^+ \implies q(\hat{M} \mid x) = q(\hat{M} \mid x)$;
**Maximal** the teacher maps negatives separately $x \not\sim x^- \implies q(\hat{M} \mid x) \neq q(\hat{M} \mid x^-)$.

Intuitively, these requirements ensure that the teacher clusters examples by equivalence classes, which will then be classified by the student. There are many ways of enforcing such requirements. In the following, we use information-theoretic quantities (entropies and KL divergences) to jointly train the teacher (online clustering) and student. Such quantities have the advantage of being interpretable and computable in closed form for the categorical distributions given by our teacher and student.

**Determinism and Invariance.** Both properties hold if and only if for any equivalent example $x \sim x^+$ the cross-entropy of the teacher's outputs is minimized $\mathbb{E}_{q(\hat{M} \mid x)}[-\log q(\hat{M} \mid x^+)] = 0$. Indeed, zero cross-entropy simultaneously minimizes the conditional entropy $\mathbb{H}[\hat{M} \mid x] = 0$ (equivalent to determinism) and the KL divergence $\mathbb{D}_{\mathrm{KL}}[q(\hat{M} \mid x)\|q(\hat{M} \mid x^+)] = 0$ (equivalent to invariance).

**Maximality.** A natural way of enforcing maximality is to train the teacher to predict differently negatives $x \not\sim x^-$. This requires accessing batches of negatives, which we wanted to avoid with DISSL. Instead, assume that we know (or have a prior on) the true distribution of equivalence classes $p(M(X))$. Then a deterministic and invariant teacher is maximal if and only if the KL divergence between its marginal $q(\hat{M}) = \mathbb{E}_{p_X}[q(\hat{M}|X)]$ and the true one is minimized $\mathbb{D}_{\mathrm{KL}}[q(\hat{M})\|p(M(X))] = 0$. We can thus avoid contrasting negatives using (a prior on) the distribution of equivalence classes.

Using a Lagrangian relaxation (with multipliers $\lambda, \beta$) of the teacher's constraints with an MC estimate of the distillation loss (Eq. (8)) we get the following empirical DISSL loss $\hat{\mathcal{L}}_D(\phi; \mathcal{D}) :=$

$$\inf_{q,w} \lambda \underbrace{\mathbb{D}_{\mathrm{KL}}\Big[q(\hat{M})\Big\|p(M(X))\Big]}_{\text{Maximality}} - \sum_{x \in \mathcal{D}} \mathbb{E}_{A(\tilde{X}|x)q(\hat{M} \mid x)}\Big[\underbrace{\beta \log q(\hat{M} \mid \tilde{X})}_{\text{Det. and Inv.}} + \underbrace{\log s_{\phi,w}(\hat{M} \mid x)}_{\text{Distillation}}\Big]. \quad (9)$$

In ideal settings ($|\mathcal{D}| \to \infty$, unconstrained $q$, known $p(M(X))$) and for $\lambda, \beta \to \infty$ we show that encoders trained with DISSL are sample optimal for linear $\mathcal{F}$ and tasks that are invariant to the equivalences defined by the connected components of the augmentation graph [8]. If the marginal is unknown, we follow the MaxEnt [33] and use a uniform prior $p(M(X)) = \mathrm{Unif}(1, C)$. The KL in Eq. (9) then corresponds to maximizing entropy $\mathbb{H}[\hat{M}]$.

As with CISSL, when $\tilde{X}, X$ are in the same space, we tie parameters by encoding $\tilde{X}$ before the teacher, i.e., $q(\hat{M} \mid X) = \mathrm{softmax}(g'(\phi(X)))$ for a head $g' : \mathbb{R}^d \to \mathbb{R}^C$. Algorithm 1 and Fig. 6 illustrate DISSL with a uniform prior, and a marginal $q(\hat{M})$ estimated from batches.

---

**Algorithm 1** Batched DISSL

**Require:** head $g$, weights $W$, enc. $\phi$, batched inputs $X$, aug. A, hyp. $\beta, \lambda$.
1: $\tilde{X}, \tilde{X}^+ \leftarrow \mathrm{sample}(\mathrm{A}(\tilde{X} \mid X), 2)$
2: $q(\hat{M} | \tilde{X}) = \mathrm{softmax}(g(\phi(\tilde{X}))$
3: $q(\hat{M} | \tilde{X}^+) = \mathrm{softmax}(g(\phi(\tilde{X}^+))$
4: $s(\hat{M} | \tilde{X}^+) = \mathrm{softmax}(W^T \phi(\tilde{X}^+))$
5: $q(M) = \mathrm{batch\_avg}(q(M | \tilde{X}))$
6: $\mathrm{mxml} = \sum_m q(m) \log q(m)$
7: $\mathrm{det\_inv} = \sum_m q(m|\tilde{X}) \log q(m|\tilde{X}^+)$
8: $\mathrm{dstl} = \sum_m q(m|\tilde{X}) \log s(m|\tilde{X}^+)$
9: **return** $\lambda * \mathrm{mxml} - \beta * \mathrm{det\_inv} - \mathrm{dstl}$

---

DISSL is one of the many distillation methods that can be derived from our teacher's requirements. These requirements generally lead to a framework for deriving, comparing, and analyzing distillation objectives. In Appx. D we provide a taxonomy of 12 previous SSL methods from this perspective—none of which recover sample optimal encoders. Typically, previous methods favor: (i) determinism by "sharpening" the teacher with a temperature parameter; (ii) invariance by making matching the teacher and student on equivalent inputs $q(\hat{M} \mid x^+) \approx s_{\phi,w}(\hat{M} \mid x)$; (iii) maximality through optimization tricks (e.g. stop gradients or momentum encoder) to avoid a constant teacher —referred to as "collapsing" [25, 34–36]. Note that avoiding constant collapse $q(\hat{M} \mid x') = q(\hat{M} \mid x)$ for any $x, x' \in \mathcal{X}$ is insufficient, e.g., mapping half the inputs to a constant is also undesirable. Maximality formalizes the desired requirement: non-equivalent examples should not be mapped together.

## 5 ISSL for non-linear predictors $\mathcal{F}^+$

Linear probes $\mathcal{F}$ are standard for evaluating representations [27, 37, 38]. However, if the goal is to maximize performance it is natural to consider non-linear predictors $\mathcal{F}^+$. The question then becomes how should we learn optimal representations for any $\mathcal{F}^+$? In Appx. B.5 we extend our framework to certain non-linear probes, such as neural networks, that separate into a hidden component $h$ and linear layer, i.e., $f(\cdot) = W^T h(\cdot)$. We informally summarize the theory and implications below.

**Theory.** There are two main differences between our characterization of optimal encoders for linear and non-linear probes. First, encoders should ensure predictability of $M(X)$ for the desired $\mathcal{F}^+$. Second, the dimensionality requirement decreases with the complexity of $\mathcal{F}^+$, e.g., for universal $\mathcal{F}^+$ it is $d \geq 1$. More generally, the necessary dimension and the sufficient dimension for optimality do not coincide and respectively depend on the predictors' VC dimension [13] and memory capacity [39].

**Implications.** Both theoretical differences lead to direct implications for non-linear ISSL. First, we can decrease the dimensionality when performing ISSL for more complex non-linear probes. Second, we should use a projection head that ensures the predictability of the maximal invariant with using the desired probes $\mathcal{F}^+$. Similarly to the linear case, we should match how the representation is used in the ISSL log loss and downstream tasks. We should thus apply the non-linear predictor $h$ in a similar asymmetric way. For CISSL we should thus compute the right hand side of Fig. 5 as $g(\phi(\tilde{X}))h(\phi(x))$. For DISSL, we should change the line 4 of Algorithm 1 to $s(\hat{M}|\tilde{X}^+) = \mathrm{softmax}(W^T h(\phi(\tilde{X}^+)))$.

## 6 Summary of insights and relation to previous work

Let summarize our framework's main insights and their relation to previous work. Details in Appx. D.

**Dimensionality.** Theorem 1 shows large representation's dimensionality is needed to ensure probes can classify all invariant tasks (note: this is different from the projection's dimensionality analyzed in [1, 40]). This suggests: (i) increasing the dimensionality of the representation; and (ii) ensuring that representations do not live in a lower dimensional subspace. Although the first has surprisingly not been investigated in SSL, there have been recent empirical analyses about the second [41, 42].

**Projection heads.** Secs. 4 and 5 theoretically show how to choose projection heads, namely, one should be as large as possible while the other should have the architecture of downstream probes.

To our knowledge, we are the first to relate the architecture of the probing family and the projection head. Furthermore, our theory suggests why projection heads empirically improve performances [1, 25, 43] in general SSL, i.e., beyond avoiding collapsing in non-contrastive learning [25, 36].

**Augmentations.** Prop. 2 shows the benefit of using coarse label-preserving augmentations, by proving the exact relation between optimal sample efficiency and the number of equivalence classes. This gives a new theoretical perspective on the use of augmentations that remove a lot of information of the input, which have been suggested to be useful for many different reasons [12, 44–48].

**DISSL.** In Sec. 4 we derive DISSL to learn optimal encoders in ideal settings, contrary to prior work. DISSL follows recent methods [49–51], in particular SwAV and DINO, that learn representations by jointly predicting and learning clusters of examples. DISSL/DINO/SWAV each distill a categorical teacher but differ in how they enforce its maximality. DINO does it implicitly through optimization tricks, e.g., exponentially moving averages and stop-gradients. SwAV explicitly enforces maximality by equiprobably clustering a queue of examples using Sinkhorn's algorithm [52]. DISSL is also explicit but uses a more efficient max entropy regularization (no queue/stop-gradient/internal algorithm). Furthermore, DISSL's student uses a linear projection head to ensure good linear probing (Sec. 4.1).

**Theoretical SSL.** Theorem 1 characterizes optimal encoders for ISSL. This contrasts with standard theoretical work in SSL that typically analyze specific SSL algorithms [3, 5, 6, 8, 53–55] and / or focus on upper-bounding downstream performance [4]. The advantage of our theory is that it provides a simple unifying framework from which we can derive a variety of SSL methods and actionable insights. The downside is that by distancing itself from specific algorithms, the framework does not provide guarantees outside of ideal settings. We thus see our theory as complementary to prior work.

## 7    Experiments

The experiments in the main paper focus on evaluating our frameworks' prescriptive implications. For more results see Appx. G. For experimental details see Appx. F and our GitHub repository.

In summary, our experimental results show that: (i) CISSL/DISSL outperform their respective baselines on standard benchmarks as suggested by Sec. 4; (ii) increasing dimensionality improves downstream probing as suggested by Theorem 1; (iii) coarser augmentations improve sample efficiency as in Prop. 2; (iv) smaller ISSL log loss improves downstream performance as suggested by Prop. 3; (v) projection heads should be related to probing family as discussed in Secs. 4 and 5.

For the first experiments, we use TinyImageNet [56], 300 pretraining epochs, and ResNet18s. Note that for ResNet18s the dimensionality of the representation (`res5avg`) is $d = 512$. For contrastive baselines we use SimCLR. For distilling baselines we use the best over DINO, SwAV, SimSiam. We use standard TinyImageNet augmentations [34] (color jittering, grayscaling, cropping) with a parameter controlling the probability and strength of augmentations to study the effect of coarsening.

**CISSL/DISSL improve linear probing**

Table 1 shows that each of our prescriptions significantly improve distillation and contrastive ISSL. "Ours" vs "Base" shows that removing one non-linear projection head (as in Fig. 5) gives a 1% gain in contrastive learning, while our new distillation objective achieves 2% gains compared to the best distillation baseline. "dim. ↑" shows that increasing the dimensionality of representation $512 \to 2048$ further improves linear probing by 2%. "Epochs ↑" and "coarse aug." show that training for longer $(300 \to 1000$ epochs ) and using coarser augmentations both fur-

Table 1: Our prescriptive implications improves ISSL for linear probes on TinyImageNet with ResNet18 encoders. 3 seeds.

|  | CONTR. | DISTIL. |
| --- | --- | --- |
| BASE. (SIMCLR ǀ DINO) | $44.9_{\pm 0.2}$ | $43.5_{\pm 0.2}$ |
| **OURS** (CISSL ǀ DISSL) | $45.8_{\pm 0.0}$ | $45.5_{\pm 0.1}$ |
| + DIM. ↑ | $47.6_{\pm 0.1}$ | $47.9_{\pm 0.1}$ |
| + EPOCHS ↑ | $48.7_{\pm 0.1}$ | $49.2_{\pm 0.1}$ |
| + COARSE AUG. | $51.0_{\pm 0.1}$ | $50.6_{\pm 0.2}$ |

ther improve performance by 1–2%. Altogether our framework's insights significantly improve linear probing accuracy: 6% gains for contrastive learning and 7% for distillation.

**Coarse augmentations improve sample efficiency.** Prop. 2 suggests that coarser label-preserving augmentations improve downstream sample efficiency. We test that by training four DISSL encoders with augmentations of varying strengths ($[25\%, 50\%, 100\%, 200\%]$ relative to standard augmentations) and evaluating them at different downstream samples sizes. As suggested by the theory, Fig. 7a shows that stronger label-preserving augmentations improve sample efficiency.

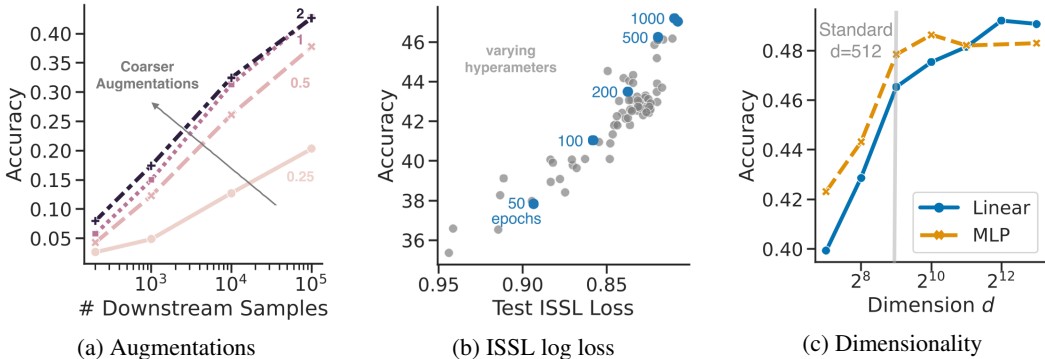

| (a) Augmentations | (b) ISSL log loss | (c) Dimensionality |

Figure 7: As suggested by our theory: (a) coarsening augmentations improves sample efficiency; (b) decreasing ISSL log loss improves representations, which can be achieved by longer training; and (c) increasing the representation's dimensionality improves performance for linear $\mathcal{F}$ but less so for MLPs $\mathcal{F}^+$. Y-axis is TinyImageNet probing performance. Each point is obtained by sweeping: (a) augmentation strengths for DISSL and downstream sample sizes; (b) CISSL's optimization hyperparameters including epochs; and (c) the dimensionality of CISSL's representations.

**ISSL log loss correlates with performance.** Previous work [57] suggested contrastive losses may not predict probing performance. On the contrary, Prop. 3 shows that minimal ISSL log loss implies the optimality of encoders if $d$ is large. This suggests that, for fixed augmentations, ISSL log loss is highly related to performance. To test this relation, we trained 80 CISSL models with various hyperparameters, while fixing augmentations and negatives $k$. Fig. 7b shows that the test ISSL loss indeed correlates with probing accuracy (gray points). Appx. G.4 shows similar results for the train ISSL loss.

**Longer training decreases ISSL log loss.** An efficient way of decreasing ISSL log loss is longer training (blue points in Fig. 7b only vary the number ISSL epochs). Prop. 3 thus provides an explanation of the well-known performance gains of longer ISSL training [1, 58]. Namely, longer training decreases ISSL log loss, which results in representations that are closer to optimality.

**Increasing dimensionality improves performance.** Theorem 1 shows that linear ISSL requires a dimensionality of $d = |\mathcal{X}/\sim| - 1$ to ensure predictability of all invariant tasks. In practice, the number of equivalence classes is most likely very large. Although such dimensionality is impractical, it does suggest increasing the dimensionality beyond the $d = 512$ of ResNet18s. To test this, we train CISSL with a larger dimensionality by increasing the output channels before the average pooling layer. To keep the number of parameters comparable, we use bottlenecks or low-rank linear layers before and after the representations layer as detailed in Appx. F. Fig. 7c shows that increasing the dimensionality has a significant effect on downstream performance for linear $\mathcal{F}$ (blue).

Previous results concern linear $\mathcal{F}$ which are standard for ISSL evaluation. In practice, one would likely use more complex probes $\mathcal{F}^+$ to improve results. In Sec. 5 we discussed how to perform ISSL for $\mathcal{F}^+$. In the following, we evaluate ISSL for MLPs $\mathcal{F}^+$ with two hidden layers of 2048 units.

**Performing ISSL for MLPs $\mathcal{F}^+$.** Table 2 shows that MLP probes ("$\mathcal{F}_{\text{EVAL}}$: MLP") outperforms linear ones ("$\mathcal{F}_{\text{EVAL}}$: lin.") even in few-shot regimes (10 shot). Furthermore, when predicting with MLP probes it is desirable to train the ISSL encoders for MLPs ("$\mathcal{F}_{\text{ISSL}}$: MLP") instead of linear probes (row 4 vs 3). As discussed in Sec. 5, the only difference between DISSL for different probes is that the student's representation is projected using a head with the same architecture as downstream probes.

Table 2: MLP probes outperform linear ones on TinyImageNet when using MLP ISSL.

|      | $\mathcal{F}_{\text{ISSL}}$ | $\mathcal{F}_{\text{EVAL}}$ | ALL | 10 SHOT |
|------|------|------|------|------|
| DINO | - | MLP | $44.7_{\pm 0.3}$ | $22.0_{\pm 0.2}$ |
| DISSL | LIN. | LIN. | $45.9_{\pm 0.0}$ | $25.3_{\pm 0.0}$ |
| DISSL | LIN. | MLP | $46.4_{\pm 0.1}$ | $25.6_{\pm 0.0}$ |
| DISSL | MLP | MLP | $47.5_{\pm 0.2}$ | $26.1_{\pm 0.1}$ |

**Dimensionality affects less MLPs $\mathcal{F}^+$.** In Sec. 5 we saw that the optimal dimensionality $d$ decreases for more complex predictors $\mathcal{F}^+$. Fig. 7c indeed shows that larger dimensions result in smaller gains when the training and evaluation probe are MLPs $\mathcal{F}^+$ (orange) compared to linear ones $\mathcal{F}$ (blue).

Given encouraging TinyImageNet results, we investigated whether our objectives can be used as a replacement for standard objectives on ImageNet [59]. We specifically replaced SimCLR with CISSL and SwAV with DISSL in VISSL's codebase [60] without modifying hyperparameters.

Table 3: Our models outperform baselines on ImageNet. All models use ResNet50, 100 epochs, 2560 batch size. For our models, we increase the dimensionality of the representations ($2048 \rightarrow 8192$).

| SimCLR | SwAV | Barlow T. | CISSL | DISSL |
|---|---|---|---|---|
| 65.1 | 64.6 | 66.1 | 67.7 | **68.9** |

**Our models outperform baselines on ImageNet.** Table 3 shows that both CISSL and our novel DISSL objective significantly outperform all considered baselines, including SwAV by a $4\%$, when using standard augmentations. For CISSL we found (contrary to Table 1) that gains compared to SimCLR were mostly due to increasing the dimensionality of the representation.

Given encouraging results on standard augmentations, we compared the performance of DISSL with a near SOTA model: SwAV trained with their special multi-crop augmentations.

**DISSL is competitive with SOTA models at scale.** Table 4 shows that DISSL outperforms SwAV. Combined with Table 3 this suggests that DISSL works well in different settings. This is encouraging given the lack of tuning and the simplicity of DISSL's out-of-the-box objective. In contrast, SwAV requires stopping gradients, storing a queue, running Sinkhorn's algorithm, and freezing certain parameters during initial training steps.

Table 4: Our DISSL outperforms SwAV using $2 \times 160 + 4 \times 96$ multi-crops on ImageNet. 2560 batch size, ResNet50.

| | 100 epochs | 400 epochs |
|---|---|---|
| SwAV | 69.5 | 73.5 |
| **DISSL** | **70.7** | **74.0** |

Table 5: DISSL is competitive on transfer tasks. Same models as in Table 4 evaluated by linear probes.

| | Food | CIFAR10 | CIFAR100 | Cars | Aircrafts | DTD | Pets | Caltech | Flowers |
|---|---|---|---|---|---|---|---|---|---|
| SwAV | 75.5 | 92.0 | 76.2 | 58.2 | **49.1** | 72.6 | 86.9 | **92.0** | 94.7 |
| **DISSL** | **77.9** | **93.6** | **77.6** | **62.2** | 48.1 | **73.9** | **88.0** | 91.5 | **95.3** |

**DISSL is competitive on transfer.** Table 5 shows that DISSL generally outperforms SwAV on the standard transfer benchmarks from [61] even though our theory does not discuss transfer.

# 8 Summary and Outlook

We presented a simple conceptual framework to understand, compare and improve ISSL methods. Inspired by recent work on optimal representations for supervised [62] and robust [63] learning, we derived such a framework by studying algorithmic-agnostic goals for ISSL. In particular, Theorem 1 provides the minimal and sufficient requirements for optimal encoders. On the algorithmic side, Sec. 4 uses our framework to derive simpler and theoretically-motivated variants of prior SSL objectives. Altogether our framework provides actionable insights for ISSL such as how to choose: SSL algorithms, the dimensionality of the representations, projection heads, and augmentations. On the empirical side, Sec. 7 shows that our prescriptions leads to significant gains on linear probing benchmarks.

There are many limitations that should be addressed for a more realistic prescriptive framework for ISSL. See Appx. E for more details. First, our theory is binary, e.g., augmentations are either label preserving or not, and encoders are optimal or not. Although our framework likely captures the right intuition, nuance would be more realistic. Second, we consider unconstrained encoders $\phi$, dimensionality, and infinite unlabeled data. As a result, our framework cannot currently improve the computational or data efficiency of the pretraining phase. Finally, we consider all invariant tasks $\mathcal{T}$ even though only a subset of those are meaningful. Despite those limitations, we showed that our framework's insights lead to substantial improvements in ISSL performance.

## Acknowledgments and Disclosure of Funding

We thank Ananya Kumar, Shengjia Zhao, Mo Tiwari, Michael Xie, Niladri Chatterji, Shibani Santurkar, Jiaming Song, Ali Malik, Colin Wei, Yangjun Ruan and Chris Maddison for helpful feedback. YD is supported by a Knights-Hennessy Scholarship. The work is partially supported by an Open Philanthropy Project Award, Sloan Fellowship, and ARO (W911NF-21-1-0125).

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
