# Appendix

## Table of Contents

# A Preliminaries

## A.1 Notation

**Probability.** Random variables (r.v.s) are denoted with an upper-case letter $X$, their realizations by the associated lower case $x$, and their sample space by calligraphic letters $\mathcal{X}$. We will say that $X$ takes value in (t.v.i) $\mathcal{X}$. We denote the probability density or mass function (which exists by Assmp. 1) of $X$ as $p_X$, its evaluation at $x$ as $p_X(x)$ or simply $p(x)$ if unambiguous, and the set of all such densities as $\mathcal{P}(\mathcal{X})$. When it is necessary to be explicit, we will denote "$X$ is distributed as $p_X$" by $X \overset{\mathrm{d}}{\sim} p_X$. Expectations are written as $\mathbb{E}_{p_X}[X]$, independence of two random variables by $X \perp\!\!\!\perp Y$ and conditional independence as $X \perp\!\!\!\perp Y \mid Z$ or by the Markov Chain $X - Z - Y$. We will use $Z \overset{\text{a.s.}}{=} X$ to denote almost sure equality of two r.v.s $Z, X$.

**Equivalence.** $x \sim x'$ denotes that $x$ and $x'$ are equivalent with respect to (w.r.t.) an equivalence relation $\mathcal{X}$ (the exact relation being implicit). The equivalence class of $x$ under $\sim$ consists of all elements that are equivalent to $x$ and is denoted $[x]_\sim$ (or $[x]$ if clear from the context), i.e. $[x]_\sim := \{x' \in \mathcal{X} \mid x' \sim x\}$. The set of all equivalence classes, i.e., the quotient set, will be denoted as $\mathcal{X}/\sim := \{[x] \mid x \in \mathcal{X}\}$. We will use $\sim_A \subseteq \sim$ to denote that $\sim_A$ is a refinement of (or finer than) $\sim$, i.e., $x \sim_A x^+ \implies x \sim x^+$. If additionally $x \sim_A x^+ \not\Longleftarrow x \sim x^+$ we say that $\sim_A$ is a strict refinement of $\sim$. If $\sim_A$ is finer than $\sim$ we also say that $\sim$ is coarser than $\sim_A$. Equivalence relations are natural way of defining invariance, specifically, we say that a function $f$ is $(\mathcal{X}, \sim)$-invariant (or $\sim$-invariant for conciseness) iff it is constant in an equivalence class $x \sim x^+ \implies f(x) = f(x^+)$. If a function $f : \mathcal{X} \to \mathcal{S}$ is $(\mathcal{X}, \sim)$-invariant we overload notation and use $f : \mathcal{X}/\sim \to \mathcal{S}$ to also denote the function whose domain is now the quotient set and such that for any $x \in \mathcal{X}$ we have $f([x]) = f(x)$.

**Predictive tasks.** Letters $X$, $Z$, and $Y$ refer to the input, representation and target of a predictive task, respectively. Representations are given by an encoder $\phi : \mathcal{X} \to \mathcal{Z}$. For binary classification, we predict the target using predictors $f : \mathcal{Z} \to \mathbb{R}$ in a specified family of probes $f \in \mathcal{F}_2 \subseteq \{f : \mathcal{Z} \to \mathbb{R}\}$, that map representations to a scalar logit. For $k$-class classification we instead predict the target using predictors $f : \mathcal{Z} \to \mathbb{R}^k$ in a specified family of probes $f \in \mathcal{F}_k \subseteq \{f : \mathcal{Z} \to \mathbb{R}^k\}$, that map representations to a vector logit. The predicted labels are then extracted by a function $\mathrm{pred} : \mathbb{R}^\Delta \to \mathcal{Y}$, which unifies binary and multi-class classification settings. Specifically, for multi-class we have $\Delta > 2$ and the predicted class is $\mathrm{pred}(logit) = \arg\max_i logit[i]$. For binary tasks we have $\Delta = 1$ and the predicted class is given by the sign of the logit $\mathrm{pred}(logit) = \mathbb{1}[logit \geq 0]$. Throughout the paper we work with tasks with a different number of labels $|\mathcal{Y}|$, and predictors with different codomain $\mathcal{F} = \bigcup_k \mathcal{F}_k$. For conciseness we then use $\inf_{f \in \mathcal{F}}$ to mean the infimum over predictors with the right codomain $\inf_{f \in \mathcal{F}_k \subseteq \mathcal{F}}$. We will be considering the 0-1 loss $\mathbb{1}[y \neq \mathrm{pred}(logit)]$, which we simply denote as $\ell(y, logit)$ for conciseness. We use "probes", "predictors", and "classifiers" as synonyms. Similarly we call $\mathcal{F}$ a "probing" or "predictive" family, or simply refer to it as "probes" or "predictors".

**Population risk minimization.** For any task $p_t(X, Y)$, we would like a predictor that achieves a risk (expected 0-1 loss) $\mathrm{R}_t(\phi, f) := \mathbb{E}_{p_t(X,Y)}[\ell(Y, f(\phi(X)))]$ close to the Bayes risk (label noise) $\mathrm{R}_t^* := \mathbb{E}_{p_t(X)}[1 - \max_{y \in \mathcal{Y}} p_t(y|X)]$. In particular, we evaluate the encoder on each task using the minimum risk over predictors in $\mathcal{F}$ (with the right codomain) denoted as $\mathrm{R}_t(\phi, \mathcal{F}) := \inf_{f \in \mathcal{F}} \mathrm{R}_t(\phi, f)$. Risk minimizers are denoted as $\mathcal{F}(t, \phi) := \arg\min_{f \in \mathcal{F}} \mathrm{R}_t(\phi, f)$ (they exist due to the boundedness of 0-1 loss and Assmp. 1). For conciseness and to emphasize that the most likely label is assumed unique (Def. 1), we denote the most likely label by $c_t(x) = \arg\max_{y \in \mathcal{Y}_t} p_t(y|x)$.

**Datasets, empirical distributions, and empirical risk minimization.** In practice we train the predictors by optimizing the empirical risk on a given dataset. Let $\mathcal{D}_t := \{(x_i, y_i)\}_i$ be a dataset of $n$ examples from $\mathrm{supp}(p_t(X, Y))$. Then we denote the empirical distribution induced by the dataset as $\hat{p}_{\mathcal{D}_t}(Y, X) := \frac{1}{|\mathcal{D}_t|} \sum_{(x_i, y_i) \in \mathcal{D}_t} \mathbb{1}[X = x_i] * \mathbb{1}[Y = y_i]$. The empirical risk minimizers on that dataset is denoted as $\widehat{\mathcal{F}}(\mathcal{D}_t, \phi) := \arg\min_{f \in \mathcal{F}} \frac{1}{|\mathcal{D}_t|} \sum_{(y_i, x_i) \in \mathcal{D}} \ell(y_i, f(\phi(x_i)))$, and the risk they achieve is denoted as $\widehat{\mathrm{R}}_{\mathcal{D}_t}(\phi, \mathcal{F})$. We will also denote the set of unlabeled examples seen during training as $\mathcal{X}_{\mathcal{D}_t} := \{x \mid (x, y) \in \mathcal{D}_t\}$ and the set of equivalence classes seen during training as $\mathcal{X}_{\mathcal{D}_t}/\sim := \{[x] \mid x \in \mathcal{X}_{\mathcal{D}_t}\}$.

**Augmentations.** We refer to augmentations as a conditional distribution $A(\tilde{X}|X)$ from inputs $X$ to augmented inputs or view $\tilde{X}$ that t.v.i. $\tilde{\mathcal{X}}$. The sample space of augmented inputs $\tilde{\mathcal{X}}$ is typically the same as the input space $\mathcal{X}$, e.g., in standard image augmentations. But they can be different, for example in CLIP [15] we view the text sentences as being augmented views of the images that t.v.i. in a different sample space.

**Information theory.** We denote the Kullback–Leibler (KL) divergence between two distributions $p_X$ and $q_X$ as $\mathbb{D}_{\text{KL}}[p_X \| q_X] = \mathbb{E}_{p_X}[\log \frac{p(X)}{q(X)}]$, the cross entropy as $\mathbb{H}[p_X, q_X] := \mathbb{E}_{p_X}[-\log q(X)]$, the entropy of $X$ as $\mathbb{H}[X] := \mathbb{H}[p_X, p_X]$, the mutual information of two r.v.s $X$ and $Y$ as $\mathbb{I}[Y; X] := \mathbb{D}_{\text{KL}}[p_{Y,X} \| p_Y p_X]$, and the conditional entropy as $\mathbb{H}[Y \mid X] := \mathbb{E}_{p_{Y,X}}[-\log p(Y \mid X)]$.

**Other.** We will use $\phi_\rightarrow(\mathcal{X}) := \{\phi(x) | x \in \mathcal{X}\}$ to denote the image of a set $\mathcal{X}$ by a function $\phi$ (here the set of all representations). We use $\dim_\mathbb{R}(V)$ to denote the dimensionality of a vector space $V$ over the reals. We use $\dim(v)$ to denote the dimensionality of a vector $v$. We denote by $\mathcal{Y}^\mathcal{X} := \{f | f : \mathcal{X} \to \mathcal{Y}\}$ the set of all functions from $\mathcal{X} \to \mathcal{Y}$. We denote one hot encodings as $\mathbf{e}_{|\mathcal{Y}|}(i)$ which is a vector in $\{0, 1\}^{|\mathcal{Y}|}$ filled with 0 at all but the i$^{\text{th}}$ position which is a 1. We use $\mathbb{1}$ to denote the indicator function. For vector/tensor manipulation we use PyTorch's notation (similar to NumPy and Matlab). For example, if $W \in \mathbb{R}^{r \times c}$ then $W[:, j]$ is the j$^{\text{th}}$ column of W which is in $\mathbb{R}^r$. Similarly, $W[:2, :]$ denotes the $\mathbb{R}^{2 \times c}$ matrix from the first two rows of $W$, which can also be denoted as $\text{cat}(W[0, :], W[1, :])$.

Finally, for any functions $f$ and $g$ we use $g(f)$ to denote the composed function $g \circ f$. For example, when $X$ is a r.v. we use the standard $f(X) := f \circ X$. We do the same beyond r.v.s, e.g., let $f_1 : \mathcal{Z} \to \mathbb{R}$ and $f_k : \mathcal{Z} \to \mathbb{R}^k$ then $2f_1$ denotes the function $z \mapsto 2f(z)$ and $\text{cat}(f_k, f_1)$ denotes a function $\mathcal{Z} \to \mathbb{R}^{k+1}$ which concatenates outputs $z \mapsto \text{cat}(f_k(z), f_1(z))$. To make the function explicit we will sometimes use $(\cdot)$, e.g., $f_k(\cdot)[i]$ denotes the function $z \mapsto f_k(z)[i]$.

## A.2 Assumptions

We make the following assumptions throughout the paper.

**Assumption 1** (Convenience assumption: finite input space). *The input space is finite $|\mathcal{X}| \leq \infty$.*

Assmp. 1 is a convenience assumption to improve clarity by avoiding unnecessary measure theory, almost sure statements, and ensuring the existence of pmf and regular conditional probabilities. It always holds in practice due to floating point precision on computers. Importantly Assmp. 1 implies that the number of equivalence classes $|\mathcal{X}/\sim|$ is also finite.

Assmp. 1 can be weakened by only assuming finite $|\mathcal{X}/\sim|$. In particular, generalization to countable $\mathcal{X}$ is trivial, while generalizations to continuous $\mathcal{X}$ are also possible under minor technical assumptions such as the measurability of the canonical projection for $\sim$ and the existence of regular conditional probabilities. For an example of general proofs (under unconstrained $\mathcal{F}$) see Dubois et al. [12]. A generalization to infinite $|\mathcal{X}/\sim|$ would be much more challenging and might require more nuanced definitions of optimality.

**Assumption 2** (Non degenerate $\sim$). *The number of equivalence classes is at least two $|\mathcal{X}/\sim| \geq 2$.*

Assmp. 2 is a trivial assumption that removes uninteresting counterexamples to our theory.

**Assumption 3** (Representation space). *We assume that the representation space is a subset of real vectors of different dimensions $\mathcal{Z} \subseteq \bigcup_i \mathbb{R}^i$ that contains at least all binary vectors of dimensions up to the number of equivalence classes, i.e., $\bigcup_{d=1}^{|\mathcal{X}/\sim|} \{0, 1\}^d \subseteq \mathcal{Z}$.*

Assmp. 3 is a simplifying assumption that ensures that one-hot encodings exist, which is used to give a simple and understandable existence proof of population optimal representation. There is nothing special about 0 and 1 and those can be easily modified. Note that for non-linear predictors the maximal dimensionality of vectors in $\mathcal{Z}$ can be much smaller as proved in Appx. B.5 Note that we do not fix the dimensionality of $\mathcal{Z}$, which allows us to hold non-trivial statements about the required dimensionality of the domain of encoders.

For clarity, the main paper and most of the appendices are given for linear predictors. In Appx. B.5 we extend our theory to more general $\mathcal{F}$, which allows for finite precision linear predictors and for non-linear predictors. The following two assumptions are thus assumed throughout the paper and are trivially satisfied for linear predictors.

**Assumption 4** (At least linear predictors). All predictive families $\mathcal{F}$ have at least all linear functions as defined in Def. 12, i.e., $\mathcal{F}_{\mathrm{lin}} \subseteq \mathcal{F}$.

Assmp. 4 is a simplifying assumption, which allows us to directly reuse standard results from shatterability and VC dimensions. One could weaken the assumption to contain linear functions with finite weights (e.g. floating-point precision). Note that the upper bound on $k$ is because we only consider $\sim$- invariant tasks.

As hinted in Appx. A.1, we must deal with predictors that have different codomains $\mathbb{R}^j$ to deal with any $k$-ary classification task. Let us denote $\mathcal{F}_k \subset \mathcal{F}$ the subset of $k$-ary predictors. Those $k$-ary predictors will be related in practice, for example, we do not expect all $k$-ary predictors to be highly expressive for some $k^-$ but only linear for some larger $k^+ > k^-$. To capture this relation we will assume that $\mathcal{F}$ is closed under indexing and concatenations. In particular: (i) indexing $k^+$-ary predictors cannot outperform $k^-$-ary predictors; (ii) concatenating $k^-$-ary predictors cannot outperform $k^+$-ary predictors. In particular, this means that one-vs-all binary classifiers cannot outperform $k$-ary classifiers in $\mathcal{F}$.

**Assumption 5** ($\mathcal{F}$ closed under vector manipulation). The predictive family $\mathcal{F}$ is closed under concatenation, indexing, and summation, i.e.,

- for any $f_k, f_{k'} \in \mathcal{F}$ we have $\mathrm{cat}(f_k, f_{k'}) \in \mathcal{F}$;
- for any $k > 1$, any $i \in 0, \ldots, k-1$, and any $f_k \in \mathcal{F}_k$ we have $f_k(\cdot)[i] \in \mathcal{F}$;
- for any $f_2, f_2' \in \mathcal{F}_2$ we have $(f_2 + f_2') \in \mathcal{F}$.

Assmp. 5 is only used in Eq. (54) to ensure that binary shatterability implies $k$-ary shatterability. Although not necessary, this has the advantage of working with a shatterability definition that is similar to the one in statistical learning theory, as a result, we can rely on well-known results such as VC dimensions for different predictive families. Assmp. 5 holds when $\mathcal{F}$ is the set of linear predictors, as is standard in ISSL. For non-linear predictive families, Assmp. 5 might not always hold, in which case the VC dimension and capacity would have to be replaced by their to $k$-ary extensions, which have not been as extensively studied.

### A.3 Definitions

In the main paper, we were relatively informal in our definitions, here we restate our main definitions more formally.

A key notion is a maximal invariant introduced by Dubois et al. [12]. It is a simple extension of maximal invariants from standard statistics and group theory [64, 65].

**Definition 4** (Maximal invariant [12]). A measurable function $M : \mathcal{X} \to \mathcal{M}$ is a *maximal invariant* w.r.t. $(\mathcal{X}, \sim)$ iff

$$\forall x, x' \in \mathcal{X} : \quad x \sim x' \iff M(x) = M(x'). \tag{10}$$

Note that $M$ is not unique. For clarity, for the rest of the paper, we will refer to the maximal invariants $M$ as specific indexing of the equivalence class. In particular its codomain will be $\mathcal{M} = \{0, \ldots, |\mathcal{X}/\sim| - 1\}$. For the rest of the paper, we will also call the random variable induced by pushing the inputs through $M$, i.e., $M(X)$, as the *maximal invariant r.v.*.

The invariance structure that we want our tasks to have is based on the most likely label $\arg\max_{y \in \mathcal{Y}} p_t(y|X)$.

**Definition 5** (Invariant $k$-ary tasks). The $(\mathcal{X}, \sim)$-invariant $k$-ary tasks, denoted $\mathcal{T}_k$, is the set of all input-label distributions $p_t(X, Y)$ that satisfies the following

- it is a $k$-ary task: $|\mathcal{Y}| = \{0, \ldots, k-1\}$;

- its most likely label is unique and we denote it by $c_t(x)$, i.e.,

$$\forall p_t \in \mathcal{T}_k, \ x \in \mathcal{X} : \quad |c_t(x)| := |\arg\max_{y \in \mathcal{Y}} p_t(y \,|\, x)| = 1; \tag{11}$$

- its most likely label is invariant, i.e.,

$$\forall p_t \in \mathcal{T}_k, \ x, x^+ \in \mathcal{X} : \quad x \sim x^+ \implies \arg\max_{y \in \mathcal{Y}} p_t(y|x) = \arg\max_{y \in \mathcal{Y}} p_t(y|x^+). \tag{12}$$

Eq. (11) is not necessary but it makes the proofs slightly more succinct. Similarly to $c_t(x)$ we denote the most likely label of a dataset as $\hat{c}_{\mathcal{D}_t}(x) := \arg\max_{y \in \mathcal{Y}_t} \hat{p}_{\mathcal{D}_t}(y|x)$.

In practice the downstream tasks of interest typically have various numbers of labels $k$, so we want to consider any possible $k$. We consider tasks that have at most $|\mathcal{X}/\sim|$ labels. More classes would be trivial and uninteresting due to the $\sim$-invariance of the most likely label.

**Definition 6** (Invariant tasks). *The $(\mathcal{X}, \sim)$-invariant tasks, denoted $\mathcal{T}$, is the set of all $(\mathcal{X}, \sim)$-invariant $k$-ary tasks for $k \in \{2, \ldots, |\mathcal{X}/\sim|\}$, i.e.,*

$$\mathcal{T} := \bigcup_{k=2}^{|\mathcal{X}/\sim|} \mathcal{T}_k. \tag{13}$$

We will say that an encoder is population optimal if the Bayes risk can be realized by probes $\mathcal{F}$.

**Definition 7** (Population optimal encoder). *We say that an encoder $\phi$ is population optimal for $\mathcal{T}, \mathcal{F}$, denoted as $\phi \in \Phi_{\text{pop}}$, iff for all task the probes $\mathcal{F}$ realize the Bayes error, i.e.,*

$$\text{for all } k \in \{2, \ldots, |\mathcal{X}/\sim|\}, \ p_t \in \mathcal{T}_k : \qquad \text{R}_t(\phi, \mathcal{F}_k) = \text{R}_t^*. \tag{14}$$

The "for all $k$ and task $p_t \in \mathcal{T}_k$" statement in Eq. (14) can be succinctly written as "for all $p_t \in \mathcal{T}$", by taking the best risk over $\mathcal{F}$ instead of $\mathcal{F}_k$. For conciseness, we do so for the main paper and for the rest of the appendices.

Population optimal only ensures that there exists a good predictor, which will be essentially learned if we had infinite downstream labeled data. In practice, we only have access to finite datasets $\mathcal{D}_t$ of possibly small size $n$ (e.g. few-shot learning). We would thus like encoders that also ensure that all ERMs perform as well as possible for any task and any dataset size $n$. We measure this using the following worst-case expected excess risk of ERMs.

**Definition 8** (Excess risk of ERMs). *The excess risk of ERMs for $\phi, \mathcal{F}, p_t, \mathcal{D}_t$ is the maximal difference between the population risk of ERMs in $\mathcal{F}$ and the Bayes error, i.e.,*

$$\text{W}(\phi, \mathcal{F}, t, \mathcal{D}_t) := \sup_{\hat{f} \in \widehat{\mathcal{F}}(\mathcal{D}_t, \phi)} \text{R}_t(\phi, \hat{f}) - \text{R}_t^*. \tag{15}$$

For simplicity and conciseness, we will work with datasets of examples with their associated most likely labels. This is always the case for $i.i.d.$ datasets with deterministic labels as in the main paper. We will denote by $\mathcal{D}_t \overset{\text{i.i.d.}}{\sim} p_t^n(X, c_t(X))$ as $n$ input-label r.v. that are given by first sampling $n$ $i.i.d.$ unlabeled inputs $\mathcal{D}_t^x \overset{\text{i.i.d.}}{\sim} p_t^n(X)$ and then labeling with the most likely label $c_t$, i.e., $\mathcal{D}_t := \{(X_i, c_t(X_i))|X_i \in \mathcal{D}_t^x\}$.

**Definition 9** (Worst-case expected excess risk of ERMs). *The worst-case expected excess risk of ERMs for $n, \phi, \mathcal{F}, \mathcal{T}$ is the worst-case (over tasks) expected (over datasets of size $n$) excess risk of ERMs, i.e.,*

$$\text{W}_n(\phi, \mathcal{F}, \mathcal{T}) := \sup_{k \in \{2, \ldots, |\mathcal{X}/\sim|\}} \sup_{p_t \in \mathcal{T}^k} \mathbb{E}_{\mathcal{D}_t \overset{\text{i.i.d.}}{\sim} p_t^n(X, c_t(X))}[\text{W}(\phi, \mathcal{F}_k, t, \mathcal{D}_t)]. \tag{16}$$

Note that the first two supremums in Eq. (16) can be succinctly written as $\sup_{t \in \mathcal{T}}$ as we do in the main paper (Eq. (3) of main paper), in which case the ERMs $\widehat{\mathcal{F}}(\mathcal{D}_t, \phi)$ are implicitly taken from predictors with the correct codomain.

We will say that an encoder is sample optimal if it is population optimal and minimizes the worst-case excess risk of ERMs for any dataset size $n$. We will call the representations induced by sample optimal encoders as *idealized representations*.

**Definition 10** (Sample optimal encoders). An encoder $\phi^*$ is *sample optimal* for $\mathcal{T}, \mathcal{F}$, denoted $\phi^* \in \Phi_*$, iff it is population optimal[2] and minimizes the worst-case expected excess risk of ERMs for any $n$, i.e.,

$$\forall n \geq 1: \quad \phi^* \in \underset{\phi \in \Phi_{\text{pop}}}{\arg\min} W_n(\phi, \mathcal{F}, \mathcal{T}). \tag{17}$$

Under Assmp. 5 (which holds for linear $\mathcal{F}$) we will show that encoders are population optimal if and only if the equivalence classes can be shattered from their representations.

**Definition 11** (Invariant shattering). Let $\mathcal{C}_k$ denote the set of $(\mathcal{X}, \sim)$-invariant functions in $\{0, \ldots, k-1\}^{\mathcal{X}}$, dubbed invariant labelings. An encoder $\phi$ is $k$-ary $(\mathcal{X}, \sim)$-*shattered by* $\mathcal{F}$ iff any $k$-ary invariant labeling can be predicted by an $f \in \mathcal{F}$, i.e.,

$$\forall c \in \mathcal{C}_k, \ \exists f \in \mathcal{F} \text{ s.t. } \forall x \in \mathcal{X}: \quad \text{pred}(f(\phi(x))) = c(x). \tag{18}$$

When $k = 2$ we drop "2-ary" and say that $\phi$ is $(\mathcal{X}, \sim)$-shattered by $\mathcal{F}$.

Note that, even in the binary case, Def. 11 differs from the classical notion of shattering in that only invariant labelings need to be predictable. Generally, (classical) shatterability by $\mathcal{F}$ thus implies $(\mathcal{X}, \sim)$-shatterability by $\mathcal{F}$ but the converse is not true.

For the next few sections, we will only focus on linear predictors $\mathcal{F}$.

**Definition 12** (Linear predictors). The set of linear predictors $\mathcal{F}$ for $\mathcal{T}$ is the set of all linear functions with the correct domain and codomain for predicting any $\mathcal{T}$, i.e.,

$$\mathcal{F}_{\text{lin}} := \bigcup_{d \in \{\dim(z) | z \in \mathcal{Z}\}} \bigcup_{k=1}^{|\mathcal{X}/\sim|} \left\{ f : z \mapsto W^T z \,|\, W \in \mathbb{R}^{d \times k} \right\}. \tag{19}$$

It is easy to show that Def. 12 satisfies (but is stronger than) Assumps. 4 and 5. In Appx. B.5 we extend our proofs to any family of predictors $\mathcal{F}$ that only satisfies Assumps. 4 and 5. Note that we will never use predictors with two-dimensional output (as binary tasks use a single logit), the union over $k$ could then drop the case $k = 2$.

Finally, the main loss that we will aim to approximate is what we call the ISSL log loss.

**Definition 13** (ISSL log loss). Let $\mathcal{S} := \{z \in \mathbb{R}^d \,|\, \|z\| = 1\}$ denote the $(d-1)$-sphere. Let $\mathcal{W}_1 := \{w : \{0, \ldots, |\mathcal{X}/\sim| - 1\} \to \mathcal{S}\}$ denote all weight functions mapping a label to a normalized weight. Let $M$ denote a maximal invariant for $\sim$ with codomain $\mathcal{M} = \{0, \ldots, |\mathcal{X}/\sim| - 1\}$. The $\sim$-ISSL log loss for of an encoder $\phi$ and unlabeled distribution $p_X$ is

$$\mathcal{L}_I(\phi; p_X) := \inf_{w \in \mathcal{W}_1} \mathbb{E}_{p_X} \left[ -\log \frac{\exp\left(w(M(X))^{\top} \phi(X)\right)}{\sum_{m'=0}^{|\mathcal{X}/\sim|-1} \exp(w(m')^{\top} \phi(X))} \right], \tag{20}$$

---

[2]Under reasonable assumptions, population optimality is actually implied by minimizing the worst-case excess risk for any $n$ (seen as $n \to \infty$). The current definition helps clarity and conciseness of proofs.

# B    Proofs and additional theoretical results

## B.1    Useful known lemmas

To begin, we collect some known results about invariances and predictions that we will use for our theory.

**Lemma 4** (Maximal invariants exist, [12], Lemma 7). *There exists at least one maximal invariant* $M$.

**Lemma 5** (Invariant functions and $M$, [12], Lemma 5). *Let $M$ be any maximal invariant w.r.t.* $(\mathcal{X}, \sim)$. *Then a measurable function $f\colon \mathcal{X} \to \mathcal{S}$ is invariant with respect to $(\mathcal{X}, \sim)$ if and only if there exists a measurable function $h\colon \mathcal{M} \to \mathcal{S}$ such that $f(x) = (h \circ M)(x)$ for all $x \in \mathcal{X}$, in which case $f$ is measurable with respect to the $\sigma$-algebra generated by $M$.*

**Lemma 6** (Maximal invariant, [12], Lemma 2). *Let $M\colon \mathcal{X} \to \mathcal{M}$ be a maximal invariant w.r.t.* $(\mathcal{X}, \sim)$. *Then $M'\colon \mathcal{X} \to \mathcal{M}'$ is also a maximal invariants w.r.t. $(\mathcal{X}, \sim)$ if and only if there exists a bijective function $f\colon \mathcal{M} \to \mathcal{M}'$ such that $M' = f \circ M$.*

Note that Dubois et al. [12] only talks about the existence of a bijection between maximal invariants, but their proof shows that it is an if and only if statement.

**Lemma 7** (Hyperplane separation theorem, [14], Lemma 1). *Two sets of points in $\mathbb{R}^d$ may be separated by a hyperplane if and only if the intersection of their convex hulls is empty.*

Note that although we cite Burges [14], the hyperplane separation theorem is much older and appeared under many forms and generalizations (eg see [66–68] ). It is typically attributed to Hermann Minkowski at the end of the 19th century.

## B.2    Linear optimal (Sec. 3.1)

In this section, we characterize optimal representations for linear predictors. Table 6 shows a summary of our claims in this section. In particular, we see that only the last statement requires the assumption of linear $\mathcal{F}_{\mathrm{lin}}$ (in addition to those from Appx. A.2). The other statements will be reused to characterize representations for general $\mathcal{F}$ in Appx. B.5.

Table 6: Summary of main results from Appx. B.2

| Ref. | Summary | Add. Assmp. | Lemmas |
|------|---------|-------------|--------|
| Lemma 11 | existence of pop. optimal $\phi \in \Phi_{\mathrm{pop}}$ | 1,3,4 | 4,5,6,10 |
| Lemma 12 | optimal excess risk $\forall p_t, n, \mathcal{D}_t$ | | 5,8,9,11 |
| Corollary 13 | exist. sample optimal $\phi^* \in \Phi_*$ | | 12 |
| Lemma 15 | sample opt. $\iff$ pop. opt. + inv. | 2 | 7, 12, P. 2, C. 14 |
| Lemma 16 | pop. optimal $\iff$ $\sim$-shat. | 5 | |
| Lemma 17 | sample opt. $\iff$ inv. + shat. + $|\mathcal{X}/\sim|$ | | 6,15,16 |
| Theorem 1 | sample opt. $\iff$ M(X) + dim. + inv. | $\mathcal{F}_{\mathrm{lin}}$ | 17 |

First, let us show three trivial lemmas that come straight from definitions but will be useful for our main results. A rewriting of the excess risk, a characterization of ERMs for population optimal encoders, and proof of the existence of invariance Bayes pred.

**Lemma 8** (Nicer excess risk). *For any $\phi, \mathcal{F}, p_t, \mathcal{D}_t$ we have that the excess risk is*

$$\mathrm{W}(\phi, \mathcal{F}, t, \mathcal{D}_t) = \sup_{\hat{f} \in \widehat{\mathcal{F}}(\mathcal{D}_t, \phi)} \mathbb{E}_{p_t(X)}\left[\max_{y \in \mathcal{Y}_t} p_t(y|X) - p_t(Y = \mathrm{pred}(\hat{f}(\phi(X)))|X)\right] \tag{21}$$

*Proof.* This trivially comes from the definition of excess risk, population risk, Bayes error, 0-1 loss, and the fact that $\mathbb{E}_{p_t(Y)}[\mathbb{1}[Y \neq y]] = p_t(Y = y)$. $\qquad\square$

**Lemma 9** (Characterizing ERM). Let $p_t \in \mathcal{T}$ and $\phi \in \Phi_{\mathrm{pop}}$ be population optimal for $\mathcal{T}, \mathcal{F}$. Let $\mathcal{D}_t \in \mathrm{supp}(p_t^n(X, c_t(X)))$ be any dataset of size $n \geq 1$, which contains pairs of inputs and their most likely label. Then a predictor is an ERM if and only if it predicts the most likely label for all examples in the dataset. I.e., $\hat{f} \in \widehat{\mathcal{F}}(\mathcal{D}_t, \phi) \iff \forall x \in \mathcal{X}_{\mathcal{D}_t}$ we have $\arg\max_{y \in \mathcal{Y}_t} p_t(y|x) = \mathrm{pred}(\hat{f}(\phi(x)))$

*Proof.* This trivially comes from the definition of 0-1 loss, the definition of ERMs, the definition of population optimality, the finiteness of sample space (Assmp. 1), and the fact that the most label empirical label is the most likely population label by definition. $\square$

**Lemma 10** (Existence of invariant Bayes predictor). For any invariant task $p_t \in \mathcal{T}$ there exists a Bayes predictor denoted as $b_t^*$ that is invariant *w.r.t.* $(\mathcal{X}, \sim)$ and whose predictions are one hot encodings.

*Proof.* $b_t^*(x) := \mathbf{e}_{|\mathcal{Y}_t|}(c_t(x))$, which exists by Assmp. 3, is a Bayes predictor by construction as $\mathrm{pred}(b_t^*(x)) = c_t(x) = \arg\max_{y \in \mathcal{Y}_t} p_t(y|x)$. Furthermore, by Lemma 5, $b_t^*$ is invariant as it is a function of $\arg\max_{y \in \mathcal{Y}_t} p_t(y|X)$ which is invariant by definition , see Eq. (12). $\square$

Now let us show that population-optimal encoders exist.

**Lemma 11** (Existence of pop. optimal $\phi$). There exists a population-optimal encoder for $\mathcal{T}, \mathcal{F}$.

*Proof.* This can easily be seen by taking $\phi$ to be a one-hot representation of the equivalence class and $f \in \mathcal{F}$ be the necessary aggregation function represented as a linear equation with binary weights.

Specifically, let $\phi_{\mathbf{e}}(x) = \mathbf{e}_d(M(x))$ be an encoder that maps any inputs $x \in \mathcal{X}$ to a one hot encoding of an index of the equivalence classes $M : \mathcal{X} \to \{0, \ldots, |\mathcal{X}/\sim| - 1\}$. Such an encoder exists by Assmp. 3 and Lemma 4. By Lemma 6 we know that $\phi_{\mathbf{e}}$ is a maximal invariant w.r.t. $(\mathcal{X}, \sim)$ as it is a composition between a bijection (the one hot encoding) and a maximal invariant. By Lemma 10, we have that for any $p_t \in \mathcal{T}$ there exists an invariant Bayes predictor. By Lemma 6 there must thus exist a function $f_t : \mathcal{Z} \to \mathcal{Y}$ s.t. $\forall x \in \mathcal{X}$ we have $f_t(\phi_{\mathbf{e}}(x)) = b_t^*$ . As predictions by the Bayes predictor $b_t^*$ (by construction see Lemma 10) is also a one hot encoding, we have that $f_t$ is a function between one hot encodings and can thus be represented by a linear function with binary weights, i.e., $f_t(\phi_{\mathbf{e}}(x)) = W^\top \phi_{\mathbf{e}}(x)$ where $W \in \{0, 1\}^{|\mathcal{X}/\sim| \times |\mathcal{Y}_t|}$. As $f_t$ is linear it is by Assmp. 4 in $\mathcal{F}$.

Putting all together we have that for any $p_t \in \mathcal{T}$ there exists an $f_t \in \mathcal{F}$ so that $\forall x \in \mathcal{X}$ we have $f_t(\phi(x)) = b_t^*$. In particular we have $\mathrm{R}_t(\phi_{\mathbf{e}}, \mathcal{F}) = \mathrm{R}_t^*$, so $\phi_{\mathbf{e}}$ is population-optimal as desired. $\square$

Now let us show that there exists an encoder that is essentially optimal for any task and dataset.

**Lemma 12** (Optimal excess risk). Let $\Phi_{\mathrm{pop}}^{\sim} \subset \Phi_{\mathrm{pop}}$ denote the set of population-optimal encoders that are $(\mathcal{X}, \sim)$-invariant. Then for any invariant task and dataset, the excess risk of ERMs for any $\phi^* \in \Phi_{\mathrm{pop}}^{\sim}$ is as small as possible, i.e.,

$$\forall \phi^* \in \Phi_{\mathrm{pop}}^{\sim}, \, p_t \in \mathcal{T}, \, n \geq 1, \, \mathcal{D}_t \in \mathrm{supp}(p_t^n(X, c_t(X))) : \quad \phi^* \in \arg\min_{\phi \in \Phi_{\mathrm{pop}}} \mathrm{W}(\phi, \mathcal{F}, t, \mathcal{D}_t). \quad (22)$$

Furthermore, $\forall \phi^* \in \Phi_{\mathrm{pop}}^{\sim}, \, p_t \in \mathcal{T}, \, n \geq 1, \, \mathcal{D}_t \in \mathrm{supp}(p_t^n(X, c_t(X)))$ the excess risk of ERMs is

$$\mathrm{W}(\phi^*, \mathcal{F}, t, \mathcal{D}_t) = \sum_{[x] \notin \mathcal{X}_{\mathcal{D}_t}/\sim} p_t([x]) \big( \max_y p_t(y|[x]) - \min_y p_t(y|[x]) \big). \quad (23)$$

*Proof.* We will first compute a lower bound on the worst-case expected excess risk and then show that this lower bound can be achieved by the one-hot encoding of the maximal invariant $\phi_{\mathbf{e}}$ (see proof of Lemma 11). As a reminder we denote by $\mathcal{X}_{\mathcal{D}_t}/\sim$ the equivalence classes seen during training and by $p_t(Y|[x])$ the distribution of labels for an equivalence class.

The key is to realize and show that for any invariant task, sample size, dataset, and population-optimal encoder there always exists an ERM that predicts correctly an example if and only if an equivalent

example is in the training set. Specifically, $\forall p_t \in \mathcal{T}$, $n \geq 1$, $\mathcal{D}_t \in \mathrm{supp}(p_t^n(X, c_t(X)))$, $\phi \in \Phi_{\mathrm{pop}}$ there exists $\hat{f}_{\mathcal{D}_t,\phi} \in \mathcal{F}$ such that $\forall x \in \mathcal{X}$ we have

$$\mathrm{pred}(\hat{f}_{\mathcal{D}_t,\phi}(\phi(x))) = l_{\mathcal{D}_t,t}(x) := \begin{cases} \arg\max_{y \in \mathcal{Y}_t} p_t(y|[x]) & \text{if } [x] \in \mathcal{X}_{\mathcal{D}_t}/\sim \\ \arg\min_{y \in \mathcal{Y}_t} p_t(y|[x]) & \text{otherwise.} \end{cases} \tag{24}$$

To see that such $\hat{f}_{\mathcal{D}_t,\phi}$ always exists, notice that $l_{\mathcal{D}_t,t}(x)$ is a deterministic labeling that by Lemma 5 is invariant to $\sim$ because it is a function of the maximal invariant ( Eq. (24) only depends on the input $x$ through $[x]$ which is a maximal invariant). As a result we can construct an invariant task $p_{t'} \in \mathcal{T}$ for which $l_{\mathcal{D}_t,t}(x)$ is the only Bayes predictor. For example, let the input distribution of that new task be $p_{t'}(X) = \mathrm{Unif}(\mathcal{X})$ and the label be deterministic and given by $l_{\mathcal{D}_t,t}$, i.e., $\forall x, y \in \mathrm{supp}(p_{t'}(X, Y))$ we have $y = l_{\mathcal{D}_t,t}(x)$. As $\phi$ is population-optimal by assumption, there must exist a predictor $\hat{f}_{\mathcal{D}_t,\phi} \in \mathcal{F}$ such that $\forall x \in \mathcal{X}$ we have $\mathrm{pred} \circ \hat{f}_{\mathcal{D}_t,\phi} \circ \phi(x) = l_{\mathcal{D}_t,t}(x)$. Where we also used the finiteness of sample spaces, the fact that by construction we have $\mathrm{supp}(p_{t'}(X)) = \mathcal{X}$, and the definition of 0-1 loss.

By construction and Lemma 9, $\forall p_t \in \mathcal{T}$, $n \geq 1 \mathcal{D}_t \in \mathrm{supp}(p_t^n(X, c_t(X)))$, $\phi \in \Phi_{\mathrm{pop}}$ we have that $\hat{f}_{\mathcal{D}_t,\phi} \in \widehat{\mathcal{F}}(\mathcal{D}_t, \phi)$ is an ERM as it predicts the most likely label $c_t(x)$ for all examples that are equivalent to those seen during training (first case in Eq. (24)) including those seen during training.

In particular this means that the excess risk of population-optimal encoders can be lower bounded by the risk of this predictor for all $p_t \in \mathcal{T}$, $n \geq 1$, $\mathcal{D}_t \in \mathrm{supp}(p_t^n(X, c_t(X)))$, $\phi \in \Phi_{\mathrm{pop}}$ we have

$$W(\phi, \mathcal{F}, t, \mathcal{D}_t) \tag{25}$$

$$= \sup_{\hat{f} \in \widehat{\mathcal{F}}(\mathcal{D}_t,\phi)} \mathbb{E}_{p_t(X)}\left[\max_{y \in \mathcal{Y}_t} p_t(y|X) - p_t(Y = \mathrm{pred}(\hat{f}(\phi(X)))|X)\right] \qquad \text{Lemma 8} \tag{26}$$

$$\geq \mathbb{E}_{p_t(X)}\left[\max_y p_t(y|X) - p_t(Y = \mathrm{pred}(\hat{f}_{\mathcal{D}_t,\phi}(\phi(X)))|X)\right] \qquad \phi \in \Phi_{\mathrm{pop}} \tag{27}$$

$$= \sum_{[x] \in \mathcal{X}_{\mathcal{D}_t}/\sim} p_t([x])\left(\max_y p(y|[x]) - \max_y p_t(y|[x])\right) \tag{28}$$

$$+ \sum_{[x] \notin \mathcal{X}_{\mathcal{D}_t}/\sim} p_t([x])\left(\max_y p_t(y|[x]) - \min_y p_t(y|[x])\right) \qquad \text{Eq. (24)} \tag{29}$$

$$= \sum_{[x] \notin \mathcal{X}_{\mathcal{D}_t}/\sim} p_t([x])\left(\max_y p_t(y|[x]) - \min_y p_t(y|[x])\right), \tag{30}$$

where Eq. (27) uses the fact that $\phi \in \Phi_{\mathrm{pop}}$ and so $\hat{f}_{\mathcal{D}_t,\phi} \in \widehat{\mathcal{F}}(\mathcal{D}_t, \phi)$ as we have just proven. Eq. (29) partitions the equivalence class $\mathcal{X}/\sim$ into examples those that were in the dataset $\mathcal{D}_t$ and those that are not, which correspond to both cases in the definition of $\hat{f}_{\mathcal{D}_t,\phi}$.

We will now show that any population-optimal and invariant encoder $\phi^* \in \Phi_{\mathrm{pop}}^{\sim}$ achieves this lower bound. Note that such encoder exists, for example, the one-hot encodings of the maximal invariant $\phi_{\mathbf{e}}$ from the proof of Lemma 11. Specifically, for all $p_t \in \mathcal{T}$, $n \geq 1$, $\mathcal{D}_t \in \mathrm{supp}(p_t^n(X, c_t(X)))$ we have:

$$W(\phi^*, \mathcal{F}, t, \mathcal{D}_t) \tag{31}$$

$$= \sup_{\hat{f} \in \widehat{\mathcal{F}}(\mathcal{D}_t,\phi)} \mathbb{E}_{p_t(X)}\left[\max_y p_t(y|X) - p_t(Y = \mathrm{pred}(\hat{f}(\phi^*(X)))|X)\right] \qquad \text{Lemma 8}$$

$$\tag{32}$$

$$= \sup_{\hat{f} \in \widehat{\mathcal{F}}(\mathcal{D}_t,\phi)} \sum_{[x] \in \mathcal{X}/\sim} p_t([x])\left(\max_y p_t(y|[x]) - p_t(Y = \mathrm{pred}(\hat{f}(\phi^*([x])))|[x])\right) \qquad \text{Inv.}$$

$$\tag{33}$$

$$= \sup_{\hat{f} \in \widehat{\mathcal{F}}(\mathcal{D}_t,\phi)} \left(\sum_{[x] \in \mathcal{X}_{\mathcal{D}_t}/\sim} p_t([x])\left(\max_y p_t(y|[x]) - p_t(Y = \mathrm{pred}(\hat{f}(\phi^*([x])))|[x])\right)\right) \tag{34}$$

$$+ \sum_{[x] \notin \mathcal{X}_{\mathcal{D}_t}/\sim} p_t([x]) \big( \max_y p_t(y|[x]) - p_t(Y = \text{pred}(\hat{f}(\phi^*([x])))|[x]) \big) \bigg) \tag{35}$$

$$= \sup_{\hat{f} \in \widehat{\mathcal{F}}(\mathcal{D}_t, \phi)} \sum_{[x] \notin \mathcal{X}_{\mathcal{D}_t}/\sim} p_t([x]) \big( \max_y p_t(y|[x]) - p_t(Y = \text{pred}(\hat{f}(\phi^*([x])))|[x]) \big) \qquad \boxed{\text{Lemma 9}} \tag{36}$$

$$\leq \sum_{[x] \notin \mathcal{X}_{\mathcal{D}_t}/\sim} p_t([x]) \big( \max_y p_t(y|[x]) - \min_y p_t(y|[x]) \big), \tag{37}$$

where Eq. (33) uses the fact that both the most likely label and $\phi^*$ are invariant. Eq. (36) uses the fact that $\hat{f}$ is an ERM and so it must predict the most likely label for example in the training set (Lemma 9). Eq. (37) shows that the lower bound computed in Eq. (30) is also an upper bound and so $W(\phi^*, \mathcal{F}, t, \mathcal{D}_t) = \sum_{[x] \notin \mathcal{X}_{\mathcal{D}_t}/\sim} p_t([x]) \big( \max_y p_t(y|[x]) - \min_y p_t(y|[x]) \big)$ which concludes the proof. $\qquad \square$

As a direct consequence of Lemma 12 we have that sample-optimal encoders exist.

**Corollary 13** (Existence of sample-optimal $\phi$). *Let $\phi^* \in \Phi_{\text{pop}}^\sim$ be any population-optimal encoder for $\mathcal{T}, \mathcal{F}$ that is also $(\mathcal{X}, \sim)$-invariant. Then $\phi^*$ is a sample-optimal encoder for $\mathcal{T}, \mathcal{F}$. Furthermore, such an encoder exists.*

*Proof.* By Lemma 12 we know that any invariant and a population-optimal encoder minimizes the excess risk for $p_t \in \mathcal{T}, n \geq 1, \mathcal{D}_t \in \text{supp}(p_t^n(X, c_t(X)))$. Such an encoder is sample optimal as it is population-optimal and trivially minimizes the worst-case expected excess risk for all $n \geq 1$ (minimal for all terms implies minimal in expectation and supremum). $\qquad \square$

Now let us compute the worst-case expected excess risk of sample-optimal encoders.

**Proposition 2** (Optimal worst-case expected excess risk). *Let $\phi^*$ be any sample-optimal encoder for the $(\mathcal{X}, \sim)$-invariant tasks and predictors $\mathcal{F}$. For any $n \geq 1$, the worst-case expected excess risk is*

$$W_n(\phi^*, \mathcal{F}, \mathcal{T}) = \left( 1 - \frac{1}{|\mathcal{X}/\sim|} \right)^n. \tag{38}$$

*Proof.* From the proof of Corollary 13 we know that the encoder $\phi^*$ from Lemma 12 is optimal. We can thus compute the optimal worst-case expected excess risk by computing $\phi^*$'s worst-case expected excess risk. For conciseness we will use $\mathbb{E}_{\mathcal{D}_t}$ instead of $\mathbb{E}_{\mathcal{D}_t \overset{\text{i.i.d}}{\sim} p_t^n(X, c_t(X))}$ and $\Delta_t([x]) := \max_y p_t(y|[x]) - \min_y p_t(y|[x])$. Then for any $n \geq 1$ we have:

$$W_n(\phi^*, \mathcal{F}, \mathcal{T}) \tag{39}$$

$$:= \sup_{p_t \in \mathcal{T}} \mathbb{E}_{\mathcal{D}_t} [W(\phi, \mathcal{F}, t, \mathcal{D}_t)] \tag{40}$$

$$= \sup_{p_t \in \mathcal{T}} \mathbb{E}_{\mathcal{D}_t} \left[ \sum_{[x] \notin \mathcal{X}_{\mathcal{D}_t}/\sim} p_t([x]) \cdot \Delta_t([x]) \right] \qquad \boxed{\text{Lemma 12}} \tag{41}$$

$$= \sup_{p_t \in \mathcal{T}} \sum_{[x] \in \mathcal{X}/\sim} \mathbb{E}_{\mathcal{D}_t} [\mathbb{1}[[x] \notin \mathcal{X}_{\mathcal{D}_t}/\sim]] \cdot p_t([x]) \cdot \Delta_t([x]) \tag{42}$$

$$= \sup_{p_t \in \mathcal{T}} \sum_{[x] \in \mathcal{X}/\sim} (1 - p_t([x]))^n \cdot p_t([x]) \cdot \Delta_t([x]) \tag{43}$$

$$= \sup_{p_t([x])} \sum_{[x] \in \mathcal{X}/\sim} (1 - p_t([x]))^n \cdot p_t([x]) \cdot \sup_{p_t(Y|[x])} \Delta_t([x]) \tag{44}$$

$$= \sup_{p_t([x])} \sum_{[x] \in \mathcal{X}/\sim} (1 - p_t([x]))^n \cdot p_t([x]) \cdot 1 \tag{45}$$

$$= \sup_{p_t([x])} \mathbb{E}_{p_t([X])}[(1 - p_t([x]))^n] \tag{46}$$

$$= \sup_{p_t([x])} \mathbb{E}_{p_t([X])}\left[(1 - \frac{1}{1/p_t([x])})^n\right] \tag{47}$$

$$\leq \sup_{p_t([x])} (1 - \frac{1}{\mathbb{E}_{p_t([X])}[1/p_t([x])]})^n \qquad \text{Jensen's Inequality} \tag{48}$$

$$= (1 - \frac{1}{|\mathcal{X}/\sim|})^n \tag{49}$$

where Eq. (44) uses the fact that equation in the supremum only depends on the probability of the equivalence class $p_t([x])$ and the label of the equivalence class $p_t(Y|[x])$. The latter is only used in $\Delta_t$ which is clearly maximized for deterministic tasks. Eq. (48) uses Jensen's inequality for the function $(1 - \frac{1}{x})^n$, which is concave on the domain $x \in (0, 1]$ (note that $x = 0$ corresponds to a deterministic equivalence class which is clearly not a maximum as the worst-case expected excess risk would be 0). [3] The upper bound is achieved for the uniform distribution over equivalence classes, i.e., $p_t([x]) = \frac{1}{|\mathcal{X}/\sim|}$ as seen from Eq. (46). We thus conclude that for all $n \geq 1$ the worst-case expected excess risk of sample-optimal encoders is $W_n(\phi^*, \mathcal{F}, \mathcal{T}) = \left(1 - \frac{1}{|\mathcal{X}/\sim|}\right)^n$ as desired. $\square$

As a direct corollary of the proof of Proposition 2 we can also characterize sample-optimal encoders in terms of their excess risk of specific tasks.

**Corollary 14** (Sample-optimal $\iff$ given excess risk). Let $\mathcal{T}_{\sup} \subset \mathcal{T}$ denote the set of $(\mathcal{X}, \sim)$-invariant tasks that are deterministic, i.e., $\max_y p_t(y|X) = 1$, and such that the equivalence classes are equiprobable, i.e., $\forall x \in \mathcal{X}$ we have $p_t([x]) = \frac{1}{|\mathcal{X}/\sim|}$. An encoder $\phi^*$ is sample-optimal for $(\mathcal{X}, \sim)$-invariant tasks $\mathcal{T}$ and predictors $\mathcal{F}$ if and only if it is population optimal for $\mathcal{T}, \mathcal{F}$ and for any $p_t \in \mathcal{T}_{\sup}$, $n \geq 1$, $\mathcal{D}_t \in \mathrm{supp}(p_t^n(X, c_t(X)))$ we have:

$$W(\phi^*, \mathcal{F}, t, \mathcal{D}_t) = \sum_{[x] \notin \mathcal{X}_{\mathcal{D}_t}/\sim} p_t([x]). \tag{50}$$

*Proof.* In the proof of Proposition 2 we have seen that for sample-optimal encoders, the worst-case invariant tasks are those that are deterministic and have equiprobable equivalence classes are equiprobable, i.e., $p_t \in \mathcal{T}_{\sup}$. As the input space is finite (Assmp. 1), $n$ is finite, and the labeling of the dataset is deterministic we have that the expectation is minimized if and only if the excess risk is minimized for every dataset (which is possible by Lemma 12). By Lemma 12 we know that the minimum is $W(\phi^*, \mathcal{F}, t, \mathcal{D}_t) = \sum_{[x] \notin \mathcal{X}_{\mathcal{D}_t}/\sim} p_t([x])\left(\max_y p_t(y|[x]) - \min_y p_t(y|[x])\right)$. Using the determinism of labeling ($\max$ is 1 and $\min$ is 0) concludes the proof. $\square$

Proposition 2 and Corollary 14 characterizes sample-optimal encoders in terms of the (worst-case expected) excess risk that they achieve. Such characterization does not give much insight into the form of those encoders or the resulting representations. We can nevertheless use it to show that sample-optimal encoders must be invariant.

**Lemma 15** (Sample opt. $\iff$ pop. opt + inv. ). An encoder $\phi^*$ is sample-optimal for and $(\mathcal{X}, \sim)$-invariant tasks $\mathcal{T}$ and predictors $\mathcal{F}$ if and only if it is population optimal for $\mathcal{T}, \mathcal{F}$ and invariant w.r.t. $(\mathcal{X}, \sim)$, i.e., $\Phi_* = \Phi_{\mathrm{pop}}^{\sim}$.

*Proof.* In the following, we use $\mathrm{conv}(\mathcal{Z})$ to denote the convex hull of a set $\mathcal{Z}$ and $\mathrm{ext}(\mathcal{S})$ to denote the extreme points of a convex polytope $\mathcal{S}$. See [69–71] for formal definitions of those concepts and background for the following proof.

We already know from Corollary 13 that population optimality and invariance is sufficient for sample optimality. We thus only need to show that invariance is necessary for sample optimality. We prove

---

[3]Instead of using Jensen's inequality we can also use the fact that the functional we are maximizing over pmfs is permutation invariant and has a unique solution, it must thus be maximized by the pmf of uniform distributions.

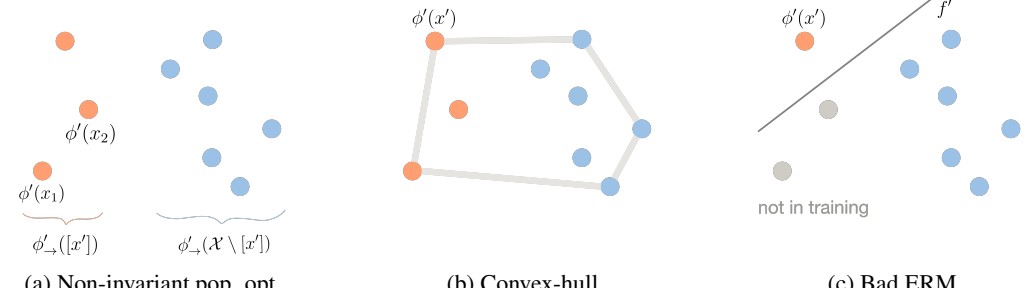

(a) Non-invariant pop. opt.  (b) Convex-hull  (c) Bad ERM

Figure 8: Illustration of the proof that non-invariant encoders cannot be sample optimal. (a) example of the representations induced by a non-invariant population-optimal encoder where colors indicate the labels of the invariant task that consist in classifying whether an example is in $[x']$; (b) the convex hull of all representations induced by a population-optimal encoder must contain at least one extreme point from every equivalence class, here $\phi'(x')$; (c) there will always be a bad ERM for the dataset containing $\phi'(x')$ and all of $\phi'_\rightarrow(\mathcal{X} \setminus [x'])$, which contradicts Corollary 14.

that by contrapositive. We use throughout the rest of the proof the fact that $\mathcal{F}$ is at least linear by Assmp. 4.

Assume that there exists a sample-optimal encoder $\phi'$ that is not $\sim$-invariant. This means that there exists an equivalence class $[x'] \in \mathcal{X}/\sim$ from which there are two points $x_1 \sim x_2 \in [x']$ that will be mapped differently by the encoder $\phi'(x_1) \neq \phi'(x_2)$. Now consider the binary task $p'_t \in \mathcal{T}_{\sup}$ of classifying whether examples come from $[x']$. I.e. $\forall x \in \mathcal{X}$ we have $p'_t(Y = 1|x) = 1$ iff $x \in [x']$, $p'_t(Y = 0|x) = 1$ iff $x \notin [x']$, and equiprobable equivalence classes $p'_t([x]) = \frac{1}{|\mathcal{X}/\sim|}$. By construction $p'_t \in \mathcal{T}_{\sup}$ and so by Corollary 14 we have that for any $n \geq 1$ and $\mathcal{D}_t \in \text{supp}(p_t'^n(X, Y))$ the excess risk is $\text{W}(\phi', \mathcal{F}, t, \mathcal{D}_t) = \sum_{[x] \notin \mathcal{X}_{\mathcal{D}_t}/\sim} p_t([x])$. In particular, if $\mathcal{X}_{\mathcal{D}_t}$ contains an example from each equivalence class, i.e., $\mathcal{X}_{\mathcal{D}_t}/\sim = \mathcal{X}/\sim$, then the excess risk of sample-optimal encoders must be 0. We will now construct a dataset $\mathcal{D}'_t$ containing at least one example per equivalence class for which $\text{W}(\phi', \mathcal{F}, t, \mathcal{D}'_t) > 0$. For an illustration of the construction see Fig. 8.

As $p_t \in \mathcal{T}$ and $\phi'$ is population-optimal (by Def. 10 and the sample optimality assumption), we have by Lemma 7 that the convex hull of the representations of $[x']$ and all other points must be disjoint to ensure that there is a hyperplane (linear probe) that can classify those sets, i.e., $\text{conv}(\phi'_\rightarrow([x'])) \cap \text{conv}(\phi'_\rightarrow(\mathcal{X} \setminus [x'])) = \emptyset$. As a result, it is easy to show that the extreme points of the convex hull of all the representations must contain at least one example in each partition, i.e., there exists $x' \in [x']$, $x \in \mathcal{X} \setminus [x']$ such that $\{x', x\} \subseteq \text{ext}(\text{conv}(\phi'_\rightarrow(\mathcal{X})))$. Now construct a dataset $\mathcal{D}'_t$ that contains only $x' \in \text{ext}(\text{conv}(\phi'_\rightarrow(\mathcal{X})))$ from $[x']$ and all other examples from other equivalence classes, i.e., $\mathcal{D}'_t := \{(x, c_{t'}(x))|x \in \mathcal{X} \setminus [x']\} \cup \{(x', c_{t'}(x'))\}\}$. By construction we have $\mathcal{D}'_t \in \text{supp}(p_t'^n(X, Y))$ for $n = |\mathcal{X} \setminus [x']| + 1$ ( where $n \geq 1$ due to Assmp. 2) and $\mathcal{X}_{\mathcal{D}_t}/\sim = \mathcal{X}/\sim$, so we must have $\text{W}(\phi', \mathcal{F}, t, \mathcal{D}'_t) = 0$. By Assmp. 1 the population risk takes an expectation over a finite number of examples $\mathcal{X}$ and so $\text{W}(\phi', \mathcal{F}, t, \mathcal{D}'_t) = 0$ if and only if for all $x \in \mathcal{X}$ we have that any ERM $\hat{f} \in \widehat{\mathcal{F}}(\mathcal{D}'_t, \phi')$ satisfies $c_{t'}(x) = \text{pred} \circ \hat{f} \circ \phi'(x)$. Let us show that this is not the case.

As $\phi'_\rightarrow(\mathcal{X})$ is finite (Assmp. 1) the number of extreme points $\text{ext}(\text{conv}(\phi'_\rightarrow(\mathcal{X})))$ must also be finite. As a result, by Strasziewicz's Theorem [72] (e.g. see Theorem 18.6 from [73] ) we have that all extreme points are also exposed points. In particular, we have that there exists a hyperplane that separates any extreme point, including $x' \in \text{ext}(\text{conv}(\phi'_\rightarrow(\mathcal{X})))$, from the rest of the points in the convex set, including all other representations $\phi'_\rightarrow(\mathcal{X} \setminus \{x'\})$. (This can be seen by the definition of exposed points, or by Lemma 7 as $\text{conv}(\phi'(x')) = \{\phi'(x')\}$ is disjoint from $\text{conv}(\phi'_\rightarrow(\mathcal{X} \setminus \{x'\}))$.) By construction, such hyperplane defines an ERM $\hat{f}' \in \widehat{\mathcal{F}}(\mathcal{D}'_t, \phi')$ as it separates correctly $\phi'(x')$ from $\phi'_\rightarrow(\mathcal{X} \setminus [x'])$. Yet, by construction, it also separates $\phi'(x')$ from all other equivalent examples that do not have the same representation $\phi'_\rightarrow([x'] \setminus \{x'\})$, which we know exist as we previously saw that $x_1 \sim x_2 \in [x']$ but $\phi(x_1) \neq \phi(x_2)$. In summary, $\hat{f}' \in \widehat{\mathcal{F}}(\mathcal{D}'_t, \phi')$ but does not satisfy $c_{t'}(x) = \text{pred} \circ \hat{f}' \circ \phi'(x)$ for $x \in [x'] \setminus \{x'\} \subset \mathcal{X}$. We thus found a bad ERM which contradicts the statement that $\phi'$ is sample optimal but not invariant. We conclude that any sample-optimal encoder is $\sim$-invaraint as desired. □

Lemma 15 is a nice characterization of optimal encoders in that invariance of encoders is a simple property to think about and, at least in theory, it is easy to see how to optimize for invariant encoders using augmentations. The population-optimality requirement is nevertheless still not very useful as it suggests having to perform all invariant tasks to get sample-optimal encoders. We now show that under Assmp. 5 (which holds for linear probing families) we can instead only consider binary invariant tasks. The main idea intuition is that we can always achieve predict a desired $k$-ary labeling by combining (due to Assmp. 5) a $(k-1)$-ary predictor with the binary predictor that distinguishes the wrong predictions from the rest.

**Lemma 16** ($\sim$-shatter. $\iff$ pop. opt. ). *An encoder $\phi \in \Phi_{\mathrm{pop}}$ is population optimal for $\mathcal{T}, \mathcal{F}$ if and only if it is $(\mathcal{X}, \sim)$-shattered by $\mathcal{F}$.*

*Proof.* First, notice that trivially an encoder $\phi$ is population optimal for $\mathcal{T}_k, \mathcal{F}$ if and only if it is $k$-ary $(\mathcal{X}, \sim)$-shattered by $\mathcal{F}$. Indeed, for any task $p_t \in \mathcal{T}$ the most likely label $c_t$ is by Def. 5 a $k$-ary invariant labeling. By Def. 11 we must thus have that $\exists f \in \mathcal{F}$ such that $\forall x \in \mathcal{X}$ we have $\mathrm{pred}(f(\phi(x))) = c_t(x)$ which is equivalent to the definition of population-optimal for $\mathcal{T}$ due to the definition of 0-1 loss and the finite sample space Assmp. 1. We thus have that an encoder is population optimal for $\mathcal{T}_k, \mathcal{F}$ if and only if it is $k$-ary $(\mathcal{X}, \sim)$-shattered by $\mathcal{F}$ for any $k \in \{2, \ldots, |\mathcal{X}/\sim|\}$.

Now let us prove that binary $\sim$-shatterability is equivalent to $k$-ary $\sim$-shatterability for any $k \in \{2, \ldots, |\mathcal{X}/\sim|\}$. We will prove the statement by induction, i.e., we suppose that for any $1 < i < k$ we have that $i$-ary $\sim$-shatterability holds and want to prove that it implies $k$-ary $\sim$-shatterability. We will prove the induction by contradiction. Suppose that $k$-ary shatterability does not hold. Then by definition there exists a $k$-ary invariant labeling $c_k \in \mathcal{C}_k$ s.t. for all $f_k \in \mathcal{F}$ there exists a $x \in \mathcal{X}$ s.t. $\mathrm{pred}(f_k(\phi(x))) \neq c_k(x)$. Construct $c_{k-1} \in \mathcal{C}_{k-1}$ by merging the two last classes, i.e.,

$$c_{k-1}(x) := \begin{cases} c_k(x) & \text{if } i \in \{0, \ldots, k-2\} \\ k-1 & \text{if } i \in \{k-1, k\} \end{cases} \tag{51}$$

By construction $c_{k-1} \in \mathcal{C}_{k-1}$ is a $(k-1)$-ary invariant labeling so by the induction assumption there exists an $f_{k-1} \in \mathcal{F}$ s.t. $\mathrm{pred}(f_{k-1}(\phi(x))) = c_{k-1}(x)$ for all $x \in \mathcal{X}$. Now let $c_2, c_2'$ be functions that indicates whether the unpredictable labeling is equal to the last class $c_2 : x \mapsto \mathbb{1}[c_k(x) = k-1]$ and similarly $c_2' : x \mapsto \mathbb{1}[c_k(x) \neq k-1]$. By construction $c_2, c_2'$ are binary invariant labeling, so again by the induction assumption there exists an $f_2, f_2' \in \mathcal{F}$ s.t. $\forall x \in \mathcal{X}$ we have $\mathrm{pred}(f_2(\phi(x))) = c_2(x)$ and $\mathrm{pred}(f_2'(\phi(x))) = c_2'(x)$.

By Assmp. 5 we can then use $f_{k-1}, f_2, f_2'$ to construct the desired an $f_k \in Q$ satisfying $\mathrm{pred}(f_k(\phi(x))) = c_k(x)$. Specifically, we can construct the function mapping to the logits of class $k-1$ by $f^{k-1} := f_{k-1}(\cdot)[k-1] + f_2'$, the function mapping to the logits of class $k$ by $f^k := f_{k-1}(\cdot)[k-1] + f_2$ and then concatenate all the logits to get the desired $f_k^* := \mathrm{cat}(f_{k-1}(\cdot)[:k-2], f_k^{k-1}, f_k^k)$. Indeed, by construction the first $k-2$ classes were already dealt with correctly, and we simply used $f_2$ and $f_2'$ to add a positive component to the right class (we used two functions $f_2, f_2'$ to deal with zero values logits) and distinguish examples from class $k$ and $k-1$. The resulting function thus satisfies $\mathrm{pred}(f_k(\phi(x))) = c_k(x)$ for all $x \in \mathcal{X}$. This leads to a contradiction. We thus have that $k$-ary shatterability, which concludes the proof due to induction (the base case being binary). $\square$

Lemma 16 shows that population optimality is equivalent to (some invariant notion of) shatterability. Now let us show that for invariant encoders this is equivalent to the classical notion of shatterability from statistical learning theory [e.g. 74].

**Lemma 17** (Sample opt. $\iff$ max inv + classical shatt. ). *An encoder $\phi^*$ is sample-optimal for $(\mathcal{X}, \sim)$-invariant tasks $\mathcal{T}$ and predictors $\mathcal{F}$ if and only if*

- *$\phi^*$ is invariant w.r.t. $(\mathcal{X}, \sim)$;*
- *all the representations $\phi_\rightarrow^*(\mathcal{X})$ are classically shattered by $\mathcal{F}$;*
- *non-equivalent examples are not encoded to the same representation, i.e., $|\phi_\rightarrow^*(\mathcal{X})| = |\mathcal{X}/\sim|$.*

*Proof.* From Lemmas 15 and 16 we know that an encoder $\phi^*$ is sample optimal for $\mathcal{T}, \mathcal{F}$ if and only if it is invariant w.r.t. $(\mathcal{X}, \sim)$ and $\sim$-shattered by $\mathcal{F}$. We will now show that, for an invariant encoder,

$\sim$-shatterability by $\mathcal{F}$ is equivalent to classical shatterability of the representations and having $|\mathcal{X}/\!\sim|$ different representations.

Let us denote by $\mathcal{X}_\sim \subset \mathcal{X}$ some arbitrary set of example that contains a single example per equivalence class, i.e., $\forall [x] \in \mathcal{X}/\!\sim, |\mathcal{X}_\sim \cap [x]| = 1$. Let us also denote the by $\mathcal{Z}_\sim := \phi^*_\rightarrow(\mathcal{X}_\sim)$ the representations of those examples, and by $\mathcal{Z}_{\phi^*} := \phi^*_\rightarrow(\mathcal{X})$ all the representations induced by the encoder. Starting from the $\sim$-shatterability definition we have:

$$\forall c \in \mathcal{C}_2, \; \exists f \in \mathcal{F}_{\text{lin}} \text{ s.t. } \forall x \in \mathcal{X}: \qquad \text{pred}(f(\phi^*(x))) = c(x) \qquad\qquad (52)$$

$$\iff \forall c \in \{0,1\}^{\mathcal{X}_\sim}, \; \exists f \in \mathcal{F}_{\text{lin}} \text{ s.t. } \forall x \in \mathcal{X}_\sim: \quad \text{pred}(f(\phi^*(x))) = c(x) \qquad \text{Inv.} \quad (53)$$

$$\implies \forall c_Z \in \{0,1\}^{\mathcal{Z}_\sim}, \; \exists f \in \mathcal{F}_{\text{lin}} \text{ s.t. } \forall z \in \mathcal{Z}_\sim: \quad \text{pred}(f(z)) = c_Z(z) \qquad \sim\text{-shat.} \quad (54)$$

$$\iff \forall c_Z \in \{0,1\}^{\mathcal{Z}_{\phi^*}}, \; \exists f \in \mathcal{F}_{\text{lin}} \text{ s.t. } \forall z \in \mathcal{Z}_{\phi^*}: \quad \text{pred}(f(z)) = c_Z(z) \qquad \text{Inv.} \quad (55)$$

where Eq. (53) uses the fact that both the encoder and the labeling are invariant; Eq. (54) uses the population optimality of $\phi^*$ (equivalent $\sim$ shatterbilty by Eq. (54)) which implies that any invariant function can be written as a function of its induced representation; and Eq. (55) uses $\mathcal{Z}_\sim = \mathcal{Z}_{\phi^*}$ due to invariance of the encoder. Eq. (54) thus shows that the restriction of $\mathcal{F}_{\text{lin}}$ to $\mathcal{Z}_{\phi^*}$ is the set of all binary functions $\{0,1\}^{\mathcal{Z}_{\phi^*}}$ which is the definition of classical shattering of $\mathcal{Z}_{\phi^*}$ by $\mathcal{F}$ (e.g. see [74]).

We thus have that sample-optimality implies invariance and classical shattering. To get an if and only if we need to ensure that $\phi^*$ is a maximal invariant, such that by Lemma 6 any invariant labelings $\{0,1\}^{\mathcal{Z}_{\phi^*}}$ from Eq. (53) can be written as a function $c_Z \in \{0,1\}^{\mathcal{Z}_\sim}$ of $\phi^*(x)$ as in Eq. (54). By Def. 4 $\phi^*$ is a maximal invariant for $\sim$ if and only if $x \sim x^+ \iff \phi^*(x) = \phi^*(x^+)$ which is equivalent to $\sim$-invariance of $\phi^*$ and $|\phi^*_\rightarrow(\mathcal{X})| = |\mathcal{X}/\!\sim|$ as desired. $\qquad\qquad\square$

As shatterability is well studied in standard statistical learning results, we can use many classical results to get our desired characterization. In particular, shatterability is related to the VC dimension [13] of the predictors $\mathcal{F}$. Here we give the desired characterization for linear $\mathcal{F}$. A similar characterization for more general $\mathcal{F}$ can be found in Appx. B.5.

**Theorem 1** (Sample-optimal encoders for linear probes). Let $\mathcal{T}$ be all invariant tasks w.r.t. $(\mathcal{X}, \sim)$, and $\mathcal{F}_{\text{lin}}$ be the set of linear predictors for $\mathcal{T}$. An encoder $\phi^*$ is sample optimal for $\mathcal{T}, \mathcal{F}_{\text{lin}}$ if and only if it satisfies the following properties:

- **Dimensionality**: the dimensionality of the span of all possible representations is at least one less than the number of equivalence classes, i.e.,

$$\dim_{\mathbb{R}}(\text{span}(\phi^*_\rightarrow(\mathcal{X}))) \geq |\mathcal{X}/\!\sim| - 1. \qquad\qquad (56)$$

- $\mathcal{F}$-**predictability of** $M$: there exists a maximal invariant $M : \mathcal{X} \rightarrow \{0, \ldots, |\mathcal{X}/\!\sim| - 1\}$ w.r.t. $(\mathcal{X}, \sim)$ that is predictable by $\mathcal{F}_{\text{lin}}$ from $\phi^*$, i.e.,

$$\exists M, f \in \mathcal{F}_{\text{lin}}: \quad \forall x \in \mathcal{X}: \quad M(x) = \text{pred}(f(\phi^*(x))). \qquad\qquad (57)$$

- **Invariance**: the encoder $\phi^*$ is invariant w.r.t. $(\mathcal{X}, \sim)$, i.e.,

$$\forall x, x^+ \in \mathcal{X}: \quad x \sim x^+ \implies \phi^*(x) = \phi^*(x^+) \qquad\qquad (58)$$

*Proof.* From Lemma 17 we know that an encoder $\phi^*$ is sample optimal for $\mathcal{T}, \mathcal{F}$ if and only if it is invariant w.r.t. $(\mathcal{X}, \sim)$, $\phi^*_\rightarrow(\mathcal{X})$ is classically shattered by $\mathcal{F}$, and $|\phi^*_\rightarrow(\mathcal{X})| = |\mathcal{X}/\!\sim|$. We will now show that for linear $\mathcal{F}_{\text{lin}}$ this is equivalent to the effective dimensionality requirement from Eq. (56). Indeed, by standard statistical learning theory results (e.g. Theorem 1 from [14] which can be shown using Lemma 7) we know that any set of $|\mathcal{X}/\!\sim|$ points $\mathcal{Z}_{\phi^*}$ in $\mathbb{R}^n$ can be classically shattered by $\mathcal{F}_{\text{lin}}$ if and only if when choosing any point any point $z \in \mathcal{Z}_{\phi^*}$ as the origin we have that all other $\mathcal{Z}_{\phi^*} \setminus z$ are linearly independent. Equivalently, any set of $|\mathcal{X}/\!\sim|$ points $\mathcal{Z}_{\phi^*}$ in $\mathbb{R}^n$ can be classically shattered by $\mathcal{F}_{\text{lin}}$ if and only if $\dim_{\mathbb{R}}(\text{span}(\mathcal{Z}_{\phi^*})) \geq |\mathcal{X}/\!\sim| - 1$.[4]

Now let us show that $\mathcal{F}_{\text{lin}}$-predictability of $M$ is necessary. Indeed, any maximal invariant $M : \mathcal{X} \rightarrow \{0, \ldots, |\mathcal{X}/\!\sim| - 1\}$ is a $|\mathcal{X}/\!\sim|$ invariant labeling $M \in \mathcal{C}_{|\mathcal{X}/\!\sim|}$ and is thus implied by $\sim$-shatterability

---

[4]The $\geq$ could be replaced by a $=$ as the effective dimensionality of $|\mathcal{X}/\!\sim|$ can never be larger than $|\mathcal{X}/\!\sim| - 1$. We use $\geq$ to make the transition to Prop. 3 more natural.

by $\mathcal{F}_{\mathrm{lin}}$ (due to Lemma 16). In other words, $M$ induces a possible invariant task and thus has to be predictable. [5]                                                                                                           □

## B.3 Augmentations (Sec. 3.2)

We proved the main result from Sec. 3.2 in Proposition 2 of the previous section. The statement is more general than in the main paper as we deal with possibly stochastic labeling (but deterministic datasets).

Lemma 12 gives an even more general statement, in that it show that the excess-risk for any dataset and sample-optimal encoders (which are invariant by Lemma 5) is

$$\mathrm{W}(\phi^*, \mathcal{F}, t, \mathcal{D}_t) = \sum_{[x] \notin \mathcal{X}_{\mathcal{D}_t}/\sim} p_t([x])\big(\max_y p_t(y|[x]) - \min_y p_t(y|[x])\big). \tag{59}$$

Eq. (59) shows that for general tasks, the incurred risk does not only depend on the number of equivalence classes induced by the augmentation but also on the distribution of the equivalence classes. In particular, to decrease the risk it is better to augment more common examples. Indeed, this would make the distribution of equivalence classes less uniform which decreases the final risk (see proof of Lemma 12).

Note that the optimal excess risk in Eq. (59) only depends on the difference between the max and minimum labeling probability. If we used truly $i.i.d.$ data ( i.e. without most likely label ) then there would also be a dependence between the probability of the first and second most likely label: $\max_y p(y|x) - \max_{y \neq y'} p(y|x)$. Indeed, with true $i.i.d.$ data we would have to model the probability that the most likely label on the training set is equal to the most likely label of the population. This is equivalent to the probability that the empirical mode is equal to the population mode in discrete data. The previous dependence on $\max_y p(y|x) - \max_{y \neq y'} p(y|x)$ can then for example be seen in Theorem 4 of Dutta and Goswami [75].

Another interesting point to note is that the excess risk does not (explicitly) depend on the number of labels $k$. Note that even if the considered ERMs predicted according to the marginal $p_t(Y)$ instead of the worst-case $\min_y p_t(y|[X])$, the excess risk would not explicitly depend on $k$. The only difference is that $\max_y p_t(y|[x]) - \min_y p_t(y|[x])$ would be replaced by $\max_y p_t(y|[x]) - p_t(Y = c_t([x]))$. This would make very little difference, for example in the case of deterministic ImageNet the excess risk would only be reduced by $\frac{1000}{999}\times$, which is neglectable.

The proof of Proposition 2 and Lemma 12 shows that, due to the invariance of $\sim$-invariant encoder, computing the risk of ERMs and related quantities amount to essentially counting the number of equivalence classes that were seen during training, and so we can use standard results from combinatorics and probabilistic problems. For example, using standard coupon collector results [76] we can show that the expected dataset size to ensure that all ERMs are Bayes predictors grows as $\Theta(|\mathcal{X}/\sim| \log |\mathcal{X}/\sim|)$ when labelings are deterministic and equivalence classes are equiprobable (for weighted cases see [77]). As another example, we could compute variance or higher-order moments of the excess risk Eq. (59) for any $n$ and sample optimal $\phi^*$ using standard occupancy results [78, 79].

## B.4 ISSL Log Loss (Sec. 4)

The main result that we will use is that the encoder will learn to be invariant due to the strict convexity of the log loss. As this is a general result that might be of interest beyond our work, we prove it without assuming finite sample spaces and for any strictly convex loss function.

**Lemma 18.** Let $p_t(X, Y)$ be any joint distribution over input and targets. Let $\mathcal{F} \subseteq \{f : \mathbb{R}^d \to \mathcal{A}\}$ be any set of predictors that is .... Let $\mathrm{R}_t[\phi, \mathcal{F}, \ell] := \mathbb{E}_{p_t(X,Y)}[\ell(Y, f(X))]$ denote the best risk of probes $\mathcal{F}$ on task $p_t$ and general (not necessarily 0-1) loss.

In this section, we prove that sample-optimal encoders can be recovered by optimizing the ISSL log loss. For conciseness, we use results from the neural collapse literature, which shows that minimizing cross-entropy gives an ETF representation.

---

[5]For the linear $\mathcal{F}_{\mathrm{lin}}$, we could drop the $\mathcal{F}$-predictability requirement as it is necessary but is implied by the other requirements. We keep it to give the right intuition for the more general $\mathcal{F}$ and to help the transition to Prop. 3.

**Proposition 3** (ISSL Log Loss is sufficient). Let $\Phi_1 := \{\phi : \mathcal{X} \to \mathcal{S}\}$ be the set of encoders mapping inputs to unit-normalized representations in $\mathbb{R}^d$. Let $p_X$ be a distribution whose support is $\mathcal{X}$ and such that equivalence classes are equiprobable, i.e., for all $x \in \mathcal{X}$ we have $p_X([x]) = 1/|\mathcal{X}/\sim|$. If $d \geq |\mathcal{X}/\sim| - 1$ then any unit-normalized encoder that minimizes the $\sim$-ISSL log loss (Def. 13) is optimal for $\sim$-invariant tasks and linear probes $\mathcal{F}_{\text{lin}}$, i.e.,

$$\underset{\phi \in \Phi_1}{\arg\min} \, \mathcal{L}_I(\phi; p_X) \subseteq \Phi_* \tag{60}$$

*Proof.* From Lu and Steinerberger's [18] Theorem 1 (see also [80, Theorem 1] and [21, Section 3]) we know that if $d \geq |\mathcal{X}/\sim| - 1$ and equivalence classes are equiprobable then the global minimizer of the ISSL log loss over unit-normalized encoders and weights, will give weights and encoders such that for all $x \in [x]$ we have $\phi(x) = w(M(x))$, $\phi_\to(\mathcal{X})$ forms a simplex equiangular tight frame, and $|\phi_\to(\mathcal{X})| \geq |\mathcal{X}/\sim|$.

Clearly such representation is invariant as $\forall x \in [x]$ we have $\phi(x) = w(M(x))$. Furthermore, as the representations form an ETF in at least $|\mathcal{X}/\sim| - 1$ dimension they must span the entire $\mathbb{R}^{|\mathcal{X}/\sim|-1}$. From Theorem 1 we thus have that the global minimizers are sample-optimal for $\mathcal{F}_{\text{lin}}, \mathcal{T}$.

$\square$

For conciseness, we proved Prop. 3 by invoking previous results from the neural collapse literature. This is the reason we assumed equiprobable equivalence classes. Such assumption is nevertheless not necessary for learning sample-optimal encoders (rather than ETFs). A full proof can easily be shown by using Jensen's inequality similarly to our derivation of CISSL at Eq. (74).

Note that some norm regularization, constraint, or inductive bias is necessary for Prop. 3. Indeed, if this is not the case, then the ISSL log loss is minimized only if the representation's norm and/or the weight's norm tends to infinity. For simplicity, we use the stringiest constraint of having a fixed norm (unit-norm here). This can be extended to more realistic settings. For example: Fang et al. [80] assumes a bounded norm; Zhu et al. [22] uses a norm regularizer, akin to weight-decay, instead of a constraint; Ji et al. [23] removes the need for normalization by instead relying on the implicit bias of SGD / gradient flow.

## B.5 Non-linear optimality (Sec. 5)

In this section, we generalize Theorem 1 to non-linear $\mathcal{F}$ that satisfy Assmp. 5. Just as in the linear case, we start from Lemma 17 and then use classical results from statistical learning to rewrite the classical shatterability requirement into the predictability of $M(X)$ and a statement about dimensionality requirement. In the linear case, we relied on the fact that any $d - 1$ points in $\mathbb{R}^d$ can be linearly shattered if they span the entire space. Such a necessary and sufficient dimensionality does not always exist. In general, the necessary dimensionality is by definition given by the VC dimension [13] of $\mathcal{F}$, while the sufficient dimension is also (essentially) by definition given by generalizations of Cover's [39] capacity of $\mathcal{F}$. Where Cover's capacity is defined as the number of general position points that can be shattered by $\mathcal{F}$. We will thus have to give two statements one for necessity and one for sufficiency.

First, let us provide the necessary requirements for sample optimality, including a tight requirement on dimensionality.

**Proposition 19** (Necessity of sample-optimal $\phi^*$ for general probes). Let $\mathcal{T}$ be all invariant tasks w.r.t. $(\mathcal{X}, \sim)$, and $\mathcal{F}$ be any set of predictors for $\mathcal{T}$ that satisfies Assmp. 5. Any sample optimal encoder $\phi^* : \mathcal{X} \to \mathbb{R}^d$ for $\mathcal{T}, \mathcal{F}$ satisfy the following requirement:

- **Dimensionality**: the dimensionality of representation space is such that the VC dimension of $\mathcal{F}$ is at least the number of equivalence classes, i.e.,

$$d \text{ s.t. } \text{VC}[\mathcal{F}] \geq |\mathcal{X}/\sim| \tag{61}$$

- $\mathcal{F}$**-predictability of** $M$: there exists a maximal invariant $M : \mathcal{X} \to \{0, \ldots, |\mathcal{X}/\sim| - 1\}$ w.r.t. $(\mathcal{X}, \sim)$ that is predictable by $\mathcal{F}_{\text{lin}}$ from $\phi^*$, i.e.,

$$\exists M, f \in \mathcal{F} : \quad \forall x \in \mathcal{X} : \quad M(x) = \text{pred}(f(\phi^*(x))). \tag{62}$$

- **Invariance**: the encoder $\phi^*$ is invariant w.r.t. $(\mathcal{X}, \sim)$, i.e.,

$$\forall x, x^+ \in \mathcal{X}: \quad x \sim x^+ \implies \phi^*(x) = \phi^*(x^+) \tag{63}$$

Furthermore, the dimensionality requirement is tight in that there exists a sample-optimal encoder for $\mathcal{T}, \mathcal{F}$ whose dimensionality is such that $\mathrm{VC}[\mathcal{F}] = |\mathcal{X}/\sim|$

*Proof.* From Lemma 17 we know that an encoder $\phi^*$ is sample optimal for $\mathcal{T}, \mathcal{F}$ if and only if it is invariant w.r.t. $(\mathcal{X}, \sim)$, $\phi^*_\rightarrow(\mathcal{X})$ is classically shattered by $\mathcal{F}$, and $|\phi^*_\rightarrow(\mathcal{X})| = |\mathcal{X}/\sim|$. By definition the maximal number of points that can be classically shattered by $\mathcal{F}$ is the VC dimension of $\mathcal{F}$ [13]. We thus have that an encoder $\phi^*$ is sample optimal for $\mathcal{T}, \mathcal{F}$ implies that $\mathrm{VC}[\mathcal{F}] \geq |\mathcal{X}/\sim|$. As the VC dimension is generally a function of the ambient dimension $d$ we have that the dimensionality needs to be such that $\mathrm{VC}[\mathcal{F}] \geq |\mathcal{X}/\sim|$ as stated in Eq. (61).

Necessity of invariance comes directly from Lemma 17. The necessity of $\mathcal{F}$-predictability of $M$ comes from the fact that any maximal invariant $M : \mathcal{X} \rightarrow \{0, \ldots, |\mathcal{X}/\sim| - 1\}$ induces a possible invariant task and thus has to be predictable. $\square$

Prop. 19 provides a tight requirement for dimensionality, but in general, not all encoders that satisfy those requirements will be sample optimal. Let us now give sufficiency requirements on the dimensionality. To give non-trivial sufficiency statements we will restrict ourselves to encoders that induce representations that are in a general linear position, i.e., non-degenerate. Such general position encoders will essentially be learned almost surely,[6] and avoid non-interesting counterexamples [39]. Using the example and definition 40.1 from MacKay [81] we have that a set of points $\{x_i\}$ is in general position in $d$-dimensional space iff any subset of size $\leq d$ is linearly independent, and no $d + 1$ of them lie in a (K-1)-dimensional plane. This formalizes the intuition of "random" points in the space in terms of linear dependence. For example, you do not expect points in three dimensions to lie on a straight line. Note that the general position of representations is only used for the sufficiency of the following proposition rather than necessity.

In the case of necessity, we measured the complexity of $\mathcal{F}$ using the VC dimension, which by definition is the maximum number of points that $\mathcal{F}$ can shatter. For sufficiency we will use another complexity measure that appeared under many names, e.g., *dense $\pm$-shattering dimension* [82] or $\mu$-dimension [83] and more generally is studied without specific name [84–87]. This complexity measure is defined as the maximum number $N$ such that any set of $N$ points in general position can be shattered by $\mathcal{F}$. The most related well-known complexity measure is Cover's [88] capacity, which is the maximum number such that $N$ such that half of the dichotomies on any set of $N$ points in general position can be predicted by $\mathcal{F}$. In the following, we thus call the desired complexity measure: Cover's 1-capacity (instead of the standard 0.5-capacity ). We denote it as $\mathrm{Cap}_1[\mathcal{F}]$.

**Proposition 20** (Sufficiency of sample-optimal $\phi^*$ for general probes)**.** Let $\mathcal{T}$ be all invariant tasks w.r.t. $(\mathcal{X}, \sim)$, and $\mathcal{F}$ be any set of predictors for $\mathcal{T}$ that satisfies Assmp. 5. Let $\phi^* : \mathcal{X} \rightarrow \mathbb{R}^d$ be any encoder that induces representations $\phi_\rightarrow(\mathcal{X})$ in general position and satisfies all requirements from Prop. 19. Then any $\phi^*$ is sample optimal for $\mathcal{T}, \mathcal{F}$ if the dimension $d$ is such that Cover's 1-capacity of $\mathcal{F}$ is at least the number of equivalence classes, $\mathrm{Cap}_1[\mathcal{F}] = |\mathcal{X}/\sim|$.

*Proof.* Suppose that all requirements from Prop. 19 are satisfied. Then $\phi^*$ is invariant and clearly $|\phi^*_\rightarrow(\mathcal{X})| = |\mathcal{X}/\sim|$ due to $\mathcal{F}$ predictability of $M$. By Lemma 17 we know that it suffices for $\phi^*_\rightarrow(\mathcal{X})$ to be classically shattered by $\mathcal{F}$ to ensure that $\phi^*$ is sample optimal for $\mathcal{F}, \mathcal{T}$. We also know by assumption that $\phi^*_\rightarrow(\mathcal{X})$ lie in general position. By definition $\phi^*_\rightarrow(\mathcal{X})$ can be shattered by $\mathcal{F}$ if $\mathrm{Cap}_1\mathcal{F} \geq |\mathcal{X}/\sim|$, which concludes the proof. $\square$

Note that the VC dimension is by definition an upper bound on Cover's capacity. So putting together both Props. 19 and 20 we essentially have that $\mathcal{F}$ predictability of $M(X)$ and invariance are necessary and that there is a tight necessary dimensionality $d_{\mathrm{nec}}(\mathcal{F})$ and a sufficient dimensionality $d_{\mathrm{suff}}(\mathcal{F}) \geq d_{\mathrm{nec}}(\mathcal{F})$, which respectively depend on the VC dimension and the capacity of $\mathcal{F}$. Using classical statistical learning theory results:

---

[6]The probability that we would learn such encoders would be 1 if we worked in continuous spaces. In arbitrary large finite spaces, the probability can be still arbitrarily close to 1.

**Linear** for linear probes $\mathcal{F}_{\text{lin}}$ we have $d_{\text{suff}}(\mathcal{F}_{\text{lin}}) = d_{\text{nec}}(\mathcal{F}_{\text{lin}}) = |\mathcal{X}/\sim| - 1$ as in Theorem 1.

**Universal** for unconstrained probes $\mathcal{Q}_{\text{univ}}$ we have $d_{\text{suff}}(\mathcal{Q}_{\text{univ}}) = d_{\text{nec}}(\mathcal{Q}_{\text{univ}}) = 1$.

**MLP** for MLP probes $\mathcal{F}_{\text{mlp}}$ we generally have $d_{\text{suff}}(\mathcal{F}_{\text{mlp}}) \leq d_{\text{nec}}(\mathcal{F}_{\text{mlp}})$ both of which depend on the number of parameters, layers, width, and activations of the MLP. For VC dimensions MLPs refer to [89, 90]. For Cover's 1-capacity MLPs and related quantities refer to [82, 84–88, 91–93].

**Monotonicity** increasing the functional family cannot increase the dimensionality requirements, i.e., for any $\mathcal{F}^{-} \subseteq \mathcal{F}^{+}$ we have $d_{\text{suff}}(\mathcal{F}^{+}) \leq d_{\text{suff}}(\mathcal{F}^{-})$ and $d_{\text{nec}}(\mathcal{F}^{+}) \leq d_{\text{nec}}(\mathcal{F}^{-})$. This comes directly from the definition of VC dimension and capacity which are monotonic.

## C  Practical ISSL objectives

In this section, we derive our objectives to approximate the ISSL log loss and provide a minimal practical implementation of both objectives. In contrast to the proofs of main theoretical results (in Appx. B) derivations will be less formal. We focus on the case of linear probes $\mathcal{F}_{\text{lin}}$ for simplicity. For simplicity, we assume throughout that the equivalence classes are equiprobable $p_X([x]) = \frac{1}{|\mathcal{X}/\sim|}$ although our claims should easily generalize to any distribution with non-zero support on equivalence classes.

Recall the ISSL log loss that we want to minimize by Prop. 3 is:

$$\mathcal{L}_I(\phi; p_X) := \inf_{w \in \mathcal{W}_1} \mathbb{E}_{p_X} \left[ -\log \frac{\exp\big(w(M(X))^\top \phi(X)\big)}{\sum_{m'=1}^{|\mathcal{X}/\sim|} \exp(w(m')^\top \phi(X))} \right], \tag{64}$$

The main difficulty is approximating the ISSL log loss using samples from augmentations $A(\tilde{X}|X)$ instead of knowing $M(X)$ or $|\mathcal{X}/\sim|$.

### C.1  Deriving contrastive ISSL (Sec. 4.1)

The ISSL log loss and the CISSL loss are equivalent in that the encoders that minimize them are the same (even though the value of the losses are different). The key result that we rely on is that Ma and Collins [31] showed that for any number of negatives $k \geq 1$ the ranking-based variant [30, 31] of noise contrastive estimations (NCE; [29]) gives consistent parameter estimates under weak assumptions. For conciseness let us denote by $\tilde{\mathbf{X}} := \{\tilde{X}^+, \tilde{X}_1^-, \ldots, \tilde{X}_k^-\}$ a sequence of augmented inputs where the positive $\tilde{X}_0 = \tilde{X}^+$ is sampled from the conditional $A(\tilde{X} \mid X)$, while the $k$ negatives $\tilde{X}_i^-$ come from the marginal $A(\tilde{X}) = \mathbb{E}_{p_X}[A(\tilde{X}|X)]$. We use $p(\tilde{\mathbf{X}}|X; A)$ to denote the distribution of such sequence of random variables. Let us also denote by $\mathcal{G}_1 := \{g : \mathcal{X} \to \mathcal{S}\}$ all functions from $\mathcal{X}$ to unit-normalized outputs in $\mathbb{R}^d$. Let us also denote $X$ by $X_0$. Finally, we assume that any equivalent inputs have the same augmentation distribution $x \sim x^+ \iff A(\tilde{X}|x) = A(\tilde{X}|x^+)$ and $\tilde{X} \sim x^+$. Note that this is the standard conditional independence assumption $\tilde{X} - M(X) - X$ (e.g. [3, 12]). As a result, we have:

$$\underset{\phi \in \Phi_1}{\arg\min} \, \mathcal{L}_I(\phi; p_X) \tag{65}$$

$$= \underset{\phi \in \Phi_1}{\arg\min} \, \inf_{w \in \mathcal{W}_1} \mathbb{E}_{p_X} \left[ -\log \frac{\exp\big(w(M(X))^\top \phi(X)\big)}{\sum_{m'=1}^{|\mathcal{X}/\sim|} \exp(w(m')^\top \phi(X))} \right] \qquad \text{def} \tag{66}$$

$$= \underset{\phi \in \Phi_1}{\arg\min} \, \inf_{w \in \mathcal{W}_1} \mathbb{E}_{p_X(\{X_i\}_k)} \left[ -\log \frac{\exp\big(w(M(X))^\top \phi(X)\big)}{\sum_{i=0}^{k} \exp(w(M(X_i))^\top \phi(X))} \right] \qquad \text{cons. NCE} \tag{67}$$

$$= \underset{\phi \in \Phi_1}{\arg\min} \, \inf_{w \in \mathcal{W}_1} \mathbb{E}_{p_X p(\tilde{\mathbf{X}}|X; A)} \left[ -\log \frac{\exp\big(w(M(\tilde{X}^+))^\top \phi(X)\big)}{\sum_{i=0}^{k} \exp\big(w(M(\tilde{X}_i))^\top \phi(X)\big)} \right] \qquad M \text{ Inv.} \tag{68}$$

$$= \underset{\phi \in \Phi_1}{\arg\min} \, \inf_{g \in \mathcal{G}_1} \mathbb{E}_{p_X p(\tilde{\mathbf{X}}|X; A)} \left[ -\log \frac{\exp\big(g(\tilde{X}^+)^\top \phi(X)\big)}{\sum_{i=0}^{k} \exp\big(g(\tilde{X}_i)^\top \phi(X)\big)} \right] \qquad \text{DPI} \tag{69}$$

where Eq. (67) uses the consistency of ranking-based NCE and Eq. (68) uses the invariance of the maximal invariant and the fact that the augmentation preserves the equivalence structure $\tilde{X} \sim x^+$. Eq. (69) uses the fact that when the Markov Chain $\tilde{X} - M(X) - X$ is satisfied, we can replace $w(M(X))$ by $g(x))$ essentially by the data processing inequality of the Bayes risk [12, 94] of convex loss functions. To see that, we clearly have the conditional independence $g(\tilde{X}) \perp\!\!\!\perp X | M(X)$ due to $\tilde{X} - M(X) - X$. Now recall that one characterization of conditional independence (CI) is that $\tilde{X} \perp\!\!\!\perp X \mid M(X)$ if and only if $\tilde{X} = g'(M(X), U)$ almost surely for some function $g'$ and $U \stackrel{d}{\sim} \text{Unif}(0, 1)$ with $U \perp\!\!\!\perp (M(X), X)$ [95, Prop. 6.13]. In the following we use $\mathbf{X} := \text{cat}(\{X_i\}_k)$, $\mathbf{U} := \text{cat}(\{U_i\}_k)$, and all functions applies to a vector to denote element-wise functions. We thus

have:

$$\inf_{g \in \mathcal{G}_1} \mathbb{E}_{p(\tilde{\mathbf{X}}|X;A)}\left[-\log \frac{\exp\left(g(\tilde{X}^+)^\top \phi(X)\right)}{\sum_{i=1}^{k} \exp\left(g(\tilde{X}_i)^\top \phi(X)\right)}\right] \tag{70}$$

$$= \inf_{g' \in \mathcal{G}_1} \mathbb{E}_{p_X(\mathbf{X})p(\mathbf{U})}\left[-\log \frac{\exp\left(g'(M(X),U)^\top \phi(X)\right)}{\sum_{i=1}^{k} \exp(g'(M(X_i),U_i)^\top \phi(X))}\right] \quad \text{CI} \tag{71}$$

$$= \inf_{g' \in \mathcal{G}_1} \mathbb{E}_{p_X(\mathbf{X})p(\mathbf{U})}\left[\log\left(1 + \sum_{i=1}^{k} \exp\left((g'(M(X_i),U_i) - g'(M(X),U))^\top \phi(X)\right)\right)\right] \tag{72}$$

$$= \inf_{g' \in \mathcal{G}_1} \mathbb{E}_{p_X(\mathbf{X})p(\mathbf{U})}\left[\log\left(1 + \mathbf{1}^\top \exp\left((g'(M(\mathbf{X}),\mathbf{U}) - g'(M(X),U))^\top \phi(X)\right)\right)\right] \tag{73}$$

$$\geq \inf_{g' \in \mathcal{G}_1} \mathbb{E}_{p_X(\mathbf{X})}\left[\log\left(1 + \mathbf{1}^\top \exp\left(\mathbb{E}_{p(\mathbf{U})}[g'(M(\mathbf{X}),\mathbf{U}) - g'(M(X),U)]^\top \phi(X)\right)\right)\right] \quad \text{Jen.} \tag{74}$$

where the last line uses the fact that $g'(M(\mathbf{X}),\mathbf{U}) - g'(M(X),U)$ is necessarily negative (there exists a $g'$ such that all labels classified correctly), $(\log(1 + \exp(-x)))$ is strictly convex, that $U$ is independent of $X$. Jensen's inequality is tight for strictly convex functions if and only if the function is constant. We thus have that $g'(M(\mathbf{X}),\mathbf{U}) - g'(M(X),U)$ must be a constant for all $\mathbf{U}$ from which we conclude that $g(\tilde{X})$ must be independent of $U$ and so there exists $g'$ s.t. $g = g' \circ M$ (i.e. $g$ will be invariant). Letting $w = g'$ we recover Eq. (68) as desired.

## C.2 Deriving distillation ISSL (Sec. 4.2)

By simply rearranging terms in the ISSL log loss we get (differences are in red) that the ISSL log loss is equal to

$$\inf_{w \in \mathcal{W}_1} \mathbb{E}_{p_X q(\hat{M}\,|\,X)}\left[-\log s_{\phi,w}(\hat{M}\,|\,X)\right], \quad s_{\phi,w}(m\,|\,x) = \frac{\exp\left(w(m)^\top \phi(x)\right)}{\sum_{m=1}^{C} \exp(w(m)^\top \phi(x))}, \tag{75}$$

if $C = |\mathcal{X}/\sim|$ and some maximal invariant r.v. is distributed as $M(X) \overset{d}{\sim} q(\hat{M}\,|\,X)$. The teacher $q(\hat{M}\,|\,X)$ and the student $s_{\phi,w}(\hat{M}\,|\,X)$ are both categorical distributions over $C$ categories. By Def. 4 of the maximal invariant, we thus have that Eq. (75) is exactly the ISSL log loss if and only if:

**Deterministic** the teacher is a deterministic distribution $\max_{m \in \{1,\dots,C\}} q(m\,|\,X) = 1$;
**Invariant** the teacher maps positives together $x \sim x^+ \implies q(\hat{M}\,|\,x) = q(\hat{M}\,|\,x)$;
**Maximal** the teacher maps negatives separately $x \not\sim x^- \implies q(\hat{M}\,|\,x) \neq q(\hat{M}\,|\,x^-)$.

Using information-theoretical quantities we have the following equivalent requirements

**Deterministic** the teacher is deterministic iff it minimizes the conditional entropy $\mathrm{H}[\hat{M}\,|\,X] = 0$;
**Invariant** the teacher is invariant iff the KL divergence between its outputs on equivalent examples is minimal $\mathbb{D}_{\mathrm{KL}}[q(\hat{M}\,|\,x)\|q(\hat{M}\,|\,x^+)] = 0$ for all $x \sim x^+$;
**Maximal** an invariant and deterministic teacher is maximal if and only if the KL divergence between its marginal $q(\hat{M}) = \mathbb{E}_{p_X}[q(\hat{M}|X)]$ and the true one is minimized $\mathbb{D}_{\mathrm{KL}}[q(\hat{M})\|p(M(X))] = 0$.

Determinism and invariance are trivial to prove by standard properties of entropy and KL divergence. It is also easy to show by contrapositive that $q(\hat{M}) = p(M(X))$ implies maximality for a deterministic and invariant teacher (the other direction is trivial when $C = |\mathcal{X}/\sim|$). Suppose that the marginals were matched but some non-equivalent examples were matched together, i.e., $\exists x \not\sim x^-$ s.t. $q(\hat{M}\,|\,x) = q(\hat{M}\,|\,x^-)$. Then by invariance of the teacher, it means that the outputs of all the examples in two equivalence classes would be mapped together. By determinism of the teacher, this means that the support of the marginal $q(\hat{M})$ would have to be on less than $C = |\mathcal{X}/\sim|$ examples. Due to the premise $q(\hat{M}) = p(M(X))$ we would have that $p(M(X))$ is supported on less than

$|\mathcal{X}/{\sim}|$ examples which contradicts the maximal invariance of $M$ or the fact that $p_X$ is supported over all equivalence classes and concludes the proof.

Now note that the cross-entropy is equal to the KL divergence plus the entropy

$$\mathbb{D}_{\mathrm{KL}}\Big[q(\hat{M}\,|\,x)\Big\|q(\hat{M}\,|\,x^+)\Big] = \mathbb{E}_{q(\hat{M}\,|\,x)}\left[\log\frac{q(\hat{M}\,|\,x)}{q(\hat{M}\,|\,x^+)}\right] \tag{76}$$

$$= -\,\mathbb{H}\Big[\hat{M}\,|\,x\Big] + \mathbb{E}_{q(\hat{M}\,|\,x)}\Big[-\log q(\hat{M}\,|\,x^+)\Big] \tag{77}$$

$$\mathbb{D}_{\mathrm{KL}}\Big[q(\hat{M}\,|\,x)\Big\|q(\hat{M}\,|\,x^+)\Big] + \mathbb{H}\Big[\hat{M}\,|\,x\Big] = \mathbb{E}_{q(\hat{M}\,|\,x)}\Big[-\log q(\hat{M}\,|\,x^+)\Big]. \tag{78}$$

As all those information-theoretic terms are positive for discrete (categorical) r.v., we have that the KL and the entropy are 0 if and only if the cross-entropy is zero. In other words, a teacher is deterministic and invariant if and only if the cross-entropy of its outputs on equivalent examples is zero. Now assume that the augmentations $\mathrm{A}(\tilde{X}|x)$ are such that for all $x \in \mathcal{X}$ we have $\mathrm{supp}(\mathrm{A}(\tilde{X}|x)) = [x]$. Then the minimization of the cross-entropy for each example can be written as an expectation: $\mathbb{E}_{p_X \mathrm{A}(\tilde{X}|X)q(\hat{M}\,|\,X)}\Big[-\log q(\hat{M}\,|\,\tilde{X})\Big] = 0$. By transitivity arguments, it is easy to show that the same holds as long as there is a path through images and preimages of augmentations that can map any example to another equivalent examples, i.e., as long as the equivalence classes are the connected component of the augmentation graph [8].

Putting it all together, the ISSL log loss can be written as:

$$\mathcal{L}_I(\phi; p_X) = \inf_{w \in \mathcal{W}_1} \mathbb{E}_{p_X q(\hat{M}\,|\,X)}\Big[-\log s_{\phi,w}(\hat{M}\,|\,X)\Big] \tag{79}$$

$$\text{s.t.}\quad \mathbb{E}_{p_X \mathrm{A}(\tilde{X}^+|X)q(\hat{M}\,|\,X)}\Big[-\log q(\hat{M}\,|\,\tilde{X}^+)\Big] = 0 \tag{80}$$

$$\mathbb{D}_{\mathrm{KL}}\Big[q(\hat{M})\Big\|p(M(X))\Big] = 0 \tag{81}$$

Using a Lagrangian relaxation and joint training of the weight and teacher (over unconstrained predictors) we get our DISSL objective:

$$\mathcal{L}_D(\phi; p_X) = \inf_{w \in \mathcal{W}_1, q} \lambda \mathbb{D}_{\mathrm{KL}}\Big[q(\hat{M})\Big\|p(M(X))\Big] \tag{82}$$

$$- \mathbb{E}_{p_X \mathrm{A}(\tilde{X}^+|X)q(\hat{M}\,|\,X)}\Big[\beta \log q(\hat{M}\,|\,\tilde{X}^+) + \log s_{\phi,w}(\hat{M}\,|\,X)\Big] \tag{83}$$

Note that because the augmentations are label-preserving and the teacher will be forced to be invariant we can also augment the input $X$ to the teacher $q(\hat{M}\,|\,X)$ and/or the student $s_{\phi,w}(\hat{M}\,|\,X)$.

Finally, by using Monte-Carlo estimates from a dataset $\mathcal{D}$ sampled from $p_X^n$ we get our empirical DISSL objective from the paper $\hat{\mathcal{L}}_D(\phi; \mathcal{D}) :=$

$$\inf_{w \in \mathcal{W}_1, q} \lambda \mathbb{D}_{\mathrm{KL}}\Big[q(\hat{M})\Big\|p(M(X))\Big] - \sum_{x \in \mathcal{D}} \mathbb{E}_{\mathrm{A}(\tilde{X}|X)q(\hat{M}\,|\,x)}\Big[\beta \log q(\hat{M}\,|\,\tilde{X}) + \log s_{\phi,w}(\hat{M}\,|\,x)\Big] \tag{84}$$

Using the strong law of large numbers, we have that as $|\mathcal{D}| \to \infty$ the empirical $\hat{\mathcal{L}}_D(\phi; \mathcal{D})$ becomes almost surely equal to $\mathcal{L}_D(\phi; p_X)$, which is equal to Eq. (79) when $q$ is optimized in a universal variational family and $\lambda, \beta \to \infty$. As we have seen, Eq. (79) is equal to the ISSL log loss when the marginal $p(M(X))$ is known and the equivalence classes are the connected component of the augmentation graph. Using Prop. 3 we thus conclude that DISSL learns optimal encoders in idealized settings.

### C.3 Minimal PyTorch implementation

In the following, we provide a minimal practical implementation of both of our objectives. For the actual code we used see `github.com/YannDubs/Invariant-Self-Supervised-Learning`.

#### C.3.1 CISSL

Source Code 1 show a minimal practical (batch) implementation of CISSL. For a version with minimal dependencies/training/evaluation see this self-contained notebook.

```
import torch.nn as nn
import torch.nn.functional as F

class CISSL(nn.Module):
    def __init__(self, proj_dim=128):
        super().__init__()
        self.encoder = resnet() # to define
        # teacher projection should be as expressive as possible
        self.teacher_proj = MLP(z_dim, proj_dim) # to define
        # student projection should be linear (note: BN is linear)
        self.student_proj = nn.Sequential(nn.Linear(z_dim, proj_dim),
                                          nn.BatchNorm1d(proj_dim))

    def loss(self, x1, x2, temp=0.07):
        x1, x2 = batch
        bs, device = x1.size(0), x1.device

        # logits shape: [2*bs, 2*bs]. Normalizes for cosine sim.
        z = self.encoder(torch.cat([x1, x2], dim=0))
        z_student = F.normalize(self.predictor(z), dim=1, p=2)
        z_teacher = F.normalize(self.projector(z), dim=1, p=2)
        logits = z_student @ z_teacher.T / self.temp

        # there are two positives for each example x1: x1 and x2
        # note: SimCLR removes x1-x1 as those are typically equal.
        # But not for CISSL due to asymmetric proj heads =>
        # CE between predicted proba and 0.5 for each positive
        log_q = logits.log_softmax(-1)
        select_pos = torch.eye(bs, device=device).bool().repeat(2, 2)
        CE = - log_q[select_pos].view(bs*2, 2).sum(1) / 2
        return CE.mean()
```

Source Code 1: Minimal PyTorch for CISSL

Compared to SimCLR we see two differences:

**Asymmetric projection heads** One of the two projections head (`self.teacher_proj`) is an MLP just as in SimCLR. The other projection head (`self.student_proj`) has the same architecture as downstream probes (here linear[7]). This ensures that downstream probes will be able to extract the desired information.

**Self-contrastive** In SimCLR the current augmented example is contrasted with all the other augmented examples in a batch except itself. Indeed, if projection heads are symmetric then the same augmented example would have the same projected output on both branches and so we can discard it as a positive. For CISSL this is not the case as projection heads are asymmetric. As a result instead of having a single positive example for every example, we now have two of them (both augmented versions of the example). The loss is then the cross-entropy between the predicted probability of both of those examples and a categorical distribution where each positive has a probability $0.5$. Using this "self-contrasting" is simpler/shorter to implement and works slightly better ($\approx 0.5\%$ accuracy gains on TinyImageNet)

### C.3.2 DISSL

Source Code 2 shows a minimal practical (batch) implementation of DISSL. Compared to other non-contrastive methods (e.g. SwAV or DINO) we see that DISSL is very simple to implement and understand. In particular, there are no stop-gradients, momentum encoders, or complicated internal algorithms (e.g. Sinkhorn-Knopp in SwAV). For a version with minimal dependencies/training/evaluation see this self-contained notebook.

---

[7]Note that a batch normalization is linear and so a linear layer followed by a batch normalization is also linear. We use batch normalization to be more consistent with SimCLR and found that this also improves performance.

```python
from torch.distributions import Categorical
import torch.nn as nn

class DISSL(nn.Module):
    def __init__(self, n_equiv=16384, zdim=512):
        super().__init__()
        self.encoder = resnet() # to define
        # teacher projection should be as expressive as possible
        self.teacher_proj = MLP(z_dim, n_equiv)
        # student projection should be same architeture as probe
        self.student_proj = nn.Linear(z_dim, n_equiv)

    def loss(self, x1, x2):
        z1, z2 = self.encoder(x1), self.encoder(x2)
        return (self.asym_loss(z1,z2) + self.asym_loss(z2,z1)) / 2

    def asym_loss(self, z1, z2, lambd=2.3, beta=0.8, temp=0.5):
        logits_t1 = self.teacher_proj(z1) / temp
        logits_t2 = self.teacher_proj(z2) / temp
        logits_s = self.student_proj(z2)
        q_Mlx = Categorical(logits=logits_t1) # q(\hat{M}|X)

        # MAXIMALITY. -H[\hat{M}]
        mxml = -Categorical(probs=q_Mlx.probs.mean(0)).entropy()

        # INVARIANCE and DETERMINISM. E_{q(M|X)}[log q(M|\tilde{X})]
        det_inv = (q_Mlx.probs * logits_t2.log_softmax(-1)).sum(-1)

        # DISTILLATION. E_{q(M|X)}[log s(M|\tilde{X})]
        dstl = (q_Mlx.probs * logits_s.log_softmax(-1)).sum(-1)

        return lambd * mxml - beta * det_inv.mean() - dstl.mean()
```

Source Code 2: Minimal PyTorch code for DISSL

# D  Relation to previous work

## D.1  Related work

**Learning theory and self-supervised learning.** There have been many recent works that aim to explain why *specific SSL algorithm* work by *bounding* the performance of downstream linear probes i.e., proving that specific algorithms are not too bad in practice. Saunshi et al. [3] (extended in [54, 96–100]) and Tosh et al. [6] bound downstream performance for contrastive learning using an approximate conditional independence assumptions similar to the one we use for CISSL. Lee et al. [5] provides similar guarantees for a reconstruction pretext task. Bansal et al. [4], provided guarantees for a wider range of SSL algorithms by assuming small rationality and robustness gaps rather than through statistical assumptions. Other works have also tried incorporating the optimization of neural networks in SSL theory [7, 101]. All these works differ from our theory in that they start from existing algorithms and are thus mostly descriptive rather than prescriptive. A notable exception is HaoChen et al. [8] which introduces the concept of augmentation graph and then proposes a simple SSL algorithm motivated by spectral decomposition of that graph. They provide downstream guarantees and show experimentally that their methods match standard SSL baselines. Their theory (and others) can be seen as providing *sufficient* conditions for achieving good downstream performance, while our theory gives *sufficient and necessary* conditions for achieving perfect performance. The advantage of HaoChen et al. [8] theory is that by analyzing their specific algorithm, they provide practical guarantees and use less stringent assumptions. The advantage of our theory is that by giving necessary conditions and working at the representation level (agnostic to the algorithm) we can give a unifying framework for SSL that suggests common improvements and can be used to derive future SSL SSL algorithms. Our framework's prescriptions also seem more useful in practice as we outperform all baselines.

**Optimal encoders and idealized representations.** In contrast to standard theoretical work, we start from our ideal requirements, then characterize all optimal encoders that satisfy those requirements, and finally derive practical algorithms and actionable insights from this idealized framework. This approach is inspired by recent work on idealized representations for supervised learning [62] and domain adaptation [63]. One advantage of dealing directly with the properties of the representations is that we can abstract away the encoder's architecture and training algorithm. This is similar to the recent "layer-peeled" [23, 80] and "unconstrained feature" [22, 102] approach to neural collapse where features are modeled as free optimization variables.

**Properties of self-supervised representations.** Wang and Isola [103] (and follow-ups, e.g., [104]) also work with properties of the representations rather than the algorithms. Specifically, they show that contrastive learning forces the positive representations to be close (alignment) while all normalized representations will be uniformly distributed on a hypersphere. The difference with our work is that we start with the ideal requirements for downstream performance and use those to derive the characterization of optimal encoders. Instead, they start from the contrastive learning algorithm and analyze the resulting representations without giving any theoretical relation between those conditions and downstream performance. In fact, these two properties provably do not ensure good linear probing [57, 100]. This can be seen by Theorem 1, as uniformity needs lower-dimensional representations. We instead show that minimizing the ISSL log loss in higher dimensions will give optimal encoders that induce ETF representations (normalized and aligned but not uniform) which leads to better downstream performance.

**Our actionable insights.** Some of our prescriptions have been hinted at in previous work:

- *Effective dimensionality*: The need for a large dimension is indirectly suggested by Saunshi et al. [57] theory which shows that one can have bad downstream performance when the dimension is small even if the SSL loss is small (their goal is to show that one needs to incorporate inductive bias in SSL theory). HaoChen et al.'s [8] theoretical guarantees for their specific ISSL algorithm also require a large ambient dimension. For a fixed ambient dimensionality previous work [41, 42] also suggested that low effective dimensionality, dubbed *dimensional collapse*, can be an issue and provided solutions to alleviate this issue and improve performance (as proved by Theorem 1).
- *Augmentations*: Many prior work have suggested that a good augmentation or view is one that is information preserving while removing as much nuisance information as possible [12, 44–48]. Prop. 2 (and Appx. B.3) can be seen as a new perspective on why coarser augmentations are useful, by proving the exact relation between optimal sample efficiency and the number of

equivalences. An example of coarse label-preserving augmentations for standard classification are the text-image pairs from Radford et al. [15].

- *Asymmetric projections heads*: It is well known empirically that using large non-linear projection heads improves performance of downstream probes (e.g. [1, 25, 43]). To our knowledge, we are the first to theoretically prove and derive the need for projection heads. In particular, we show that we should only project one of the two representations and that this helps in practice. We are not aware of any previous empirical or theoretical work that uses such asymmetric heads (SimSiam uses asymmetric projection heads but still projects both sides with a non-linear mapping).

**Non-linear probes.** The standard evaluation of SSL representations uses linear probes [27, 37, 38]. However, if the goal is to maximize performance it is natural to consider non-linear predictors. For example, Dubois et al. [12] uses MLP probes. To our knowledge, we are the first to provide a theoretical SSL framework for non-linear probing.

**Understanding non-contrastive SSL.** Many recent work [55, 101, 105, 106] have studied how distilling SSL methods work and the importance of optimization tricks such as stop-gradients, exponential moving average, and normalizations. From our framework's perspective, those works explain how previous distillation methods enforce the maximality of the teacher. Using our formal requirement we provide a new objective derived from first principles, DISSL, that does not require any optimization tricks and performs better than previous non-contrastive methods.

**Invariances and augmentations.** Empirically, SSL have been shown to learn invariances from the data augmentations [12, 107], and downstream performance improves when using methods to increase invariance [108]. The standard way of modeling invariances to data augmentations is through group theory [10, 11]. Such a framework is nevertheless too constrained as standard augmentations are not group actions (e.g. cropping). Dubois et al. [12] proposed modeling invariances to data augmentations using the more general framework of equivalence relations, which we follow in our work. Mitrovic et al. [47] also used equivalence relations to formalize SSL using an invariant causal mechanism perspective. von Kügelgen et al. [109] takes a similar invariant causal mechanism perspective and analyses whether and when the invariant component, i.e. $M(X)$, is identifiable. One potential issue with an invariance perspective on SSL is that real augmentations might not be exactly label-preserving. In our framework we somewhat deal with this issue by only considering invariance of the most-like label $\arg\max_y p_t(y|X)$ rather than the entire distribution $p_t(Y|X)$ as in [12, 47]. Still, this might not perfectly hold. HaoChen et al. [8] instead uses a more general framework (augmentation graph) which can be seen as formalizing approximate invariances.

**Neural collapse.** Prop. 3 shows that the ISSL log loss is sufficient for optimality by using arguments from the neural collapse literature [16, 18, 22, 23]. Similar arguments were used by Galanti et al. [110] to try to explain transfer learning. Concurrently to our work, Awasthi et al. [99] also used similar arguments in SSL to explain why contrastive learning does not degrade with more negatives (contrary to Saunshi et al.'s [3] claims). A standard criticism (e.g. [111]) of neural collapse in supervised learning is that the phenomena seem to only hold on the training set. For SSL, we found in Fig. 16b that neural collapse happens also on the test set. This is most likely due to the fact that equivalent classes given by standard augmentations are much more fine-grained than those given by class labels.

**Our DISSL objective.** Our DISSL objective is most similar and can be seen as a simpler and theoretically-motivated version of DINO and SwAV. In particular, all those methods (and others, e.g., [49, 50]) can be seen as simultaneously training a teacher to perform online clustering and then distilling it into a student. Using the perspective and notation from our framework (to make the similarities more obvious) we have that:

- *DINO* [24]: also uses a categorical teacher $q(\hat{M}|X)$ that they distill in a categorical student $s_{\phi,w}(\hat{M}|X)$. The main difference in the student is that they use a non-linear projection head before the softmax, which as discussed in Sec. 4.1 does not ensure linear predictability of downstream tasks. For the teacher, DINO aims at ensuring maximality by setting the teacher to the exponential moving average of the student, stopping the gradients, and applying some centering. In contrast, DISSL does not require any optimization trick and enforces maximilaity by maximizing the entropy of the teacher's output (assuming uniform prior).

- *SwAV* [2]: also uses a categorical teacher $q(\hat{M}|X)$ that they distill in a categorical student $s_{\phi,w}(\hat{M}|X)$. Just as with DINO, SwAV's student uses a non-linear projection head and thus does

not ensure linear predictability of downstream tasks. For SwAV's teacher, they assume like us that the equivalent classes are equiprobable. But instead of performing essentially soft equiprobable clustering like us, they essentially perform hard equiprobable clustering using the Sinkhorn algorithm [52]. Specifically, they keep in memory a queue of previous examples, and at every step they essentially perform equiprobable hard clustering using 3 steps of the Sinkhorn algorithm. The advantages are that this clearly ensures maximality and determinism. The disadvantage is that their algorithm is significantly more complicated than ours, uses essentially hard constraints, requires storing a queue of previous examples, and requires stop-gradients.

### D.2 Taxonomy

Table 7: Taxonomy of previous models using our framework. 'Opt. if dim ↑' whether all trained encoders would be optimal in idealized settings if the dimension was large. 'Optimal' whether all trained encoders would be optimal in idealized settings if the dimension was large enough. "Opt. if asym." denotes whether an objective would be objective when using our asymmetric projection heads. If we can write down a solution to the objective that is not optimal we use "✗" if we can show the converse in idealized settings we use "✓", if we do not know we use "?". "Jensen" denotes the lower loss of deterministic outputs due to Jensen's inequality. "temp" denotes a temperature rescaling in a softmax. "stop" denotes stop-gradient. "pred" denotes the use of a prediction head in addition to the projector head. "decor" denotes decorrelation of different dimensions of the representations. "var" denotes increasing the marginal variance of the representation. "sinkhorn" denotes the use of Sinkhorn-Knopp algorithm for equiprobable clustering. "cluster" denotes forcing an essentially deterministic clustering. "transfer" denotes the use of a pretrained teacher. "optim?" denotes that optimization process (SGD and batchnorm) might play an important role but it is not clear yet. "ema" denotes momentum encoder. "ce" denotes cross-entropy loss. "mse" denotes mean squared error loss. "neg" denotes contrastive with negatives.

| | Determinism | Invariance | Maximality | Optimal | Opt. if asym. |
|---|---|---|---|---|---|
| SimSiam [25] | Jensen,temp | ce | stop,pred,optim? | ✗ | ✗ |
| DINO [24] | Jensen,temp | ce | stop,ema,center,optim? | ✗ | ✗ |
| BYOL [36] | Jensen,temp | ce | stop,ema,optim? | ✗ | ✗ |
| W-MSE [34] | Jensen | mse | whitening | ✗ | ? |
| Barlow T. [40] | Jensen | mse | decorr | ✗ | ? |
| VICReg [112] | Jensen | mse | decorr,var | ✗ | ? |
| SwAV [2] | cluster | ce | sinkhorn | ✗ | ✓ |
| SELA2 [2, 50] | cluster | ce | sinkhorn | ✗ | ✓ |
| DC2 [2, 49] | cluster | ce | stop,optim? | ✗ | ✓ |
| ClusterFit [113] | cluster | ce | transfer | ✗ | ✓ |
| SimCLR [1] | Jensen,temp | ce | neg | ✗ | ✓ |
| MOCO [35] | Jensen,temp | ce | neg | ✗ | ✓ |
| **CISSL** | Jensen,temp | ce | contr. | ✓ | ✓ |
| **DISSL** | $\min \mathbb{H}[M \mid Z]$ | ce | $\max \mathbb{H}[M]$ | ✓ | ✓ |

Table 7 provides a unifying perspective/taxonomy of some previous SSL algorithms from the perspective of our distillation ISSL framework. As a reminder, distillating the teacher into a student is equivalent to ISSL log loss, which gives give optimal encoders in idealized setting, if and only if:

**Deterministic** the teacher is a deterministic distribution $\max_{m \in \{1,...,C\}} q(m \mid X) = 1$;
**Invariant** the teacher maps positives together $x \sim x^+ \implies q(\hat{M} \mid x) = q(\hat{M} \mid x)$;
**Maximal** the teacher maps negatives separately $x \not\sim x^- \implies q(\hat{M} \mid x) \neq q(\hat{M} \mid x^-)$.

Note that we also provide three contrastive methods (SimCLR,MOCO,CISSL) as those can be seen as a specific instantiation of our distillation ISSL where maximality is enforced through negative examples and the denominator of the distillation loss is approximated using noise contrastive estimation.

As all recent methods use asymmetric heads, we have that none of them are optimal even in idealized settings. (see "optimal" column). So we also provide an "opt. if asym." column that shows whether the objectives would recover optimal encoders in the case where losses were we used asymmetric projection heads (and idealized assumptions). We see that all methods that are based on clustering

and contrastive learning would be optimal, but those that rely on optimization tricks are not. This suggests that understanding why such methods work requires analyzing the training dynamics as in [7, 101].

## D.3 Additional insights in relations of previous work

Using our framework we can also provide new insights or perspectives into common framing and questions about ISSL.

**SSL does not maximize Shannon's information but the $\mathcal{F}$-information $\mathbb{I}_{\mathcal{F}}[\phi(X) \to M(X)]$.** Many previous work have framed SSL as mutual information maximization between encoded views $\mathbb{I}\left[\phi(\tilde{X}); \phi(\tilde{X}^+)\right]$ or even with the input $\mathbb{I}[X; \phi(X)]$ [e.g. 27, 44, 46, 114–116]. Tschannen et al. [117] has shown empirically that the amount of mutual information is uncorrelated to downstream performance. This is best seen by the fact that due to the data processing inequality, any encoding of the input would necessarily give the worst representations in terms of information. Our framework shows two things: (1) if you had unconstrained downstream probes then you would want to maximize the information between the representation and the maximal invariant $\mathbb{I}[M(X); \phi(X)]$ as shown in [12]; and (2) in the case of constrained probes $\mathcal{F}$, e.g., linear, you instead want to minimize the risk when predicting $M(X)$ for $\phi(Z)$ which is equivalent to maximizing the (generalized [62]) $\mathcal{F}$-information $\mathbb{I}_{\mathcal{F}}[\phi(X) \to M(X)]$ [118]. $\mathcal{F}$-information is a generalization of Shannon's information that takes into account whether the information can be used by the desired predictors. For example $\mathcal{F}$-information does not satisfy the data processing inequality and can thus explain why a representation is more useful than raw inputs.

**Contrastive SSL should *only* project augmented representations.** It is well known large non-linear projection heads improves downstream performance compared to no projection heads (e.g. [1, 25, 43]). As a result, most SSL algorithms use projection heads on both views. Our work shows theoretically and empirically that one should apply the projection head asymmetrically.

**Non-contrastive learning should care about maximality rather than avoiding collapsing.** Most work [24, 25, 34, 36, 55, 101, 105, 106] in designing and understanding non-contrastive methods concerns how to avoid the "collapsing" of the teacher to a constant. Pokle et al. [119] empirically showed that there are many non-collapsing solutions that are just as bad. Indeed, intuitively there is nothing special about collapsing to a single constant solution. What if the teacher collapses to 2 possible constants, this is also intuitively bad. What about $3, 4, \ldots, k$ constants? To our knowledge, we are the first to formalize (and prove) exactly what is needed: maximality. I.e. no equivalent examples should be mapped together. This shows that the minimum number of representations to still ensure perfect downstream prediction is $|\mathcal{X}/\sim|$ which is much larger than a single constant.

**Larger encoder might help due to larger dimensionality rather than more parameters.** It is well known that larger encoders can improve downstream probing accuracy, which is typically attributed to the complexity/number of parameters of the encoder [e.g. 1, 2, 24, 25, 36]. The standard way of increasing parameters is by increasing the width of every bottleneck of a ResNet. In particular, this means that the dimensionality of the representation will also increase. The gains that we see from increasing dimensionality (Fig. 7c in the main paper) suggest that such a confounder might be important. Indeed, we show in Appx. G.4 that much of the gains when going from a ResNet18 to a ResNet50 are due to an increase in dimensionality (512 to 2048) rather than the number of parameters (11M to 23M). Practically, this suggests that we might be able to get nearly as much gains from increasing the dimensionality of the representations rather than training prohibitively large models.

# E Limitations

In the main paper, we briefly mentioned important limitations of our current framework. Here we discuss them in more detail.

**Need for approximate optimality.** First, we only consider optimal encoders which will never be exactly achieved in practice, even in the simpler settings. As a result, we cannot currently give any theoretical guarantees on encoders learned in practice. A more useful notion would be some $\epsilon$-approximate optimality which quantifies how far an encoder is from optimality and provide practical guarantees based on that $\epsilon$. The resulting framework would likely enable us to give more fine-grained insights into some of the requirements, e.g., quantify theoretically how much one gains by increasing the dimensionality by a certain amount (currently we have a minimum and sufficient dimensionality for optimality which is a binary statement). Such extension would help bridge our current framework to more standard statistical learning theory perspectives. It is nevertheless not clear whether such extension would be most interesting in theory or whether it would provide additional actionable insights.

**Need for constrained encoders and finite ISSL data.** The main theoretical simplification is that we analyze optimal encoders without modeling the constrained or inductive bias from realistic functional family and optimization schemes. This is a major simplification, which allows us to study the form of the representations without really considering how they are learned. As a result, our current simplified framework provides no insights into how to improve the computational or data efficiency of the ISSL pretraining. We believe that incorporating such constraints could provide many new actionable insights into the designing of ISSL algorithms.

**Need for considering meaningful tasks.** In our work, we consider all possible invariant tasks $\mathcal{T}$. In reality, only a subset of those tasks is meaningful and likely of interest. This subset of meaningful tasks cannot only be defined using augmentations. A potential solution to model this subset would be to incorporate the inductive bias of the model. This might yield interesting actionable insights by relating the downstream tasks with the choice of encoder's architecture or training algorithm.

**What are idealized representations?** The main starting point of our work is the definition of optimal encoders: the population and sample optimal ones. An obvious limitation is that not everyone might agree on that goal, in particular the sample-optimal one. For example, one could want to consider average case ERMs, average case tasks, worst-case datasets, or only specific dataset sizes $n \ldots$ We note that replacing the sup over tasks (resp. expectation over dataset) with an expectation whose support is all invariant tasks (resp. sup over datasets) would not change the resulting representations as sample-optimal representations are actually optimal for any $n$, dataset, task as proven in Lemma 12. The biggest possible remaining point of disagreement (besides perfect optimality already discussed above) concerns the worst-case ERM. For example, one could argue that in practice the ERMs would have some inductive bias or implicit regularization towards margin maximizing ERMs.[8] This would definitely give different results for our main theorem and our current choice can then be seen as a limitation of our framework. We hope that our work will encourage others to derive a different framework for alternative definitions of optimality.

---

[8]For the specific case of max-margin classifiers, the resulting algorithms would likely not change as ETFs can be shown to already maximize the expected margin of ERMs.

# F Reproducibility

Table 8: Details of all datasets used in the paper.

| Dataset | Classes | Train size | Test size | Evaluation metric |
|---|---|---|---|---|
| ImageNet [59] | 1000 | 1 281 167 | 50 000 | accuracy |
| TinyImageNet [56] | 200 | 100'000 | 10'000 | accuracy |
| Food [120] | 102 | 75'750 | 25'250 | accuracy |
| Cifar10 [121] | 10 | 50'000 | 10'000 | accuracy |
| Cifar100 [121] | 100 | 50'000 | 10'000 | accuracy |
| Cars [122] | 196 | 8'144 | 8'041 | accuracy |
| Aircrafts [123] | 100 | 6'667 | 3'333 | mean per class |
| Describable Textures (DTD) [124] | 47 | 3'760 | 1'880 | accuracy |
| Pets [125] | 37 | 3'680 | 3'669 | mean per class |
| Caltech [126] | 102 | 3'060 | 6'085 | mean per class |
| Flowers [127] | 102 | 2'040 | 6'149 | mean per class |

## F.1 TinyImageNet experiments

For TinyImageNet experiments, all results come from our own implementation. The code to reproduce all our experiments is at `github.com/YannDubs/Invariant-Self-Supervised-Learning`.

Unless stated otherwise we have the following hyperparameters for all experiments. All experiments ran with 16 bits precision.

**Data.** For the first experiments in the main paper, we use TinyImageNet [56], which contains 100k ImageNet [59] images of 200 classes downscaled to 64×64. We normalize each image with `mean=[0.480, 0.448, 0.398]` and `std=[0.277, 0.269, 0.282]`.

**Encoder's architercture.** The encoder is a pre-activation ResNet18 [128] with a one hidden layer of 1024 neurons MLP projection head. Both the MLP and the ResNet use batch normalization [129]. All activations are ReLUs. As is standard, the representation is the 512-dimensional output of the `res5avg` block.

**ISSL training.** For training all the ISSL encoders we use a batch size of 512, 300 epochs, the optimizer is Adam [130], the learning rate follows a cosine schedule with a maximum of 4e-3 with 10 epochs warmup, the weight decay is 1e-6.

**Hyperparameter tuning.** For tuning hyperparameters, we first tuned optimizers (weight decay, learning rate, . . . ) on a $10\%$ held-out training data for SimCLR and DINO which were respectively the best performing contrastive and distillation methods. We found that the hyperparameters were relatively similar so we chose values that worked for both. We then used those values for CISSL and DISSL without further tuning those training parameters. The main hyperparameters that were tuned during the development of CISSL and DISSL were tuned on CIFAR10 [121]. We tuned the $\beta, \lambda$ Lagrangian parameters of DISSL on a 10 held-out training set of TinyImageNet.

**Baselines.** For contrastive baselines, we use SimCLR with standard hyperparameters (output dimensionality 128, temperature 0.07). For distillation baselines, we chose the best over DINO, SwAV, SimSiam. SwAV did not perform well on TinyImageNet. SimSiam could generally perform slightly worst than DINO but sometimes better (it had high variance and dependence on hyperparameters, especially weight decay). We thus used DINO as a baseline but were surprised to see that all distillation baselines significantly underperformed compared to SimCLR on TinyImageNet. For DINO we use all the hyperparameters from their paper besides the size of the teacher outputs. Indeed, we found that the large output of 65000 used in ImageNet did not perform well, we thus decreased it to 1000 (which performed similarly to 10000).

**CISSL.** For CISSL we essentially use the implementation provided in Source Code 1.

**DISSL.** For DISSL we essentially use the implementation provided in Source Code 2. The only difference is that to avoid having a large number of parameters when we increase dimensions we use low-rank linear layers (rank 512) for the last layers of the projection heads. This can be efficiently

implemented by stacking linear layers, e.g., in Pytorch we have `[nn.Linear(z_dim,512),nn.BatchNorm1d(512),nn.Linear(z_dim,n_equiv)]`.

**Augmentations.**. We augment every input twice in a batch (e.g. once for the teacher and once for the student). For augmentations we follow [34] and use: (i) grayscaling with probability p=$0.1\alpha$; (ii) color jittering with strength `brightness`=$0.4\alpha$, `contrast`=$0.1\alpha$, `saturation`=$0.2\alpha$, `hue`=$0.1\alpha$; (iii) cropping with scale `scale`=$[0.1/\alpha, 1]$. By default $\alpha = 1$. When sweeping over augmentation strength in Fig. 7a we sweep over $\alpha \in \{0.25, 0.5, 1, 2\}$. For "coarse aug." in Table 1 we use $\alpha = 2$ and additionally apply a Gaussian blur with probability 0.5 and kernel size of 10% of the image.

**Modifying the dimensionality of the representation.** Naively increasing the dimensionality by mapping the representation through a linear layer is not sufficient as the effective dimensionality of the representation would still be small (the representation would live on a manifold of at most the dimensionality of the previous layer). To increase dimensionality we thus increased the number of channels right before the average pooling layer. We tried two ways that worked equally well. The first and simplest method, is to pass the latent image (ie before average pooling) through a linear layer (i.e. convolution layer of kernel size 1) followed by batchnorm and ReLU. Although this worked well for ResNet18, it would significantly increase the number of parameters for larger models. For example, if we want the dimension of a ResNet50 to be $8192$ we would need a linear layer of $8192 \times 2048 \approx 16M$ parameters. We thus ended up using a bottleneck so that our method is a general way of increasing dimensionality while keeping the number of parameters manageable. Specifically, we used a small convolution bottleneck block that consists of `[cbr(out_chan=512,k=1),cbr(out_chan=512,k=3),cbr(out_chan=z_dim,k=1)]`, where `cbr` denotes convolution, batchnorm,relu and `k` denotes kernel size. We emphasize that those layers are not necessary and are just a way of keeping the number of parameters small if the input is already relatively high dimensional (which is not the case with ResNet18s).

**Linear probing.** For the linear probe, we use a logistic regression trained with SGD, `momentum=0.9`, cosine learning rate schedule with a maximum of $0.6$, batch-size $512$, $100$ epochs. We found the best weight-decay to be different for different models and so we evaluate all models with a probe trained with weight decay 1e-4, 1e-5, 1e-6 and always give the best result.

**Non-linear probing.** For non-linear probing, we use a 2 hidden layer (2048 units) MLP with batch-normalization. The optimization is the same as for linear probing. When using non-linear probes we use non-linear projection heads of the same architecture for our objectives, as discussed in Sec. 5.

**Distance to ETF.** To compute the distance to ETFs in Fig. 4 we consider how close positive examples and how far negatives are from one another. In particular, we first center each representation at $\mathbf{0}$ by subtracting the (exponential moving average) of the mean. We then unit-normalize the representation. Let $\tilde{\phi}(\tilde{x})$ denote this unit-normalized centered representation of the augmented example $\tilde{x}$. To compute how close positives $\tilde{x}, \tilde{x}^+$ are to one another we consider one minus their expected dot products $\text{pos} = 1 - \mathbb{E}_{\tilde{x},\tilde{x}^+}\left[\tilde{\phi}(\tilde{x})^\top \tilde{\phi}(\tilde{x}^+)\right]$, which is always a positive quantity due to the unit+centering processing. To compute how far positives $\tilde{x}, \tilde{x}^-$ are from one another we consider their expected dot product $\text{neg} = 1 - \mathbb{E}_{\tilde{x},\tilde{x}^-}\left[\tilde{\phi}(\tilde{x})^\top \tilde{\phi}(\tilde{x}^-)\right]$, which is at least $-\frac{1}{|\mathcal{X}/\sim|-1} \approx 0$ (the minimal value is typically positive and grows as the number of equivalences grows for a fixed ambient dimension). Note that we have an ETF if and only if both metrics are minimized $\text{pos} = 0$ and $\text{neg} = -\frac{1}{|\mathcal{X}/\sim|-1}$ we will thus loosely refer to $\text{pos} + \text{neg}$ as "distance to ETF".

### F.2 ImageNet experiments

For ImageNet experiments we used Facebook's `VISSL` package [60]. All the implementations of all baselines are thus well tested, furthermore this package is maintained in part by the authors of SwAV which is our main baseline. We reran all baselines using the same batch size, optimization schemes, evaluation pipeline, and numerical precision for a fair comparison. For all the implementation details and hyperparameters we do not discuss, we refer the reader to VISSL's code and documentation. After the anonymous period, we will push our code and configuration files to `VISSL`.

**Our methods.** For our CISSL and DISSL, we did not perform any hyperparameter tuning and used all the same hyperparameters as in the TinyImagenet experiments (besides for the optimization, in which case we used the one from baseline as discussed below). The only difference compared to

TinyImageNet is that in all our ImageNet experiments we used a dimensionality of 8192 for the representation (using the method discussed in Appx. F.1).

**Standard augmentations (Table 3).** For Table 3 we compare methods with standard ImageNet experiments (augmentations from SimCLR [1]). We optimize all methods using the optimization procedure from SwAV and SimCLR in VISSL. Namely, cosine learning rate schedule with 10 epochs of linear warmup, learning rather of 0.3 for a batch size of 256 but linearly rescaled for larger batch sizes, SGD+LARC optimizer with 0.9 momentum, batchnorm synchronization over GPUs. We use 100 epochs (due to computational constraints) and a batch size of 320 per GPU (maximum we could fit) on 8 different GPUs, i.e., a total batch-size of 2560. For all models, we used 16-bit floating precision.

**Multi-cropping (Table 4).** For Table 4 we use the same hyperparameters than Table 3 except we use $160 \times 2 + 96 \times 4$ multi-crop augmentations.

**Linear probing in distribution.** For training the linear probe efficiently we first featurize all of ImageNet by the model to be evaluated. This has the advantage of being very computationally efficient but means that we cannot train the linear probe with augmentations. Once ImageNet featurized, we train 10 different linear probes using PyTorch and chose the best on a validation set. Each of those probes is trained with a batch-size of 2048, 16 fp precision, 100 epochs, SGD with momentum, a standard cosine learning rate schedule. They only differ on their learning rate ($[0.03, 0.3]$) and weight decay ($[0, 1e-5]$). We generally found that a learning rate of $0.1$ and weight decay of $3e-6$ worked well for most models.

**Transfer (Table 5).** For Table 5 we evaluate the best SwAV and DISSL models from Table 4 on (a subset of) the standard transfer benchmarks from [61]. All datasets and their evaluation metrics are shown in the second part of Table 8. For training the linear probe we use Sklearn [131]. In particular, we first featurize the training and testing sets (i.e. we do not apply augmentations when training the linear probes). We then min-max scale features to $[0, 1]$ and train both a linear SVM [132] and a logistic regression (with lbfgs solver) using Sklearn. The only parameter we tune is the regularization strength $C$. In particular, we do so by choosing the best $C$ out of 10 values that are log-uniformly space in $[1e-4, 100]$. The final result is then the best between the linear SVM and the logistic regression.

# G   Additional experimental results

In the main paper, we only showed experiments that show our framework's actionable insights in realistic settings. In the following sections, we provide additional results in realistic settings, as well as test more carefully our theoretical claims in controlled settings. In Appx. G.1 we validate our theoretical claims in a setting that is as close as possible to the theory, i.e., the idealized setting. In Appx. G.4 we add additional results in the more realistic settings considered in the paper.

## G.1   Validating our theory in the idealized setting

First, we tested our theory in a controlled setting as close as possible to our theory, i.e., the idealized setting. Specifically, unless stated otherwise, we used (i) ISSL log loss Eq. (6) to avoid approximate objectives; (ii) the maximal invariant $M_A$ given by the true labels $Y$ (CIFAR10; [121]) which simulates knowledge of the tasks' equivalence structure; (iii) the labeled train distribution as testing distribution to simulate access to infinite unlabeled data $p(X)$; (iv) the worst accuracy over 10 coarsenings of the CIFAR10 task $\mathcal{T}$ to estimate the supremum over tasks in Def. 9; (v) a regularizer enforcing the encoders invariance as in Theorem 1 by minimizing $\|\phi(x) - \phi(x^+)\|$. All results are over 3 seeds.

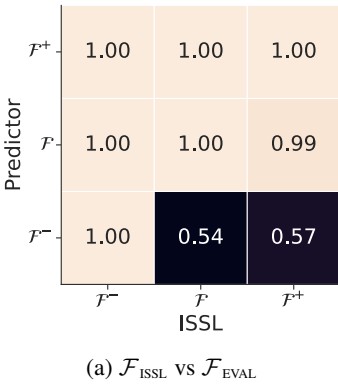
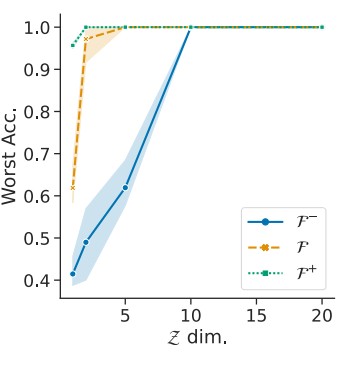

(a) $\mathcal{F}_{\text{ISSL}}$ vs $\mathcal{F}_{\text{EVAL}}$        (b) Effect of $\mathcal{F}$ on dim.

Figure 9: Different downstream functional family $\mathcal{F}_{\text{EVAL}}$ require (a) corresponding or smaller functional family during ISSL $\mathcal{F}_{\text{EVAL}} \subseteq \mathcal{F}_{\text{ISSL}}$; (b) smaller dimensionality of $Z$ when $\mathcal{F}$ is larger. The X-axis of the heatmap "ISSL" shows the predictors $\mathcal{F}_{\text{ISSL}}$ used during ISSL to predict $M(X)$. The Y-axis of the heatmap "predictor" shows the downstream probes $\mathcal{F}_{\text{EVAL}}$ used for evaluation. We used three families $\mathcal{F}^- \subseteq \mathcal{F} \subseteq \mathcal{F}^+$: a linear $\mathcal{F}^-$, a small MLP $\mathcal{F}$ (hidden unit: $[10]$), and a large MLP $\mathcal{F}^+$ (hidden units: $[2048, 2048]$). The values of the heatmap, as well as the Y-axis of the line-plot, show the performance of the worst (over 10) binary invariant tasks. The X-axis shows the dimensionality of the learned representation. The data is CIFAR10 and the underlying invariance structure is given by the labels of CIFAR10.

First, we considered the effect of predictors $\mathcal{F}$ on ISSL. The results are shown in Fig. 9.

**Increasing $\mathcal{F}$ decreases the required dimensionality.** Props. 19 and 20 show that the necessary and sufficient dimensionalities $d_{\min}(\mathcal{F}), d_{\text{suff}}(\mathcal{F})$ decrease (monotically) with the complexity of $\mathcal{F}$, while Theorem 1 shows that for linear $\mathcal{F}^-$ we have $d_{\min}(\mathcal{F}^-) = d_{\text{suff}}(\mathcal{F}^-) = |\mathcal{X}/\sim| - 1$. To test that we swept over dimensionality of the representation and complexity of the probing family. As predicted, Fig. 9b shows that for linear $\mathcal{F}^-$ the required dimensionality is $d(\mathcal{F}^-) = |\mathcal{X}/\sim| - 1 = 9$, while it shrinks for more complex predictors: $d(\mathcal{F}) \approx 5$, $d(\mathcal{F}^+) \approx 2$.

**One should consider $\mathcal{F}$ during ISSL.** Prop. 19 suggests that one should predict the maximal invariant using a family $\mathcal{F}_{\text{ISSL}}$ that is (at most) a subset of the true downstream probes $\mathcal{F}_{\text{EVAL}}$ to ensure that $M(X)$ is predictable by some $f \in \mathcal{F}_{\text{EVAL}}$. We tested this by sweeping predictor size for $\mathcal{F}_{\text{ISSL}}$ and $\mathcal{F}_{\text{EVAL}}$. As predicted, Fig. 9a shows perfect performance only when $\mathcal{F}_{\text{ISSL}} \subseteq \mathcal{F}_{\text{EVAL}}$. This suggests that it is important to consider downstream $\mathcal{F}_{\text{EVAL}}$ during ISSL. Furthermore, we see that, at least in theory, the performance can be very low when using the wrong predictors and worst case. Table 2 of the main paper shows much smaller gains for realistic settings and realistic tasks (not worst-case).

We then investigated the augmentations or equivalences $\sim_A$ used for ISSL.

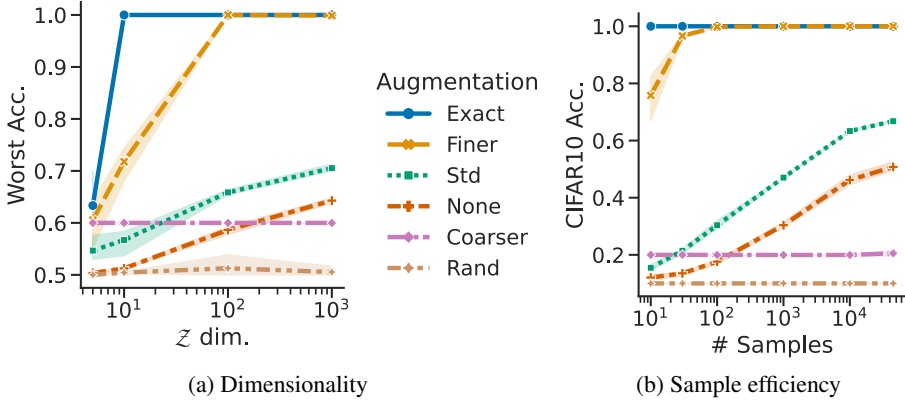

(a) Dimensionality       (b) Sample efficiency

Figure 10: Using coarser but sufficient augmentations decreases the required (a) dimensionality; and (b) the number of samples to perfectly predict all invariant tasks. Each line style/color corresponds to a different equivalence structure used for ISSL: "Exact" denotes 10 equiv. classes given by CIFAR10 labels; "Finer" denotes 100 equiv. classes given by aggregating 10% of same-label images; "Std" denotes standard data augmentations; "None" denotes no augmentations; "Coarser" denotes 2 equiv. classes given by aggregating labels; "Rand" denotes 1000 equiv. classes given by aggregating any images together. Note that both "coarser" and "rand" are not invariant tasks. In (a) the X-axis shows the dimensionality of the representation and the Y-axis shows the performance on the worst binary invariant task over 10 samples. In (b) the X-axis shows the number of samples used to train downstream probes and the Y-axis shows the performance of a standard ERM on CIFAR10. 3 seeds.

**Coarser augmentations improve sample efficiency.** As predicted by Prop. 2, Fig. 10b shows that augmentation that give finer equivalences ("Finer" / "Std" / "None") than desired ("Exact") achieve optimal performance but require more downstream samples. Furthermore, for "Exact" and "Finer" we see that as long as one example is seen per equivalence class the performance is perfect (respectively achieved at a number of classes 10 and 100) as predicted by Corollary 14.

**Coarser augmentations require smaller dimensionalities.** Theorem 1 shows that the dimensionality requirement depends on the invariance structure. So although any label-preserving augmentation can give optimal encoders, coarser augmentation will need smaller dimensionalities. Fig. 10a indeed shows that ISSL with finer equivalences ("Finer" / "Std" / "None") than desired ("Exact") achieve optimal performance on invariant tasks but require higher dimension of the representation $d$. In contrast, other augmentations ("Coarser" / "Rand") underperform. By construction "Exact" and "Finer" have 10 and 100 equivalence classes, and we see that the required dimensionality $d(\mathcal{F}_{\text{lin}})$ is 9 and 99 as predicted by Theorem 1.

## G.2    Validating our theory in a more practical setting

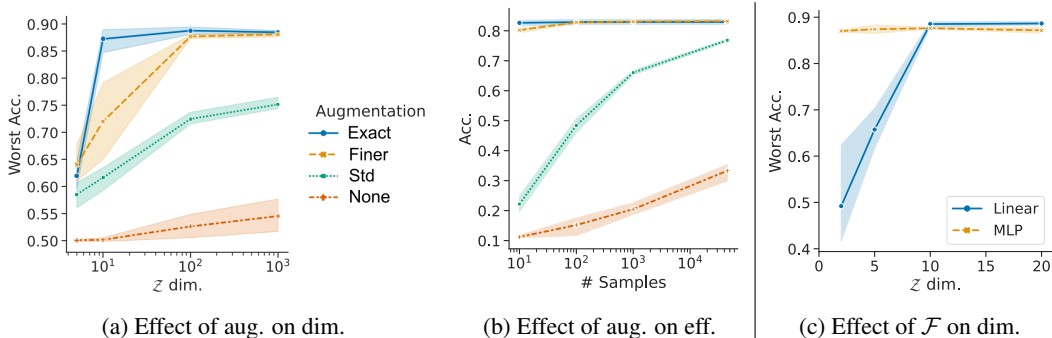

(a) Effect of aug. on dim.      (b) Effect of aug. on eff.      (c) Effect of $\mathcal{F}$ on dim.

Figure 11: Fine-grained predictors from our theory hold beyond the ideal setting. In particular: (a) shows the same trend as Fig. 10a; (b) as Fig. 10b; and (c) as Fig. 9b. The difference in this figure is that we use CISSL and consider unseen test examples.

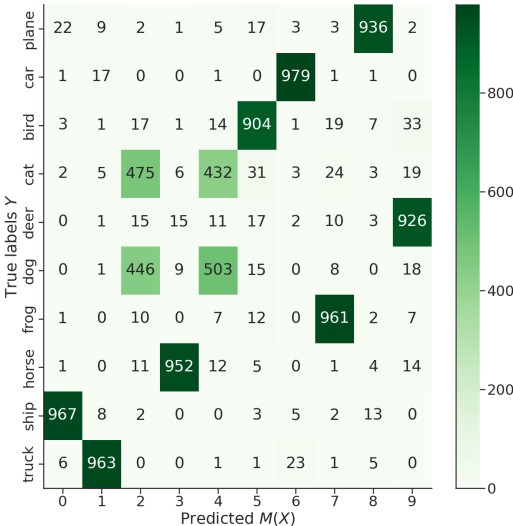

Figure 12: DISSL teacher estimates well $M(X)$ for CIFAR10 test set. DISSL is trained with supervised augmentations so that the true $M(X)$ is the underlying labels (up to permutation).

**Our theory generalizes to more realistic settings.** In the previous section, we validated some of our theoretical claims in a setting that is as close as possible to our theory. Fig. 11 show the same results in a more practical setting: where we use CISSL instead of ISSL log loss and we do not assume access to the underlying distribution, i.e., we use the test set for evaluating the probe. In particular: Fig. 11a shows the same trend as Fig. 10a with the best performance achieved from a dimensionality of $d(\mathcal{F}) = 9$ when using optimal augmentations; Fig. 11b shows the same trend as Fig. 10b with best performance achieved using 10 sample for exact augmentations; Fig. 11c shows the same trend as Fig. 9b. This suggests that our theory provides the right intuition in practical settings and can likely be generalized to cover those such cases of practical/approximate ISSL.

**DISSL's teacher estimates well** $M(X)$**.** DISSL's and CISSL's teacher respectively estimates of $M(X)$ and $w(M(X))$, a natural question concerns the quality of those estimates. For DISSL we can easily test that by using supervised augmentations, in which case $M(X)$ should be the supervised labels (up to permutations). Fig. 12 shows the confusion matrix between the predicted $M(X)$ and the underlying labels for CIFAR10's test set. We see that the teacher recovers $M(X)$ for all but cats and dogs. The teacher's $M(X)$ is 85% accurate, while linear probing on the student's representations gives 90% accuracy.

### G.3 Monitoring and understanding DISSL's training dynamics

As stated in Sec. 4.2, one of the advantages of DISSL is that it uses information theoretic losses, which are relatively interpretable. In the following, we discuss and analyze some of the quantities which we found helpful to monitor in order to understand and debug DISSL. All the following correspond to the DISSL model used for the second row of Table 1, i.e., a ResNet18 trained for 300 epochs on TinyImageNet with standard (512) dimensions and uniform prior over $C = 16384$ equivalence classes. All shown curves are on the training set and we use natural logarithms (base $e$).

**Maximality and marginal entropy** $\mathbb{H}[\hat{M}]$**.** A fundamental quantity in DISSL training is the KL divergence between the teacher's marginal and the prior probability of equivalence classes. Indeed, deterministic and invariant teachers are maximal if and only if $\mathbb{D}_{\mathrm{KL}}[q(\hat{M})\|p(M(X))] = 0$. In practice, the prior is typically uniform so minimizing the desired divergence is equal to maximizing the entropy of the teacher's marginal because $\mathbb{D}_{\mathrm{KL}}[q(\hat{M})\|p(M(X))] = \log C - \mathbb{H}[\hat{M}]$. We thus monitor the marginal entropy, which is maximized at $\log C = \log 16384 \approx 9.7$ nats. Fig. 13a shows that the marginal entropy stays high over the course of training. The final marginal entropy is $\mathbb{H}[\hat{M}] \approx 8.4$ nats, which means that the effective number of equivalence classes is $\exp(8.4) \approx 4447$. This is lower than the 16384 number of classes, which suggests either that the equivalence classes are not actually uniform (as is likely the case) or that the hyperparameter $\lambda$ in Algorithm 1 could be increased. Importantly, $\mathbb{H}[\hat{M}] \approx 8.4$ nats is still very far from a collapsed teacher, which would have

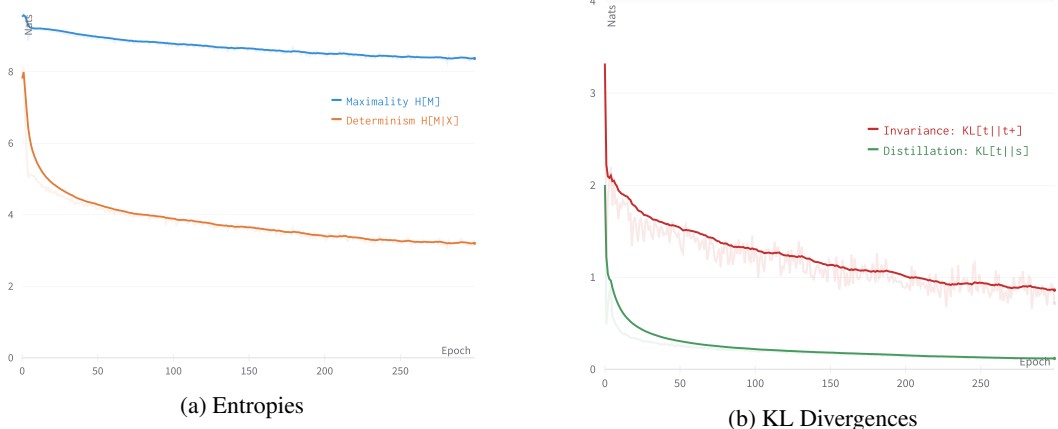

(a) Entropies          (b) KL Divergences

Figure 13: Components of the DISSL objective are useful metrics to monitor during training. Left: The marginal $\mathbb{H}[\hat{M}]$ (in blue) and conditional $\mathbb{H}[\hat{M}|X]$ (in orange) entropies respectively quantify the maximality and determinism of the teacher. Both entropies are non-negative and upper bounded by $\log 16384 \approx 9.7$. Right: The teacher-augmented teacher $\mathbb{D}_{\text{KL}}[q(\hat{M}\,|\,X)\|q(\hat{M}\,|\,X^+)]$ (in red) and student-teacher $\mathbb{D}_{\text{KL}}[q(\hat{M}\,|\,X)\|s_{\phi,w}(\hat{M}\,|\,X)]$ (in green) KL divergences respectively quantify the invariance of the teacher and distillation of the student. ResNet18 trained with DISSL on TinyImageNet for 300 epochs.

$\mathbb{H}[\hat{M}] = 0$ (one effective class). In practice, we found that having a large marginal entropy is key for learning good representations and so $\lambda$ is an important hyperparameter. In our TinyImageNet and ImageNet experiments we use $\lambda = 2.3$ and generally found that $\lambda \in [1.8, 2.8]$ gives good results.

For the following two metrics recall that in Sec. 4.2 we saw that we minimize the cross-entropy $\mathbb{E}_{q(\hat{M}\,|\,x)}[-\log q(\hat{M}\,|\,x^+)]$ between teacher's outputs on equivalent examples $x \sim x^+$ as this is equivalent to minimizing the conditional entropy $\mathbb{H}[\hat{M}\,|\,x]$ (equivalent to determinism) and the KL divergence $\mathbb{D}_{\text{KL}}[q(\hat{M}\,|\,x)\|q(\hat{M}\,|\,x^+)]$ (equivalent to invariance). As a result we use the same hyperparameter $\beta$ to control both determinism and invariance.

**Determinism and conditional entropy** $\mathbb{H}[\hat{M}|X]$**.** Maximizing the previous marginal entropy is trivial if the teacher is not deterministic, the teacher can simply predict a uniform distribution regardless of the input. Indeed, maximality and maximizing entropy are equivalent only for deterministic encoders. It is thus very important to monitor the conditional entropy $\mathbb{H}[\hat{M}|X]$, which quantifies the "distance" to a deterministic teacher ($\mathbb{H}[\hat{M}|X] = 0 \iff$ deterministic teacher). Fig. 13a shows that the conditional entropy greatly decreases during training and so the teacher becomes closer to determinism. At the end of training, we have $\mathbb{H}[\hat{M}|X] \approx 3.2$ nats, which intuitively means that the teacher hesitates in average between $\exp(3.2) \approx 25$ different clusters/equivalences classes for every example. Although the teacher is not actually deterministic, this shows that it clusters examples relatively confidently—compared to the trivial solution where the teacher would "hesitate" between 16384 classes and have a maximal conditional entropy of $\mathbb{H}[\hat{M}|X] = \log 16384 \approx 9.7$ nats. In practice, we found that as long as we avoid this trivial solution the value of the conditional entropy always decreased significantly during training. If the conditional entropy does not decrease $\mathbb{H}[\hat{M}|X] \approx 9.7$ then the teacher is essentially useless and this suggests increasing the hyperparameter $\beta$ in Algorithm 1, which controls both determinism and invariance. In our TinyImageNet experiments we use $\beta = 0.8$ and $\beta = 0.6$ for ImageNet experiments. Generally, we found that $\beta \in [0.4, 1]$ gives good results and any of those can work well if $\lambda$ is tuned accordingly.

**Invariance and KL divergence** $\mathbb{D}_{\text{KL}}[q(\hat{M}\,|\,X)\|q(\hat{M}\,|\,X^+)]$**.** The third and last requirement of the teacher is that it is invariant w.r.t. $\sim$, meaning that the KL divergence on any equivalent inputs $x \sim x^+$ must be zero $\mathbb{D}_{\text{KL}}[q(\hat{M}\,|\,x)\|q(\hat{M}\,|\,x^+)] = 0$. As with previous requirement, we can also monitor this during training and Fig. 13b shows that this divergence indeed decreases during training until $\mathbb{D}_{\text{KL}}[q(\hat{M}\,|\,X)\|q(\hat{M}\,|\,X^+)] \approx 0.9$. This small value shows that the teacher is close to being invariant but the curve seems to suggest that the model could benefit from longer training (not yet converged). Note that contrary to the previous entropies, the KL divergence does not have an upper bound.

**Distillation and KL divergence** $\mathbb{D}_{\mathrm{KL}}[q(\hat{M} \,|\, X)\|s_{\phi,w}(\hat{M} \,|\, X)]$. In addition to the three requirements from the teacher, we also monitor the distillation loss between teacher and student. Recall that in Eq. (9) we minimize the cross-entropy $\mathbb{E}_{p_X q(\hat{M} \,|\, X)}[-\log s_{\phi,w}(\hat{M} \,|\, X)]$ for distillation. This is nevertheless not a good monitor for distillation if the teacher $q(\hat{M} \,|\, X)$ is also being trained. Indeed, distillation can be perfect and cross-entropy not be zero because $\mathbb{E}_{p_X q(\hat{M} \,|\, X)}[-\log s_{\phi,w}(\hat{M} \,|\, X)] = \mathbb{H}[\hat{M}|X] + \mathbb{D}_{\mathrm{KL}}[q(\hat{M} \,|\, X)\|s_{\phi,w}(\hat{M} \,|\, X)]$, i.e., that cross-entropy is lower-bounded by the entropy of the teacher which is non-zero during training. To monitor distillation irrespective of the teacher's entropy we thus use the KL divergence. Fig. 13b shows that the distillation decreases during training until $\mathbb{D}_{\mathrm{KL}}[q(\hat{M} \,|\, X)\|s_{\phi,w}(\hat{M} \,|\, X)] \approx 0.11$, which shows that distillation is very effective.

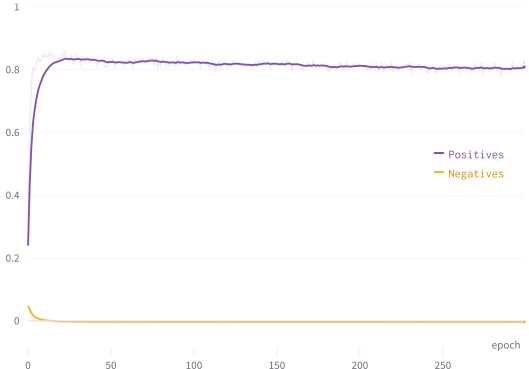

Figure 14: Cosine similarity between representations of equivalent (positives in purple) and non-equivalent examples (negatives in orange). ResNet18 trained with DISSL on TinyImageNet for 300 epochs.

**Cosine similarity between equivalent and non-equivalent representations.** In addition to the previous components of the losses, it can be useful to analyze and monitor the cosine similarity between representations of equivalent and non-equivalent examples. Recall that Prop. 3 is based on the fact that our objective can recover sETF representations. In particular, the representations of equivalent examples should be the same, while non-equivalent ones should have the largest possible angle between them. We thus monitor the expected cosine similarity between equivalent and non-equivalent examples. Note that for equivalent examples the cosine similarity would ideally achieve its maximum of 1. For negatives, the cosine similarity would ideally be minimized, and the expected minimum is $-\frac{1}{d} \approx -0.002$ where $d = 512$ is the dimensionality of the representation.[9] Fig. 14 shows that, as desired, the cosine similarity between positives increases during training, while the opposite is true for negatives. The final expected cosine similarity for positives is $\approx 0.81$ and $\approx -0.002$ for negatives. This shows that the learned representations are indeed close to ETFs but that representations are not yet completely invariant—which makes sense given that the teacher still seems to be learning to be invariant as seen in Fig. 13b).

### G.4 Realistic

Table 9: ResNet18 and ResNet50 with the same dimensions. Linear probing TinyImageNet, DISSL.

|           | RESNET18 | RESNET50 |
| --------- | -------- | -------- |
| $d$=512   | 45.9     |          |
| $d$=2048  | 47.7     | 49.1     |

**Gains from ResNet50 come from increasing dimensionality.** In Fig. 7c of the main text, we say that increasing dimensionality has a large impact on downstream performance. This naturally begs

---

[9] $-\frac{1}{d}$ is only achievable if the number of equivalence classes is $|\mathcal{X}/\sim| = d + 1$, more generally the expected minimum is $-\frac{1}{|\mathcal{X}/\sim|-1}$ but the true number of equivalence classes is typically unknown and typically larger than the dimensionality.

the question of whether the gains when increasing the model size are due to the increase in capacity (number of parameters) or the fact that such models use larger dimensionalities. Table 9 shows that for DISSL on TinyImagenet, half of the gains are due to an increase in dimensionality (512 to 2048) rather than the number of parameters (11M to 23M).[10]

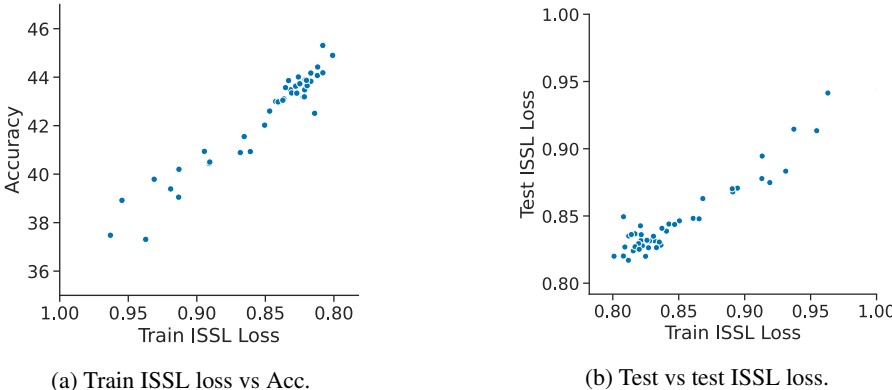

(a) Train ISSL loss vs Acc.

(b) Test vs test ISSL loss.

Figure 15: Training ISSL log loss is highly correlated with (a) linear probing accuracy on TinyImageNet; (b) test ISSL log loss. Each 44 point corresponds to a CISSL model trained with different hyperparameters as in Fig. 7b.

**ISSL training log loss correlates with performance.** In Fig. 7b we have seen that the test ISSL log loss correlates highly with the downstream performance. Fig. 15a shows that the same holds for the training log loss. We also see that the test and train ISSL log loss are very similar Indeed Fig. 15b shows that train ISSL loss is highly correlated to test ISSL loss. Note that the testing loss is slightly better likely due to freezing of the batch normalization layer. This suggests that in ISSL generalization of the pretext task is not an issue. The points in Fig. 15a are a subset of those from Fig. 7b as we did not originally log the training ISSL log loss.

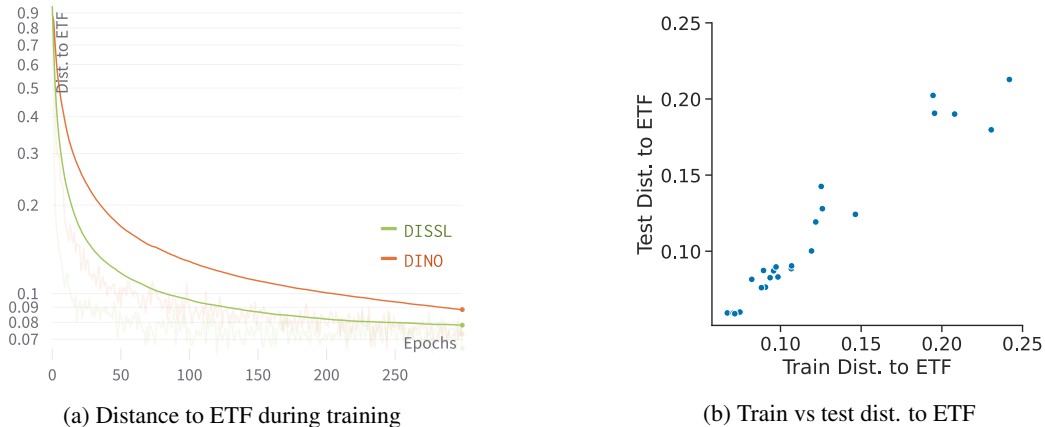

(a) Distance to ETF during training

(b) Train vs test dist. to ETF

Figure 16: ISSL representations tend towards ETFs both during training and on the test distribution of TinyImageNet. (a) Distance of the representations learned by DISSL and DINO to an ETF. Both seem to converge to ETFs although DISSL does it quicker. (b) Correlation between the training and the test distance to ETFs for various CISSL and SimCLR models trained with different hyperparameters.

**ISSL learns ETFs during training.** Prop. 3 suggests that optimizing ISSL log loss achieves optimal representations that are essentially ETFs. To investigate that, we computed during training the distance between the representations in a batch and an ETF (see Appx. F.1). Fig. 16a suggests that the train representations become closer to an ETF as training advances, i.e., ISSL log loss decreases. Furthermore, this seems to hold both for our methods (here DISSL) and baselines (here DINO), although our DISSL converges quicker and closer to an ETF.

---

[10]For reasons that we do not yet understand, $d = 512$ with ResNet50 gave NaNs.

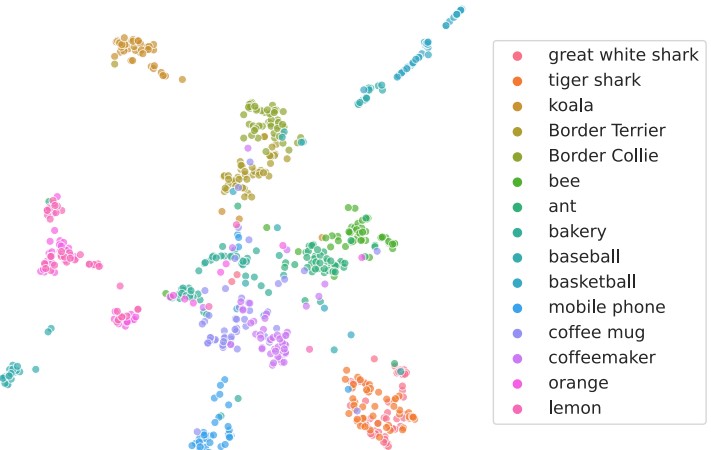

Figure 17: Test representations learned by DISSL on 15 labels of ImageNet. The representations are reduced from 8192 to 2 dimensions using UMAP [133] with 15 nearest neighbors and an effective distance of 0.05 between embedded points. There are $15 \times 50 = 750$ examples.

**ISSL learns ETFs even on test.** A standard criticism (e.g. [111]) of neural collapse in supervised learning is that the phenomena seem to only hold on the training set. For ISSL, we see in Fig. 16b that neural collapse happens also on the test set (the values are not only correlated but nearly identical). This is likely because equivalent classes given by standard augmentations are much more fine-grained than those given by class labels.

**ISSL clusters representations meaningfully.** Fig. 17 shows (using UMAP) the representations learned by DISSL on ImageNet. We see that DISSL clusters meaningfully each of the test examples for a subset of 15 labels. In particular, we see that examples are either clustered with examples from the same labels or from a similar one, e.g., "great white shark" with "tiger shark" or "coffee mug" close to "coffeemaker". This shows that the equivalence class perspective does not explain everything: examples from semantically similar equivalence classes are typically clustered together. This suggests that one should consider also the relation between equivalence classes to fully understand ISSL.

Table 10: DISSL results on ImageNet for different epochs, dimensionalities and multi-crops. 2056 batch size, ResNet50.

| EPOCHS | DIM. | MULTI-CROPS | TOP 1 ACC. (NO AUG) | TOP 1 ACC. | TOP 5 ACC. |
|--------|------|-------------|---------------------|------------|------------|
| 100 | 2048 | $2 \times 224$ | 66.3 | 66.9 | 87.5 |
| 100 | 8192 | $2 \times 224$ | 67.7 | 68.9 | 88.5 |
| 100 | 8192 | $2 \times 160 + 4 \times 96$ | 69.7 | 70.7 | 89.4 |
| 400 | 2048 | $2 \times 224$ | 70.4 | 71.1 | 90.2 |
| 400 | 2048 | $2 \times 160 + 4 \times 96$ | 71.4 | 73.0 | 91.3 |
| 400 | 8192 | $2 \times 160 + 4 \times 96$ | 72.6 | 74.0 | 91.9 |
| 800 | 8192 | $2 \times 224 + 4 \times 96$ | 72.8 | 73.9 | 91.9 |

**Additional DISSL results on ImageNet.** Table 10 shows additional results of DISSL on ImageNet. In particular, it shows that the trends we saw on TinyImageNet (Table 1) hold on ImageNet. Namely, increasing dimensionality, training for longer, and coarsening augmentations give significant linear probing gains. We show both results of a linear probe trained with augmentations (as is standard in the literature) and without augmentations (as is more realistic in small compute regimes). Indeed, without augmentations, one can featurize the training set once, which is much more efficient.