# OpenReview forum: "Improving Self-Supervised Learning by Characterizing Idealized Representations"
_NeurIPS.cc/2022/Conference — NeurIPS 2022 Accept_

### Official Review · Reviewer_L1ta · 2022-07-11

**Rating:** 7
**Confidence:** 3
**Soundness:** 3 good
**Presentation:** 3 good
**Contribution:** 3 good

**Summary:**

In this work, the authors characterize the properties of an ideal set of representations for self-supervised learning. Namely, they use the notion of equivalence classes with respect to some transformation, and show that under certain assumptions, it is possible to characterize a sample optimal encoder for the desired equivalence classes. Using intuition from this characterization, they formulate a variant of self-supervised loss which can learn these optimal encoders under ideal conditions. The authors also evaluate practical objectives based on this loss, and show their competitiveness against other similar techniques from the literature.

**Questions:**

I would like to know whether the authors have performed any experiments regarding transferability of the learned representations for their methods. This is an experiment commonly seen in the literature of this topic, and it would be good if the authors could include it in the paper.

As mentioned above, I would also suggest the authors to include parts of the related work and proof sketches in the main paper, to help with reader understanding.

**Limitations:**

The authors have adequately adressed the limitations of their work (namely, the idealized setting of their theory and the lack of constraints on the encoders and tasks considered), both briefly in the main paper and extensively in the appendix. Moreover, I don't believe there is any negative societal impact arising from this work. As such, I believe the authors have addressed these topics sufficiently.

**Strengths And Weaknesses:**

This is a solid work overall, with several strong points:

- First of all, the topic of this work (self-supervised learning) is of great interest to the NeurIPS community. The authors analyze in great detail this paper’s related work, and properly demonstrate how it is positioned with relation to previous works in the literature. The paper is also very cleanly written overall. As a minor comment, although I understand that the authors moved the related work to the appendix due to space limitations, I believe that at least a short version of its should appear in the main paper.

- Secondly, the theoretical foundation of this work is solid. While the idealized encoders are defined using ideas which exist as a notion in the literature (for example, the fact that we want our encoders to be invariant to augmentations and predictive of the latent classes, which are conditions set in Theorem 1), this work provides a careful formalization of these ideas. Moreover, with Propositions 2 and 3 the authors provide an expression for the expected risk of those encoders and a loss that when minimized recovers these optimal encoders. Even though these theorems are only applicable under idealized assumptions, they still provide a good theoretical characterization of the idealized representations.

- Moreover, the link between the theoretical formulation of the ISSL loss and the practical versions defined in the paper is clear and well-motivated. The authors accurately assess that the main problem in the formulation of the ISSL loss is the lack of knowledge regarding the underlying classes and take measures to rectify this in the practical versions of the loss. I also appreciate the discussion on how each of the two formulations proposed by the authors (CISSL and DISSL) relate to contrastive and non-contrastive technique in the literature, respectively. In my opinion, this provides a good foundation of the proposed techniques, in relation to existing literature. The experimental evaluation of the two proposed methods also demonstrates that they are competitive and in most cases outperform the previous techniques, despite their simplicity.

- Finally, the authors extensively prove their theoretical statements and expand upon connections to the related work in the appendix. The added experiments in the appendix also go beyond standard performance evaluation based on accuracy, and examine various effects such as the augmentations used and the dimensionality of the representations. This is very useful in order to fully understand the proposed method.

A few weaknesses I can identify are the following:

- At the beginning of Section 4, the introduction of the loss function is somewhat abrupt, and while Proposition 3 demonstrates the link of the loss to the optimal encoders discussed above, I believe that including a proof sketch for the proposition or intuition on why it holds would help the connection with the previous parts of the paper. In a similar vein, I believe expanding upon Figure 4 would be useful, because currently it seems to be somewhat disconnected from the rest of the paper.

- A key limitation in the idealized setting is that it holds only in if the dimension of the representation is at least the number of equivalence classes minus 1. Combined with the assumption that the encoder is invariant to the equivalence class, this geometrically implies that there exists one point for each class in the representation space, and they can be shattered by halfspaces (which can be easily done if the points are general enough and they are at most 1 plus the representation dimension). Lemma 13 in the appendix expands upon this, but I believe it should also be pointed out in the main paper, because it may seem as an oversimplification otherwise.

- I believe that Section 5 should be shortened and included in the Section 4. This would allow parts of the appendix (namely, related work and some proof sketches) to be included in the main paper without harming the experimental section (since non-linear probes are only a small part of the main paper).

Overall, this is a strong and clearly written work. The above flaws I pointed out are minor, and I believe that the main issues I have can be addressed in the author response.

Minor comments:
- In Section 3, consider explicitly highlight the fact that equivalence classes are different from downstream classes, to avoid confusion.
- Line 43, $\mathcal{T}$ is not defined.
- Line 224: missing "and"
- Appendix H.2 seems to be missing.

---

> ### Author Response · Authors · 2022-08-02
> **We made all the suggested changes to the writing.**
>
> > The authors analyze in great detail this paper’s related work [...]. As a minor comment, although I understand that the authors moved the related work to the appendix due to space limitations, I believe that at least a short version of its should appear in the main paper
>
> We agree with the reviewer’s suggestion and moved part of our related work (Appendix E) to the main paper. In particular, we now summarize the related work concerning our theory and our actionable insights (dimensionality, augmentations, projection heads, DISSL). **Please see section 6 in the updated manuscript**.
>
> ---
>
> > Section 4, the introduction of the loss function is somewhat abrupt, and while Proposition 3 demonstrates the link of the loss to the optimal encoders discussed above, I believe that including a proof sketch for the proposition or intuition on why it holds would help the connection with the previous parts of the paper. In a similar vein, I believe expanding upon Figure 4 would be useful.
>
> We thank the reviewer and we acted upon their useful suggestions to improve the flow of the beginning of section 4. In particular, we extended the legend of Figure 4 and now use the **following intuitive explanation of proposition 3** (cf lines 139-150 in the updated manuscript):
>
> >> Our key insight is that we can learn sample-optimal encoders by jointly training the encoder and a logistic regression predicting $M(X)$ from the representations. Indeed, as shown in the neural collapse literature [16 – 21], the loss of logistic regression is minimized when representations form a simplex equiangular tight frame (sETF) as illustrated in Fig. 4. Namely, when: (i) M (X) is linearly predictable; (ii) the representation of same-class examples collapse; (iii) the angle between representations of different classes is maximal. Intuitively, each of those properties implies a requirement from Theorem 1, namely: (i) corresponds to F-predictability and comes from the linearity of logistic regression; (ii) corresponds to invariance and comes from the fact that log loss decreases variance due to Jensen’s inequality; (iii) implies maximal effective dimensionality and comes from the symmetry of log loss.
>
> ---
>
> > I believe that Section 5 should be shortened and included in the Section 4. This would allow parts of the appendix (namely, related work and some proof sketches) to be included in the main paper.
>
> We think that the additional content page (10th page) is sufficient to address the reviewer's previous points and include part of the appendices in the main paper (cf updated manuscript). If it isn’t sufficient to add the desired experiments, we will follow the suggestion and shorten the discussion on non-linear probes (section 5).
>
> ---
>
> > A key limitation in the idealized setting is that it holds only in if the dimension of the representation is at least the number of equivalence classes minus 1. Combined with the assumption that the encoder is invariant to the equivalence class, this geometrically implies that there exists one point for each class in the representation space, and they can be shattered by halfspaces [...]. Lemma 13 in the appendix expands upon this, but I believe it should also be pointed out in the main paper, because it may seem as an oversimplification otherwise.
>
> Disclaimer: we are not sure we correctly understood the reviewer’s suggestion on what to point out in the main paper.
>
> Concerning shatterability: we **added the following footnote** after the discussion of VC dimension in section 3.1 (cf line 101 in updated manuscript). Does it address that concern?
>
> >> The difference with statistical learning theory is that instead of (binary) shatterability of all examples, we want $k$-ary shatterability of all equivalence classes from representations. The key is that both notions coincide when using specific probes (eg linear) and invariant encoders (which are necessary for sample optimality).
>
> Concerning dimensionality: we agree that assuming unconstrained dimensionality of representations is an important simplification. This simplification allowed us to derive our actionable insights on increasing the dimensionality of the representation (Table 1), as theorem 1 shows that $d_* = |\mathcal{X}/\sim| -1$ is necessary and sufficient for optimality. Studying a fixed $d < d_*$ could provide additional actionable insights. We hope that our work will serve as a stepping stone for studying optimal representations under constraints (on $d$ or on the encoder). We now clarified in our limitations (cf line 424 in the updated manuscript)  that by “unconstrained encoders” we mean both the mapping and its codomain (representation space).
>
> ---
>
> We thank the reviewer for all minor comments. We addressed all of them.

---

> ### Author Response · Authors · 2022-08-02
> **We will evaluate our new model on transfer learning.**
>
> > I would like to know whether the authors have performed any experiments regarding transferability of the learned representations for their methods. This is an experiment commonly seen in the literature of this topic, and it would be good if the authors could include it in the paper.
>
> We had not tried any transfer experiments given that our theory does not concern this setting. We nevertheless agree that this is a natural question to ask for any novel SSL method and **we will thus include a comparison of the ImageNet-pretrained DISSL vs SwAV on the standard transfer suite** from [1] in the main paper.
>
> [1] S. Kornblith, J. Shlens, and Q. V. Le. “Do better imagenet models transfer better?” In: Conference on Computer Vision and Pattern Recognition (CVPR). 2019.

---

> ### Author Response · Authors · 2022-08-02
> **We thank the reviewer and we will act upon all their suggestions.**
>
> **We thank the reviewer for the thoughtful, high-quality, and constructive review**. We are thankful that the reviewer looked in more detail at our theory and parts of our appendices. We will address the reviewer’s feedback, which we believe will improve the final manuscript. In particular, we will take advantage of the additional content page to:
> 1. Make the suggested changes to the writing; in particular, expand the first few paragraphs of section 4 and summarize the related work section from the appendices in the main paper
> 2. Add transfer experiments.
>
> **We already made the changes to the writing (cf updated manuscript) and will update the main paper once we have transfer results.**

---

> > ### Comment · Reviewer_L1ta · 2022-08-08
> > **Thank you for your response.**
> >
> > I'd like to thank the reviewers for their very detailed response to my concerns.
> >
> > Regarding my comment about the dimensionality of the respresentations, I believe that the current discussion along with the added footnote are sufficient in order to make the point that the dimension should be at least the number of equivalence classes minus 1. As such, I believe this concern of mine has been properly adressed.
> >
> > Again, I believe this to be a solid work, given that my comments have been adressed.

---

### Official Review · Reviewer_j7cH · 2022-07-11

**Rating:** 6
**Confidence:** 4
**Soundness:** 3 good
**Presentation:** 3 good
**Contribution:** 3 good

**Summary:**

In this work, the authors characterize properties that representations learned in self-supervised way should ideally satisfy. They prove necessary and sufficient conditions such that for any task invariant to given data augmentations, the probes (e.g., linear or MLP) trained on such representation attain perfect accuracy. Based on these requirements the authors propose a unifying framework for improving existing SSL methods and deriving new ones. In contrastive learning regime, they propose asymmetric projection heads and in non-contrastive learning regime, they derive novel objective.

In the evaluation, the authors focus on the implications of the proposed framework using the TinyImageNet and ImageNet data sets. It consists of experiments and results
(i)   about the use of the proposed objectives to ensure sample optimal encoders in ideal settings,
(ii)  regarding the finding by the theoretical result (Theorem 1),
(iii) related to the use of coarser label-preserving augmentations (Proposition 2) and
(iv) for training with the smallest ISSL log loss to ensure optimality (Proposition 3).

The results show that the proposed self-supervised algorithms lead to improvements in performance with respect to the baselines on the used datasets for different ResNet neural network architectures.

**Questions:**

Could the authors comment more about how close the proposed theoretical foundation is to the practical cases?
Have the authors evaluated the proposed approach on other datasets (common in computer vision or other public ones)?


**Limitations:**

The authors address the limitations of the paper exposition in Section 7 Discussion and Outlook. Regarding the potential negative societal impact might be nice that the authors comment/discuss about the theoretical connections/aspect of their framework related to learning of possible biases/"short cuts" that might be undesirable as well as misused.

**Strengths And Weaknesses:**

Originality: The authors present a framework in an attempt to characters self-supervised learning and describe the problem grounded on invariant self-supervised learning and idealized representations. The authors formalized the notion of invariant self-supervised learning using an equivalence relation between original data and it's augmented version that should partitions the inputs X into  equivalent classes. Further they define optimal encoders at least theoretically, stating/defining population optimal and sample optimal cases that can produce ideal representations in terms of classification with minimum error (minimum empirical risk).

They give result about the required dimensionality of the population optimal encoders (at the output) by which the representations are linearly predictable. They argue that ERMs encoders that correctly predict one example in an equivalence class, also correctly predict all the others. They prove that such invariance of population optimal encoders is necessary and sufficient for sample optimality.

They also derive simple objectives that learn optimal encoders in ideal settings (infinite data, perfect optimizers, universal function approximators). In addition, they try to explain why previously proposed invariant self-supervised learning methods work, and comment how they might be improved in practice for contrastive invariant self-supervised learning and distillation invariant self-supervised learning.

Quality: The quality seems good. It is interesting how the authors state the problem formulation and relate it to empirical risk minimization in order to characterize the ideal representations (under ideal scenario) as well as how the authors relate to existing work and to practical considerations.

In the evaluation, the authors focus on experiments that are towards verifying the postulated theory. On the TinyImageNet data set they show large gaps of improvement of recondition accuracy compared to the used baselines. On the ImageNet data set they are somewhat on pair (with small improvement) compared to some of the state-of-the-art methods.

Clarity: The paper attempts to outlay an unified theory. Overall, the paper is well organized and the writing is good. The readers have to keep track of the many characteristics, propositions, derivations, etc, sometimes pushes the paper a bit on the heavy side, which might suggest that the content in this work needs more space. Maybe a table of content about the most important concepts could help.

The authors take on practical considerations for contrastive invariant self-supervised learning and distillation invariant self-supervised learning. Maybe focusing only on one of these two (or putting one in the appendix) might have made the paper lighter and the authors could have had more space to do more experiments as well as overall make the paper a more easy of a read. On the other hand, there are also a lot of additional details that the authors leave for the appendix

Significance: Unsupervised and self-surprised learning might be very important aspect towards the long-lasting goal of human-capable machine learning and artificial intelligence in practice. Theoretical foundation in this direction are of particular interest for understanding and improving existing works, and designing novel algorithms that could brings us closer to this goal.

---

> ### Author Response · Authors · 2022-08-02
> **We thank the reviewer.**
>
> **We thank the reviewer for taking the time to review our paper and for all the encouraging points. Given the generally positive review, we wonder if the reviewer has any feedback or concern that we could address to improve our paper and their evaluation of it?** If so, we would be happy to make the necessary changes to the manuscript.
>
> ---
>
> > Overall, the paper is well organized and the writing is good. [...] sometimes pushes the paper a bit on the heavy side, which might suggest that the content in this work needs more space. Maybe a table of content about the most important concepts could help.
>
> We agree that more scaffolding around the results could be helpful. **We used the additional content page to address this** by changing the following:
> 1. we added more scaffolding in section 3.1 and now discuss each concept more incrementally (cf section 3.1 in the updated manuscript);
> 2. we added a brief summary of our framework's insights in the new related work (cf section 6 in the updated manuscript);
> 3. we updated the first paragraph of the experiments to summarize our main experimental results (cf lines 325-329 in the updated manuscript);
> 4. we updated the first paragraph of the conclusion to summarize our main results (cf section 8 in the updated manuscript).
>
> ----
>
> > Have the authors evaluated the proposed approach on other datasets (common in computer vision or other public ones)?
>
> **In Appendix H1 we test many of our theoretical claims on CIFAR10. We will also add transfer experiments in the updated manuscript.**  In particular, we will evaluate our ImageNet pretrained DISSL and SwAV on the standard transfer suite from [1]. We note that although this is empirically interesting, transfer learning is not discussed in our theory.
>
> [1] S. Kornblith, J. Shlens, and Q. V. Le. “Do better imagenet models transfer better?” In: Conference on Computer Vision and Pattern Recognition (CVPR). 2019.
>
> ---
>
> > Could the authors comment more about how close the proposed theoretical foundation is to the practical cases?
>
> **As discussed in the last paragraph of the main paper (further detailed in Appendix F), the theory is only a simplification of reality.** Most importantly, we consider infinite unlabeled data for pretraining and arbitrarily large encoders.
>
> **Empirically, we nevertheless show in appendix H.1.** that fine-grained predictions of our theory (e.g. exact sample efficiency, dimensionality, …) hold in carefully constructed versions of standard contrastive learning settings  (CIFAR10, training data, known equivalence, and ISSL log loss). We have similar results for slightly more realistic settings (CIFAR10, test data, standard augmentations, and CISSL), which we will include in the updated appendices. **This suggests that our proposed theory can be surprisingly predictive even when the precise assumptions are violated.** The empirical gains from our actionable insights further support that claim.
>
> ---
>
> > The authors address the limitations of the paper exposition in Section 7, Regarding the potential negative societal impact might be nice that the authors comment/discuss about the theoretical connections/aspect of their framework related to learning of possible biases/"short cuts" that might be undesirable as well as misused.
>
> Our work highlights and clarifies the important role of equivalence classes and augmentation in the geometry of SSL representations. This suggests that to understand the theoretical and experimental impact of biases in SSL, we should understand the impact of augmentations on biases. In particular, there might be ways of reducing downstream biases by using the correct augmentations and invariances. We think that this is an important and interesting research direction, and hope that our paper will motivate future work on that topic.

---

> ### Comment · Area_Chair_TrRf · 2022-08-08
> **Please respond to author feedback**
>
> Thank you for reviewing this paper. Could you respond to the author feedback, or at least acknowledge that you've read the reply? Does the author reply address your concerns?
>
> Best, AC

---

### Official Review · Reviewer_uR7p · 2022-07-11

**Rating:** 5
**Confidence:** 3
**Soundness:** 2 fair
**Presentation:** 2 fair
**Contribution:** 2 fair

**Summary:**

This paper tends to figure out crucial properties of self-supervised learning (SSL) methods, which promote good performance on downstream tasks. To reach this target, the authors propose a unifying conceptual ISSL framework. In particular, their primary contributions contain: i) increasing the dimensionality of presentation and using asymmetric projection heads; ii) presenting a new non-contrastive ISSL objective; iii) applying non-linear probes.

**Questions:**

- Are there any different insights in your experiments? If yes, please elaborate the difference between this paper and [1,2].
- Can the authors discuss the difference between DINO and the proposed DISSL?

**Limitations:**

Yes, the authors have addressed the societal impact of their work.

**Strengths And Weaknesses:**

The motivation is clear, and I appreciate the heavy theoretical and empirical efforts.

However, my main doubts/concerns regarding the paper are the following:

- This paper is hard to follow because of the lack of organization. I strongly recommend that the paper needs significant improvement in writing.
- The novelty of the mentioned contributions is limited:
1. Increasing the dimensionality of presentation. The effect of dimensionality has been thoroughly discussed in previous literature [1,2]. Are there any different insights in your experiments? If yes, please elaborate the difference between this paper and [1,2].
2. Using asymmetric projection heads. In [3,4,5], they have theoretically and empirically discussed the impact of asymmetric projection heads.
3. Novel non-contrastive ISSL objective. To my knowledge, DINO[6] may first introduce "self-distillation" into self-supervised learning. The authors mention DINO in line 210, yet I did not find any analysis on comparison between DINO and DISSL. Can the authors discuss the difference between DINO and the proposed DISSL?
4. Non-linear probes. Although non-linear probes improve downstream tasks, they cannot serve as a property of SSL representation. Thus, this result can hardly support your original motivation.
5. Coarser augmentation. InfoMin [7] has thoroughly analyzed the effect of augmentations or view selections. Besides, they claimed that "we should reduce the mutual information (MI) between views while keeping task-relevant information intact.", which is nearly the same as your proposed "stronger label-preserving augmentations".
- In Table 3, a lack of comparisons with state-of-the-art methods mentioned in the "related work" section, including BYOL[3], DINO[7], MoCo[8], Simsiam[9].
- Inconsistent hyper-parameters between main paper and supplementary material. For instance, "3072 batch size" in Table 3 and "a total batch-size of 2560" in line 1034.

Minor:

- The experimental setting on TinyImageNet is not standard. I kindly suggest that you could follow some previous work [10,11,12] if the computation resources are constrained.
- Some informal theorems and algorithms. Take Algorithm 1 (line 237) for example, the inputs are required to be clearly described.

[1] T. Chen, et al., "A Simple Framework for Contrastive Learning of Visual Representations", ICML 2020.

[2] J. Zbontar, et al., "Barlow Twins: Self-Supervised Learning via Redundancy Reduction", CVPR 2022.

[3] J. Grill, et al., "Bootstrap your own latent: A new approach to self-supervised Learning", NeurIPS 2020.

[4] Y. Tian, et al., "Understanding Self-supervised Learning Dynamics without Contrastive Pairs", ICML 2021.

[5] X. Wang, et al., "On the Importance of Asymmetry for Siamese Representation Learning", CVPR 2022.

[6] M. Caron, et al., "Emerging Properties in Self-Supervised Vision Transformers", ICCV 2021.

[7] Y. Tian, et al., "What Makes for Good Views for Contrastive Learning?", NeurIPS 2020.

[8] K. He, et al., "Momentum Contrast for Unsupervised Visual Representation Learning", CVPR 2020.

[9] X. Chen, et al., "Exploring Simple Siamese Representation Learning", CVPR 2021.

[10] Y. Tian, et al., "Contrastive Multiview Coding", NeurIPS 2021.

[11] Y. Kalantidis, et al., "Hard Negative Mixing for Contrastive Learning", NeurIPS 2020.

[12] C. Chuang, et al., "Debiased Contrastive Learning", NeurIPS 2020.

---

> ### Author Response · Authors · 2022-07-28
> **Other**
>
> > The experimental setting on TinyImageNet is not standard. I kindly suggest that you could follow some previous work [10,11,12] if the computation resources are constrained.
>
> We thank the reviewer for their suggestion.  The cited papers [10,11,12] seem to use ImageNet-100, CIFAR10, and STL10 in their resource-constrained experiments. We already use CIFAR10 for our toy experiments in appendix H.1. For our ablations / controlled comparison, we followed the following prior works and used the TinyImageNet dataset:
> - Zheng, Mingkai, et al. "Ressl: Relational self-supervised learning with weak augmentation." NeurIPS (2021).
> - Patacchiola, Massimiliano, and Amos J. Storkey. "Self-supervised relational reasoning for representation learning." NeurIPS (2020).
> - Bhat, Prashant, Elahe Arani, and Bahram Zonooz. "Distill on the Go: Online knowledge distillation in self-supervised learning." CVPR (2021).
> - Ermolov, Aleksandr, et al. "Whitening for self-supervised representation learning." ICML (2021).
> - Miao, Ning, et al. "Learning Instance-Specific Data Augmentations." arXiv preprint arXiv:2206.00051 (2022).
>
> We would like to emphasize that our CIFAR10 and TinyImageNet experiments are used as controlled experiments, while our ImageNet experiments are the ones that are meant to be more comparable to previous work.
>
> ---
> > Non-linear probes. Although non-linear probes improve downstream tasks, they cannot serve as a property of SSL representation. Thus, this result can hardly support your original motivation.
>
> Disclaimer: we are not sure we correctly understood that concern.
>
> Our motivation is to improve our understanding and designing of SSL algorithms by characterizing the set of all optimal representations. In our theory we show that optimal representations depend on the family of the downstream probes, our experiments in table 2 are then meant to validate our theory. We think that this theory is useful for the following two aspects:
> - *Understanding SSL*: we characterize exactly how optimal representations (and projection heads) depend on the probing family, which to our knowledge is new.  For example, this explains why the common mutual information perspective of many SSL algorithms (eg [13,14,15]) empirically leads to worse algorithms [16]. Namely, mutual information does not take into account the probing family. Details in Appendix E.3.
> - *SSL algorithms for non-linear probes*: in real applications, where the goal is to achieve high downstream performance (rather than evaluation), it is natural to consider non-linear probes (see e.g., [17]). To our knowledge, we are the first to show how to design such algorithms.
>
> Papers:
> - [13] Oord, Aaron van den, Yazhe Li, and Oriol Vinyals. "Representation learning with contrastive predictive coding." arXiv preprint arXiv:1807.03748 (2018).
> - [14] Hjelm, R. Devon, et al. "Learning deep representations by mutual information estimation and maximization." ICLR (2018).
> - [15] Federici, Marco, et al. "Learning robust representations via multi-view information bottleneck." ICLR (2020).
> - [16] Tschannen, Michael, et al. "On mutual information maximization for representation learning." ICLR (2019).
> - [17] Dubois, Yann et al. “Lossy Compression for Lossless Prediction.” NeurIPS (2021).
>
> ---
> > In Table 3, a lack of comparisons with state-of-the-art methods mentioned in the "related work" section, including BYOL[3], DINO[7], MoCo[8], Simsiam[9].
>
> The main goal of table 3 is to test, in a non-toy setting, the gains due to our prescriptions compared to not using them. As a result, we compare our method to the most comparable ones: SimCLR vs CISSL and DISSL vs SwAV with the same batch-size and ResNet50. Other methods are less comparable, for example, DINO is typically trained on ViTs at this scale, while MoCo and SimSiam use much smaller batch sizes.
>
> A comparison to better methods is given in Table 4, where we compare DISSL and SwAV + multi-crop, which typically outperforms all the methods the reviewe cited (see table 1 in [19]).
>
> [19] Zbontar, Jure, et al. "Barlow twins: Self-supervised learning via redundancy reduction." International Conference on Machine Learning. PMLR, 2021.
>
> ---
> > This paper is hard to follow because of the lack of organization. I strongly recommend that the paper needs significant improvement in writing.
>
> We are happy to address any specific writing comments outside of the ones addressed above.
>
> ---
> > [...] "3072 batch size" in Table 3 and "a total batch-size of 2560" in line 1034.
>
> We thank the reviewer for noticing this typo. The batch size is 2560 for table 3 and 3072 for table 4.
>
> ---
> > Take Algorithm 1 (line 237) for example, the inputs are required to be clearly described.
>
> The symbols used for each input were already described in the main text. As suggested, the algorithm in the updated manuscripts now contains a short description for each symbol to improve readability.

---

> ### Author Response · Authors · 2022-07-28
> **Augmentations: we do not claim novelty of empirical observation and discuss InfoMin in Appendix E.1.**
>
> > Coarser augmentation. InfoMin [7] has thoroughly analyzed the effect of augmentations or view selections. [...]
>
> We agree that the use of coarser augmentations has already been suggested in previous work, which is why **we do not claim it to be a novel empirical observation and do not use coarser augmentations in our ImageNet experiments. In fact, we discuss InfoMin and other similar papers in Appendix E.1** (lines 781-785). We agree this is an important discussion and moved it to the main paper (cf lines 308-310 of the updated manuscript).
>
> Essentially our coarser augmentation experiment (figure 7a) is simply a validation of our theory (proposition 2), which to our knowledge is the first to give an exact relation between optimal sample efficiency and the number of equivalence classes. Here is the discussion from lines 781-785 in appendices:
>
> >> Many prior work have suggested that a good augmentation or view is one that is information preserving while removing as much nuisance information as possible [2, 65–69]. Prop. 2 (and Appx. B.3) can be seen as a new perspective on why coarser augmentations are useful, by proving the exact relation between optimal sample efficiency and the number of equivalences.

---

> ### Author Response · Authors · 2022-07-28
> **DINO: DINO and DISSL are different and discussed in Appendix E.**
>
> > Novel non-contrastive ISSL objective. [...] yet I did not find any analysis on comparison between DINO and DISSL. Can the authors discuss the difference between DINO and the proposed DISSL?
>
> **We previously discussed DINO vs DISSL in Appendix E** (lines 825-834). We agree this is important and now summarize this discussion in the main paper (cf lines 295-301 of the updated manuscript).
>
> Essentially **DISSL can be seen as a theoretically motivated and simplified DINO (removes EMA/stop-grad/centering) that also works better in practice (41.7% -> 49.7% on TinyImageNet in table 1)**. Here is the discussion from lines 825-834 in appendices:
>
> >> DINO: The main difference in the student is that they use a non-linear projection head before the softmax, which as discussed in Sec. 4.1 does not ensure linear predictability of downstream tasks. For the teacher, DINO aims at ensuring maximality by setting the teacher to the exponential moving average of the student, stopping the gradients, and applying some centering. In contrast, DISSL does not require any optimization trick and enforces maximality by maximizing the entropy of the teacher’s output (assuming uniform prior).

---

> ### Author Response · Authors · 2022-07-28
> **Asymmetry: previous work consider a different asymmetry and for unrelated reasons.**
>
> > Using asymmetric projection heads. In [3,4,5], they have theoretically and empirically discussed the impact of asymmetric projection heads.
>
> **The motivation, practical instantiation, and type of theoretical results of previous work on asymmetric SSL (including the cited [3,4,5]) are unrelated to ours**. In particular, [3,4] consider various asymmetry between the teacher and student (EMA, additional predictor, stop-grad) to **avoid collapsing in non-contrastive SSL**. In contrast, we show what projection head to use in any SSL method (including contrastive) depending on the probing family to allow **optimal downstream performance**. Figure 2a of our appendices shows that the gains are not due to asymmetry per se but the exact and new asymmetry we argue for.
>
> In the case of linear probing, our theory suggests dropping one of the two projection heads. To our knowledge, we are the first to do so. In contrast, the asymmetry in BYOL [3,4] consists in using EMA, stop-gradients, and a non-linear predictor head in addition to both non-linear projection heads. The asymmetry in [5], concerns augmentations and batch-norms but not the architecture of the projection head.
>
> We thank the reviewer for pointing out this potential point of confusion and now explicitly mention it in the related work section (cf line 315 of the updated manuscript).

---

> ### Author Response · Authors · 2022-07-28
> **Dimensionality: previous work analyze the dimensionality of a different vector than us.**
>
> > Increasing the dimensionality of representations. The effect of dimensionality has been thoroughly discussed in previous literature [1,2].
>
> In SSL it is typical to use a projection head during pretraining, which is discarded in downstream tasks. The dimensionality of the representation (before the projection head) is then unrelated to the dimensionality of the projector’s output, i.e., one can change one without changing the other.
>
> **In particular, the cited papers ([1,2]) empirically analyze the effect of the dimensionality of the output of the projector. In contrast, we theoretically+empirically analyze the dimensionality of the representation**. Note that we already use the best dimensionality of the projector’s output from [1], i.e, the gains are orthogonal and in addition to the ones seen in previous work.
>
> From the cited papers:
> - [1] In SimCLR figure 8: “The representation h (before projection) is 2048-dimensional here.”
> - [2] In Barlow Twins:  “This result is quite surprising because the output of the ResNet is kept fixed to 2048”
>
> We thank the reviewer for pointing out this potential misunderstanding. To avoid confusion, we now explicitly mention in the related work section that we are not considering the projector’s dimensionality (cf line 304 of the updated manuscript).

---

> ### Author Response · Authors · 2022-07-28
> **General: we updated the main paper to avoid potential misunderstandings.**
>
> We thank the reviewer for taking the time to write down all their doubts about the paper. This helped us identify points that should be made more explicit to avoid possible misunderstandings.  We will update the manuscript by highlighting those points in the main paper, which should improve our manuscript.
>
> The main doubts seem to concern whether our actionable insights (dimensions, asymmetry, DISSL objective, augmentations) are already discussed in the cited works. In the following posts, we show that **those works are either only superficially related or were already discussed in our paper**. In particular, we study a different dimensionality (representation vs projector’s output) and asymmetry than previous work, and discuss in details DINO and coarser augmentations in Appendix E.1. We now clarify those points in the main paper (cf section 6 in the updated manuscript).
>
> Furthermore, we would like to emphasize that our **primary contributions concern a simple unifying theoretical framework** to derive from first principles prescriptions for SSL. The practical prescriptions above fall directly out of our unifying framework, and are simply meant to support the usefulness of that framework. This is why we focus on controlled experiments to show the empirical improvements of our prescriptions. Our hope is that this will motivate others to use our framework to derive new actionable insights about ISSL.

---

> > ### Comment · Reviewer_uR7p · 2022-08-06
> > **Follow up**
> >
> > I thank the authors for addressing most of my above-mentioned concerns. I appreciate your empirical and theoretical efforts. I am increasing my rating accordingly.
> >
> > However, I still have some concerns about the non-linear probes. To my knowledge, the linear probe is one way to evaluate the quality of the pre-trained representation. Of course, the extra non-linear operator will improve the downstream performance. My main concern is about the connection between the ideal representation for self-supervised learning (i.e., your motivation) and non-linear probes. It would be better to see more discussion about the connection.

---

> > > ### Author Response · Authors · 2022-08-08
> > > **Non-linear probes are an extension of our framework mostly discussed in appendices.**
> > >
> > > Thank you for taking the time to engage in a discussion and for taking into account our rebuttal.
> > >
> > > ---
> > > Concerning probes: **we would like to highlight that most of our paper focuses on linear probes and this is the main contribution of our paper**. In particular, we consider linear probes for all our theory in section 3, all our algorithmic insights in section 4, and our main experiments (table 1,3,4) including all ImageNet experiments.
> > >
> > > We emphasize that the **extension of our framework to any desired (eg MLP or linear) probes: (1) is mostly done in the appendices, and (2) is strictly more general than the linear framework, i.e., we do not lose anything from it**.  So if someone does not care about linear probe they can simply ignore Appx B.5 and its short summary in section 5.
> > >
> > > Concerning your specific question:
> > > > My main concern is about the connection between the ideal representation for self-supervised learning (i.e., your motivation) and non-linear probes. It would be better to see more discussion about the connection.
> > >
> > > **Our approach to defining ideal representations is to take into account how they would be used in downstream tasks, which is why optimal representations (definition 3) depend on the desired probing family $\mathcal{F}$**. For example, if practitioners use linear probes then they should consider “ideal representations for linear probes” but if they are interested in MLP probes then they should consider “ideal representations for MLP probes”.
> > >
> > >
> > > We thank the reviewer for flagging this potential misunderstanding and will emphasize in the updated manuscript that non-linear probes are simply an extension of our framework and that we define ideal representations depending on how they will be used in downstream tasks.

---

> ### Author Response · Authors · 2022-08-05
> **Any other concerns?**
>
> Dear reviewer,
>
> Given that the discussion period is quickly progressing we were wondering whether you had any follow-up questions that we could answer. We believe that we clarified your doubts in the updated manuscript but any pointers to where our revision is lacking would be highly appreciated to improve our paper (changes are highlighted in blue). We would also be happy to engage in a discussion to address any other concerns that you might have.
>
> Thank you for your time reviewing our paper.

---

### Official Review · Reviewer_bMPi · 2022-07-21

**Rating:** 7
**Confidence:** 4
**Soundness:** 3 good
**Presentation:** 3 good
**Contribution:** 3 good

**Summary:**

Characterizing the desirable properties for representations learned via self-supervised learning objectives,  presents open theoretical and empirical questions, particularly in the context of downstream classification tasks. The authors propose a framework to study popular contrastive and non-contrastive learning algorithms as invariant-SSL (iSSL). Under some technical assumptions, they present certain properties of population-optimal and sample-optimal feature encoders (c.f. Eq 2, Sec 3) for evaluation with linear probes. Building on these characteristics, the paper presents practical design primitives to learn encoders with such properties. In particular, with approximations to a log-loss training objective, and different estimators to approximate the true loss, the framework recovers popular SSL objectives (SimCLR, Dino). Finally, these estimators are evaluated via experiments on the Tiny Imagenet dataset, where ISSL motivated modifications translate to ~3% improvement in both regimes of pretraining. Notably, Distillation ISSL is a simplified training algorithm which recovers representations which are competitive to, and often better than strong baselines (SwAV).

**Questions:**

A couple of questions for the authors:
1. How well do the estimators approximate the desired quantities (e.g. DISSL approximates $M(x)$) ? For instance, are there controlled experiments with toy datasets to demonstrate the effectiveness of the auxillary losses?

**Limitations:**

The authors are encouraged to consider relaxing some of the strict assumptions, namely:
1. Coarse-grained effects of data-augmentation operators (the operators are often smooth, low-variance).
2. While the authors demonstrate that stronger data-augmentations improve in-distribution generalization, I think it also important to consider out-of-distribution generalization (i.e. performance on different tasks).

**Strengths And Weaknesses:**

Strengths:
+ The paper presents a novel framework (ISSL) to study the paradigm of contrastive learning, SSL pretraining objectives. Notably, the authors study invariance to data-augmentation with downstream classification tasks and use ISSL to find simplifications to popular SSL algorithms (e.g. SimCLR with asymmetric projection head, Distillation SSL as an alternative to SwAV).
+ The authors perform extensive ablations on TinyImagenet dataset and compare to strong baselines to establish the efficacy of insights from there proposed framework. The simplifed learning objectives could further encourage research into more fine-grained questions about these training algorithms (such as learning dynamics, sample efficiency etc).


Weaknesses:
- Beyond evaluation on linear probing, the authors are encouraged to present visualizations, discussion on emergent structure in representations learned with their proposed algorithm as compared to prior work.
- Certain assumptions in the framework are too restrictive or unrealistic (such as label being either invariant to class of augmentations, or not i.e. binary effects of the data-augmentations).

---

> ### Author Response · Authors · 2022-08-02
> **Some results in appendices answer some of the questions and we will add the suggested other experiments.**
>
> **We thank the reviewer for their insightful feedback and additional ideas for our experimental section. We will update the paper with all the experimental results the reviewer asked for**, which we think will improve the final manuscript.
>
> ---
> > While the authors demonstrate that stronger data-augmentations improve in-distribution generalization, I think it also important to consider out-of-distribution generalization (i.e. performance on different tasks).
>
> We had not tried any out-of-distribution (OOD) experiments given that our theory does not concern this setting. We nevertheless agree that this is a natural question to ask for any novel SSL method and **we will thus include a comparison of the ImageNet-pretrained DISSL vs SwAV on the standard OOD transfer suite** from [1].
>
> [1] S. Kornblith, J. Shlens, and Q. V. Le. “Do better imagenet models transfer better?” In: Conference on Computer Vision and Pattern Recognition (CVPR). 2019.
>
> ---
>
> > How well do the estimators approximate the desired quantities (e.g. DISSL approximates M(X) ) ? For instance, are there controlled experiments with toy datasets to demonstrate the effectiveness of the auxillary losses?
>
> This is an interesting question and a good suggestion. **We will add an experiment that does exactly that**, namely, testing how well DISSL recovers M(X) in a controlled setting (CIFAR10 with supervised augmentations).
>
> **We would like to highlight that appendix H.1. already shows many controlled experiments to evaluate our theory** (CIFAR10, training data, known equivalence, and ISSL log loss). We have similar results for slightly more realistic settings (CIFAR10, test data, standard augmentations, and CISSL), which we will add in the updated appendices.
>
> ---
>
> > Certain assumptions in the framework are too restrictive or unrealistic (such as label being either invariant to class of augmentations, or not i.e. binary effects of the data-augmentations).
> > The authors are encouraged to consider relaxing some of the strict assumptions, namely coarse-grained effects of data-augmentation operators (the operators are often smooth, low-variance).
>
> Disclaimer: we are not sure we understand which assumption the second point is referring to. We wrote our response under the interpretation that this was referring to augmentations being either label-preserving or not.
>
> Considering (binary) equivalences is a simplification that we are aware of, and we discuss it in the last section of the main paper. The controlled experiments in Appendix H.1. nevertheless show that this assumption (and others) matter less than one might initially think. In particular, **Figure 3 in appendices shows that standard augmentations seem to follow a similar trend** (for sample efficiency and dimensionality) as controlled augmentations for which the binary equivalence assumption holds by construction.
>
> **In general, our theory is purposely simple to make it easy to reason about, while getting at the core insights.** There are thus important simplifications, which we discuss in detail in the last section of the paper and appendix F. **Despite those simplifications, appendix H.1. shows that our theory closely predicts results on real data** but controlled settings (e.g. it predicts exactly the required number of samples, dimensionality, or projection heads). As said above, we have similar results for more realistic settings (CIFAR10, test data, standard augmentations, and CISSL), which we will add in the updated appendices.
>
> ---
>
> > Beyond evaluation on linear probing, the authors are encouraged to present visualizations, discussion on emergent structure in representations learned with their proposed algorithm as compared to prior work.
>
> **Figure 5a in our appendices already partially investigates the geometry of our learned representations**. Namely, it shows that, as suggested by Proposition 3, one learns a representation that is close to an ETF (see figure 4 to see what those representations look like). We have similar plots that show that representations learned by baselines (eg DINO) also get closer to ETFs during training but not as quickly and never as close as our models (eg DISSL). We will add such plot in Appendix H.3.
>
> We initially did not include dimensionality reduction visualizations due to the sensitivity of such visualizations to hyperparameters, but **we will include UMAP plots** with the appropriate caveats in the appendix.

---

> ### Comment · Area_Chair_TrRf · 2022-08-08
> **Please respond to author feedback**
>
> Thank you for reviewing this paper. Could you respond to the author feedback, or at least acknowledge that you've read the reply? Does the author reply address your concerns?
>
> Best, AC

---

### Author Response · Authors · 2022-08-02
**Response summary**

We thank the reviewers for their insightful feedback that will help us improve our manuscript.

We are glad that the reviewers found our paper generally well written and of high quality [j7cH,L1ta], well-motivated [uR7p,j7cH,L1ta], and the related work and limitations discussed in detail [j7cH,L1ta]. We are also happy that the reviewers recognized the simplicity [bMPi,L1ta] yet strong empirical performance of our algorithms compared to previous work [j7cH,bMPi,L1ta]. Finally, we are thankful that [j7cH,L1ta] recognized the strength of the theoretical foundation of our work, which we view as an important contribution.

Reviewers' feedback mostly concerned highlighting important discussions in the appendices and adding specific experiments. We will take advantage of the additional content page to incorporate this feedback. In particular, as suggested by various reviewers, we will update the manuscript with the following.
- **Summarizing related work from appendices E to main paper [uR7p,L1ta].** In particular, we will discuss previous work related to our actionable insights (dimensionality, augmentations, asymmetric head, DISSL) and our theory.
- **Transfer experiments [bMPi,j7cH,L1ta].** We will evaluate our algorithms on OOD/transfer.
- **Additional targeted experiments [j7cH,bMPi].** We will extend the controlled experiments from Appendix H.1. with experiments that investigate the quality of our objective and assumptions in controlled but toy settings.

**We already uploaded the updated manuscript with all the changes to the writing (text highlighted in blue means that it was suggested by a reviewer).
We will upload another updated manuscript with the new experiments as soon as we get the results.** We thank the reviewer for their feedback as we believe that this will improve our manuscript.

---

### Author Response · Authors · 2022-08-09
**We updated the paper and appendices with suggested experiments.**

We uploaded the updated manuscript and appendixes with every additional experiment that reviewers have suggested.

In particular, we added to the main paper the following results that compare DISSL and SwAV on the standard transfer learning benchmarks as suggested by [bMPi,j7cH,L1ta].

|           |   Food   |  CIFAR10 | CIFAR100 |   CARS   | Aircrafts | DTD      | Pets     | Caltech  | Flowers  |
|-----------|:--------:|:--------:|:--------:|:--------:|:---------:|----------|----------|----------|----------|
| SWAV      |   75.5   |   92.0   |   76.3   |   58.2   |  **49.1** | 72.6     | 86.9     | **92.0** | 94.7     |
| **DISSL** | **77.9** | **93.6** | **77.6** | **62.2** |    48.1   | **73.9** | **88.0** | 91.5     | **95.3** |

We further added the following targeted experiments/results to the appendices:
- UMAP visualization of ImageNet representations  ([see this link](https://i.postimg.cc/BbLtGWSh/UMAP-dissl.png)) as suggested by [bMPi]
- toy CIFAR10 experiments showing that DISSL recovers M(X) ([see this link](https://i.postimg.cc/5ybp7MwC/confusion-matrix.png)) suggested by [bMPi]
- An appendix H.2. that shows that the fine-grained predictions (eg exact required dimensionality and samples) of our theory generally hold outside of idealized settings

We thank the reviewer for those suggestions as we think that this further improved the experimental results of our paper.

---

### Meta-Review · Area_Chair_TrRf · 2022-08-20

**Recommendation:** Accept
**Confidence:** Certain

**Metareview:**

Decision: Accept

This paper provides a theoretical analysis on the invariance properties of representation learned by self-supervised learning, and derived an algorithmic framework from the theory which includes approaches similar to existing SSL methods. Empirical results demonstrate the competitiveness of the proposed approach.

Reviewers found the proposed approach simple but effective, and the related work and limitations are discussed. Initially there are concerns regarding novelty & comparisons to previous work, which are largely addressed in author feedback & revision.

After reviewer-AC discussions, it is concluded that we should accept this paper. In revision for camera ready, I'd suggest the authors to follow what they promised in their summary feedback, and improve the clarity of the presentation, especially on section 4 as suggested by one of the reviewers. The additional results provided by the authors in feedback period should also be added.

**Award:**

No

---

### Decision · Program_Chairs · 2022-09-14

Accept